# Janus microparticles-based targeted and spatially-controlled piezoelectric neural stimulation via low-intensity focused ultrasound

Mertcan Han [1,2], Erdost Yildiz [1], Ugur Bozuyuk [1], Asli Aydin [1,3], Yan Yu[1], Aarushi Bhargava [1], Selcan Karaz[1,2] & Metin Sitti [1,2,4] ✉

Electrical stimulation is a fundamental tool in studying neural circuits, treating neurological diseases, and advancing regenerative medicine. Injectable, free-standing piezoelectric particle systems have emerged as non-genetic and wireless alternatives for electrode-based tethered stimulation systems. However, achieving cell-specific and high-frequency piezoelectric neural stimulation remains challenging due to high-intensity thresholds, non-specific diffusion, and internalization of particles. Here, we develop cell-sized 20 μm-diameter silica-based piezoelectric magnetic Janus microparticles (PEMPs), enabling clinically-relevant high-frequency neural stimulation of primary neurons under low-intensity focused ultrasound. Owing to its functionally anisotropic design, half of the PEMP acts as a piezoelectric electrode via conjugated barium titanate nanoparticles to induce electrical stimulation, while the nickel-gold nanofilm-coated magnetic half provides spatial and orientational control on neural stimulation via external uniform rotating magnetic fields. Furthermore, surface functionalization with targeting antibodies enables cell-specific binding/targeting and stimulation of dopaminergic neurons. Taking advantage of such functionalities, the PEMP design offers unique features towards wireless neural stimulation for minimally invasive treatment of neurological diseases.

Electrical stimulation and modulation of the nervous system is the basis of various clinically-approved tools for treating neurological diseases[1,2], sensory impairments[3–5], and movement disorders[6]. However, existing clinical approaches often involve the implantation of stationary, rigid, metal electrodes with limited spatial resolution, low selectivity[7], and potential long-term side effects[8]. Despite the recent advances in transducer micro/nanoparticle systems that convert magnetic[9–12], optical[5,13–15], or mechanical[16–23] energy into bioelectrical

modulation, and their integration with genetically encoded proteins, there are still fundamental challenges to overcome[7]. Particularly, required genetic modifications, undesired diffusion of the nanoparticles away from the target cells[24], accumulation of nanoparticles (NPs) in off-target tissues[25], and rapid decay of the generated electric field in the proximity of nanoparticles make their stimulation performance unreliable[7] and highly dependent on concentration. The latter also causes high-intensity excitation thresholds to achieve neural

[1]Physical Intelligence Department, Max Planck Institute for Intelligent Systems, 70569 Stuttgart, Germany. [2]Institute for Biomedical Engineering, ETH Zurich, 8092 Zurich, Switzerland. [3]Department of Neurosurgery, Maastricht University Medical Centre, Maastricht, Netherlands. [4]School of Medicine and College of Engineering, Koç University, 34450 Istanbul, Türkiye. ✉e-mail: sitti@is.mpg.de

stimulation at clinically-relevant frequencies (50–200 Hz)[9,26]. Hence, it is crucial to develop new strategies that address the aforementioned challenges and ensure the successful implementation of particle-based systems in electrical stimulation applications. These strategies should aim to achieve both particle and electric field confinement while enabling precise control over the temporal and spatial characteristics of neuromodulation.

Here we report piezoelectric magnetic Janus microparticles (PEMPs), at the size scale of neural cells, for wireless low-intensity focused ultrasound (LIFU)-mediated neural stimulation at therapeutic frequencies (Fig. 1a, left). PEMPs are composed of 20 μm-diameter spherical, porous silica microparticles with a BaTiO$_3$ nanoparticle (BTNP)-conjugated half-surface and a magnetically-responsive half-surface for on-demand locomotion and reorientation via external uniform magnetic fields. We evaluated and verified the safety and neural stimulation performance of PEMPs in vitro under LIFU via patch-clamp electrophysiology recordings on primary neurons. Furthermore, the population response and spatial neural stimulation characteristics for a single PEMP were identified. Owing to the asymmetric design of PEMPs and the confinement of the electric field on BTNP-conjugated half surface, we showed four distinct features: (1) low threshold ultrasound intensity (<100 mW.cm$^{-2}$), (2) high-frequency (up to 200 Hz) neuromodulation, (3) orientational and positional control of the stimulator particle, and (4) cell-specific neural stimulation capability on dopaminergic neurons by targeting GIRK2 antibodies. Therefore, by rendering piezoelectric nanoparticles that would normally be dispersed in the extracellular matrix into a microparticle surface and by having locomotion ability, we can either steer the PEMPs towards their target cells, control effective stimulation area, and stimulate the target cells, or we can modify their surface with targeting antibodies to enable selective stimulation of specific cell types. This proof-of-concept PEMP design paves the way for achieving non-genetic piezoelectric neural stimulation under low-intensity ultrasound with high spatiotemporal resolution and on-demand control, with potential implications for basic neuroscience research and neurotherapeutic applications.

## Results

### Design and fabrication of piezoelectric magnetic Janus microparticles (PEMPs)

The PEMP design comprises a magnetically-responsive Ni/Au thin film on the half side and bioconjugated piezoelectric BTNPs on the other half for electrical stimulation of neurons (Fig. 1a). Our main fabrication scheme (Fig. 1b) sequentially consists of the magnetic Janus microparticle (MP) fabrication, the magnetization of MPs, surface functionalization, and bioconjugation of BTNPs. To accomplish that, we rationally chose non-conductive porous spherical silica microparticles of 20 μm in diameter (Supplementary Fig. 1a) since the porosity increases the surface area of the microparticle. Thus, we could bind a higher amount of BTNPs, and the 20 μm diameter results in cell-sized structures. The metallic thin films were fabricated onto half microparticle surface by sequential deposition of Ni (60-nm-thick) and Au (20-nm-thick) (Fig. 1b), where Au was utilized as the coating film for preventing oxidation on the Ni surface and promoting biocompatibility[27]. By using a 1.8 T uniform magnetic field inside a vibrating sample magnetometer (VSM)[27], the magnetization direction of the microparticles was preprogrammed to the out-of-plane direction (Fig. 1a), where the Ni film provided the coercivity of 13.8 mT. Therefore, the magnetic thin film behaved like a hard-magnetic material under the 10 mT uniform field, which was utilized in the study for magnetic actuation (Fig. 1c, d).

Next, we utilized the bioconjugation of BTNPs onto the non-magnetic silica half-surface for facile and high-throughput fabrication of PEMPs. At first, amino groups were grafted onto the tetragonal ~260-nm-diameter BTNPs (Fig. 1e, Supplementary Fig. 1b, c) and magnetic Janus microparticles using 3-aminopropyltrimethoxysilane (APTES). This enabled the conjugation of biotinylated N-hydroxysuccinimide (NHS) to the amino groups and resulting modified BTNPs with ~433 nm hydrodynamic radius (Supplementary Fig. 1d), allowing the binding of biotin-conjugated BTNPs via biotin-avidin-biotin coupling[27,28] between the microparticle surface and BTNPs (Fig. 1b (iii)). The conjugation concentration of BTNPs was optimized for the lowest amount that covers the whole microparticle half-surface, verified under a scanning electron microscope (SEM) using energy dispersive X-ray analysis (EDX) (Fig. 1f) and also by bright field and fluorescence microscopy (Supplementary Fig. 2a, b). The simple, yet effective, bioconjugation strategy enabled the bulk fabrication of PEMPs, scalability of the overall fabrication procedure, and further biochemical functionalities, such as loading various cargo and targeting antibodies[27,29–31].

### Estimation of single PEMP behavior under LIFU

We started our investigations by measuring the single PEMP response in an interconnect-free configuration[32,33] using a patch clamp system under LIFU. To accomplish that, we built a measurement system consisting of a patch-clamp amplifier system integrated into an upright fluorescence microscope, a water tank, and a focused ultrasound probe with a 2 MHz center frequency (Supplementary Fig. 3). We first prepared a very low concentration of PEMPs in live cell imaging solution put single particles inside of pulled glass patch pipettes, and mounted them onto the patch-clamp system (Supplementary Fig. 4a). The measurement pipette was positioned on the bottom surface of the recording chamber to its position to be used in vitro experiments on primary neurons. The system voltage is held at zero in voltage-clamp mode to generate the virtual ground in the system. When the FUS pulses were applied to the recording chamber without any particles, we did not observe any current deviation from the baseline signal (Supplementary Fig. 4b). Later, a single PEMP was enclosed and LIFU intensity of 10, 20, 50, and 100 mW.cm$^{-2}$ was applied while keeping the excitation period at 10 ms (Supplementary Fig. 4c). We observed unipolar piezo-electrochemical current with a rising period and sustained current generation, particularly under LIFU intensity of >20 mW.cm$^{-2}$. Following the end of the LIFU pulse, the generated current decays and reaches the baseline before the stimulation pulse. Therefore, we could hypothesize that the piezoelectric current generation suggests Faradaic processes[34], which could be due to the electron transfer between BTNPs and the ionic extracellular medium. As no current generation under LIFU without PEMPs was observed and the sustained current levels increased with higher LIFU intensity, we could claim that recorded current transients represent piezoelectric current generation. In addition, current transients may not fully represent the actual piezoelectric currents in the ionic media since the low amplitude current peaks might be screened in the extracellular medium and fast current peaks will be inevitably filtered out due to the system bandwidth and via low pass filters during signal amplification. Once the piezoelectric current generation of PEMPs was explored, we proceeded to electrophysiology experiments to evaluate the neural modulation potential of PEMPs.

### In vitro piezoelectric neural stimulation

We conducted in vitro electrophysiology experiments (Fig. 2a top) to evaluate the performance of our PEMPs in converting LIFU into bioelectrical modulation. We focused on primary hippocampal neurons as a model system for our initial evaluation. To assess the effectiveness of piezoelectric charging in inducing repeatable depolarization of the neural membrane, we performed patch-clamp electrophysiology experiments. Firstly, the membrane potential of primary neurons was recorded under LIFU excitation to evaluate any possible effects due to LIFU or the presence of magnetic silica microparticles (MPs) without any BTNPs (Supplementary Fig. 5). As the control experiments revealed no significant response induced by LIFU or MPs, we

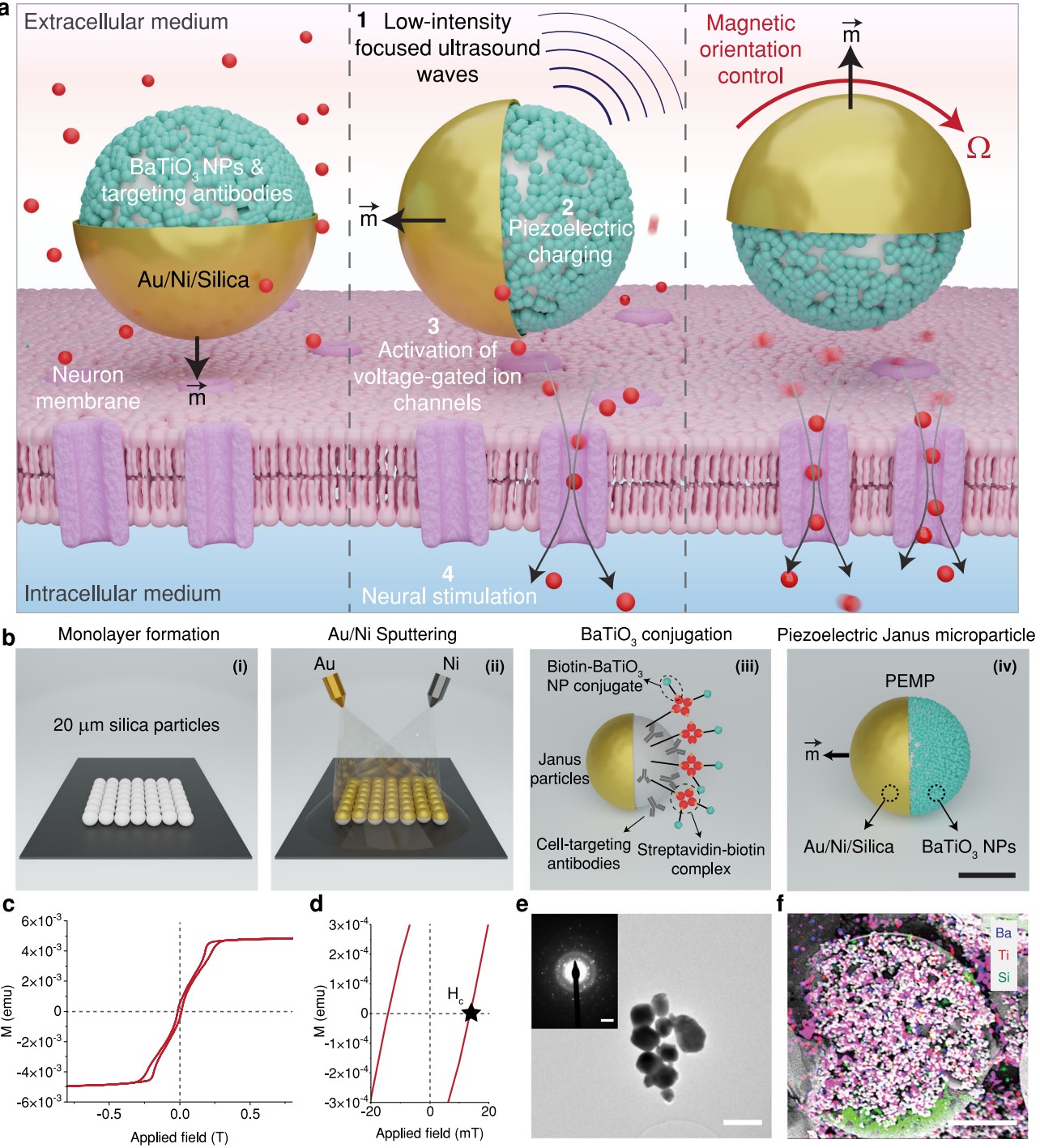

**Fig. 1 | Design, fabrication, and working principle of piezoelectric magnetic Janus microparticles (PEMPs) for wireless neural stimulation. a** Three possible scenarios of PEMPs in the extracellular matrix. (left) The PEMPs could be utilized either in the freestanding mode in the extracellular medium or the cell-attached mode targeted for specific cell types of interest. The Au/Ni coated half surface provides the magnetically actuated locomotion and steering capability, while the BaTiO$_3$ nanoparticle (BTNP) conjugated other half acts as the biointerfaces for neural stimulation. (middle) Upon low-intensity focused ultrasound excitation (1), BTNPs generate piezoelectric charging in the extracellular space (2), and this charging induces depolarization in the cell membrane and activation of voltage-gated ion channels (3), which generates the neural stimulation (4). (right) Moreover, the orientation, and consequently the piezoelectric field profile could be modified and reoriented by electromagnetic actuation of PEMPs, which controls the spatial neural stimulation profile. While the neural stimulation performance is the lowest for the BTNP-coated surface on top (left), it increases after rotation

(middle) and reaches its maximum for the configuration on the right. **b** Fabrication scheme of PEMPs. Fabrication starts with the monolayer formation of silica microparticles (i), followed by the sequential sputtering of Ni and Au thin films (ii), and magnetization. The other half surface was functionalized and bioconjugated with BTNPs (iii). The final particle has a premagnetized surface for magnetic steering and orientation control, and the BTNP conjugated part provides piezo-electric charging for neural stimulation (iv). Scale bar, 10 μm. **c** Hysteresis loop of the Ni film of 60 nm sputter on the PEMP. **d** The inset shows the region indicating the coercivity, H$_c$, of 13.8 mT. **e** Transmission electron microscope (TEM) image and the diffraction pattern of BTNPs. Scale bar, 300 nm. **f** Scanning electron microscope (SEM) image with energy dispersive X-ray analysis (EDX) of a single PEMP. Scale bar, 10 μm. For (**e**) and (**f**), TEM and SEM imaging utilized 3 and 5 times, respectively for independent batch of fabricated particles. No significant difference was observed between the batches.

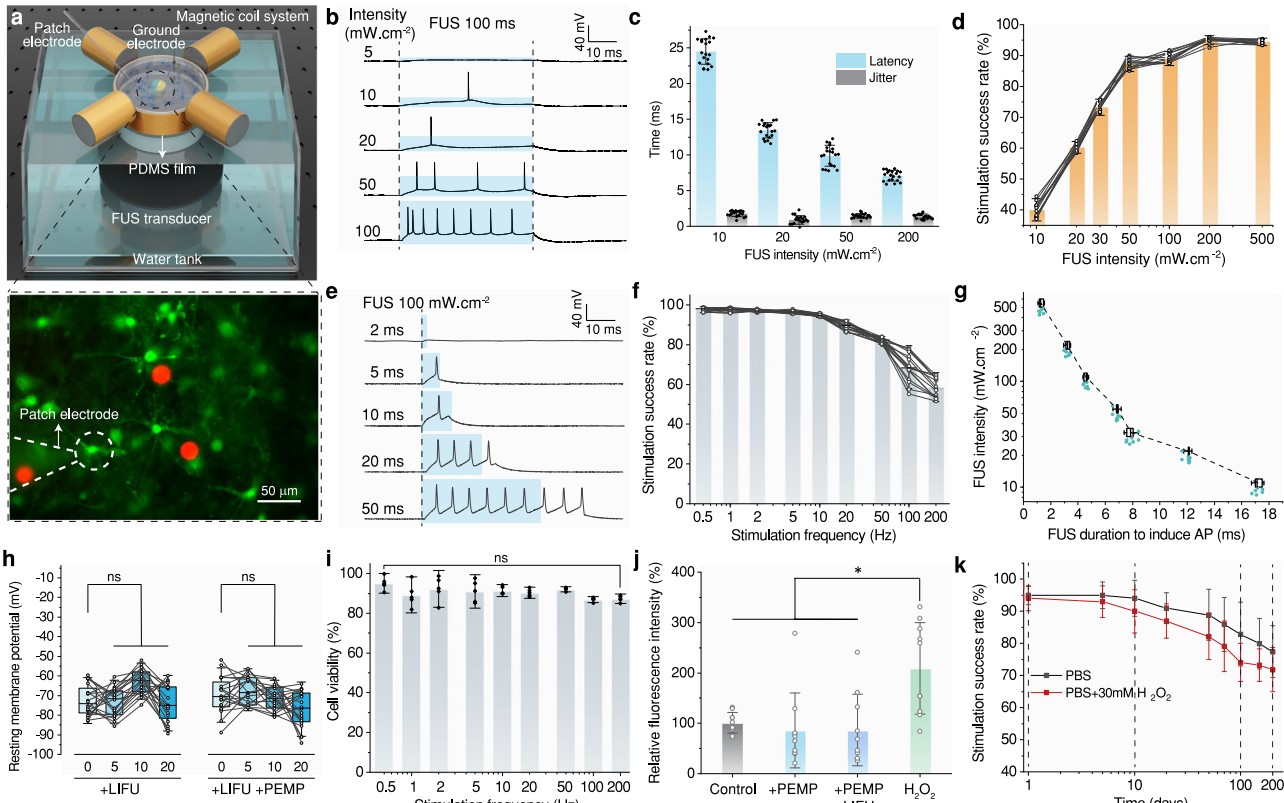

**Fig. 2 | Evaluating piezoelectric neural stimulation on primary hippocampal neurons in vitro. a** Experimental schematic of focused-ultrasound (FUS) integrated patch-clamp electrophysiology system with the three-axis five-coiled electromagnetic coil system for magnetic locomotion and orientation control. The bottom shows a representative patch-clamped primary neuron and PEMPs, and similar field-of-views were chosen for all experiments. Scale bar, 50 μm. **b** Representative membrane potential traces of primary neurons excited with PEMP under 100 ms FUS with increasing intensities. **c** The mean latency of the action potential peaks and the jitter as the standard deviation of latencies of all measured neurons under changing FUS excitation intensities ($n = 20$). For each neuron latency and jitter were calculated by 20 independent FUS pulses. **d** Neural stimulation success of PEMPs under 100 ms FUS under different intensities. The connected dots represent a single neuron data for at least three trials ($n = 12$). **e** Representative membrane potential traces of primary neurons excited with 100 mW.cm⁻², 2 MHz FUS with changing burst durations. **f** Neural stimulation success of PEMPs under 100 mW.cm⁻², 2 MHz FUS with changing burst stimulation frequency. The connected dots represent a single neuron data for each trial ($n = 16$ neurons examined over 4 independent experiments). **g** Excitability curve for primary neurons ($n = 6$) under PEMPs excited with changing FUS intensity and duration. Data are presented as mean ± s.d. **h** Resting membrane potentials of primary neurons after 20 min of excitation periods under continuous 100 mW.cm⁻² FUS excitation with and without PEMPs in the extracellular medium ($n = 20$ different neurons). Box plot limits present the 25th and 75th percentile from the mean line and whiskers represent the outliers (coefficient 1.5) of the distribution. **i** Cell viability using the LIVE/DEAD™ cell imaging kit assessed after 20 min, 100 mW.cm⁻² 2 MHz FUS with changing burst stimulation frequency ($n = 5$ independent experiments). **j** Quantification of reactive oxygen species (ROS) intake of primary neurons subjecting the PEMPs and continuous 200 mW.cm⁻² FUS excitation for 20 min (mean ± s.d. for $n = 9$ different neuron slides, control: $p = 0.0069$, PEMP: $p = 0.0018$, PEMP + LIFU: $p = 0.0019$, two-tailed unpaired t-test). **k** Functional stability test for PEMPs stored at 37 °C in PBS and 30 mM H₂O₂ added to PBS. Neural stimulation success of PEMPs was evaluated on each day of measurement under 100 mW.cm⁻², 2 MHz FUS with 50 Hz burst stimulation frequency ($n = 12$). All data are presented as mean ± s.d. Statistical significance is determined by two-sided Student's t-test and *$p < 0.05$ was considered statistically significant. ns, not significant.

hypothesized that the piezo-electrochemical effects generated by a single PEMP could induce membrane depolarization. To investigate this hypothesis, we applied continuous LIFU excitation cycles for 100 ms at intensities between 5–100 mW.cm⁻², while the BTNP-conjugated half surface of the PEMPs always faced the cultured neurons to eliminate the orientation-dependent effects.

Our results revealed that the threshold FUS intensity required for repeatable neural stimulation was reached at ~9.6 mW.cm⁻² for the PEMPs (Fig. 2b). We observed effective membrane depolarization for action potential generation, accompanied by proper repolarization and hyperpolarization phases that restored the resting membrane potential (Supplementary Fig. 6). As the FUS intensity was increased from 10 to 50 mW.cm⁻², latency for generating neural activation decreased from ~24.2 to ~10.6 ms (Fig. 2c). We can conclude that higher FUS intensity leads to increased charge generation during the charging phase and accelerates the build-up of electric potential near the neural membrane[35] (Supplementary Note 1, Supplementary Fig. 7).

Similarly, increased FUS intensity provided a better success rate for neural stimulation (defined as the ratio of number of action potential responses over burst stimulation cycles), while the success rate was ~89% for 50 mW.cm⁻² FUS and saturated in higher intensities (Fig. 2d).

After setting an intensity threshold for neural activation, we proceeded to determine the temporal benchmark values for high-success rate stimulation. We found out that a single PEMP required ~4.4 ms of excitation period under 100 mW.cm⁻² FUS intensity to provide sufficient charge for threshold depolarization (Fig. 2e). Notably, this low excitation period threshold provides information about the maximum achievable stimulation frequency, indicating the limit of ~250 Hz considering the excitation and refractory periods. To find out the stimulation capability at the therapeutic frequency band (100–200 Hz)[9,26], we modulated the FUS driving signal (Supplementary Fig. 8) for low-to-high burst frequencies of 0.5–200 Hz. This frequency-dependent experiment indicated that PEMPs could stimulate primary neurons up to ~200 Hz with a moderately high success rate, particularly ~69%

and ~59% for 100 and 200 Hz, respectively (Fig. 2f). The reduced success rate at 200 Hz burst frequency could be attributed to the reduced charging time of PEMPs and consequently the lower potential build-up in each charging cycle. Once the neural stimulation benchmark values have been established for FUS intensity and excitation period, we constructed the excitability curve of PEMPs for the primary neurons. We found that as the LIFU excitation intensity increased, the threshold excitation duration decreased (Fig. 2g), which would be expected for charge-generating biointerfaces[32].

The piezoelectric response of the BTNP-coated magnetic particles under LIFU-generated pressure waves would be charge-balanced biphasic electrical signals, innately, and expected to generate safe piezoelectric charging. On the other hand, it is still essential to evaluate the cell viability and bioelectrical stability of the stimulated neurons. For the latter, we monitored the resting membrane potential of primary neurons in two particular conditions: first, only under FUS at our highest intensity of interest (200 mW.cm⁻²), and next, during the presence of PEMPs. We found no significant difference in resting membrane potential after 20 min of continuous 2 MHz FUS excitation with and without PEMPs (Fig. 2h). This result under continuous excitation suggests high bioelectrical stability under long stimulation cycles. Moreover, if we consider the therapeutic applications utilizing <200 Hz continuous excitation, we expect even safer bioelectrical stability, as less energy is delivered during burst pulses reducing the 2 MHz FUS driving signal down to <200 Hz pulses in comparison with the 2 MHz stimulation. To support this claim, the cell viability was assessed using the live/dead assay in response to changing FUS burst frequency up to 200 Hz, which showed no significant change under 0.5-to-200 Hz stimulation for 20 min (Fig. 2i).

As the recent studies utilized the piezocatalytic effect of BTNPs and reactive oxygen species (ROS) generation under high-intensity ultrasound excitation[36], it is crucial to evaluate ROS, which could potentially limit the long-term use of PEMPs due to oxidative stress on primary neurons and particle degradation. Therefore, we quantified the ROS exposure on the primary neurons using a fluorescent-based intracellular ROS measurement method, DCFDA assay. We utilized cultured neurons on a bare cover glass as the negative control, 100 μM $H_2O_2$ treated neurons as the positive control, and neurons excited with 2 MHz continuous FUS for 20 min as the experimental groups. While we observed high fluorescence intensity in positive control due to $H_2O_2$ intake, there was no statistically significant difference between negative control, only PEMP-treated neurons, and FUS-exposed neurons in addition to PEMPs (Fig. 2j and Supplementary Fig. 9). We attributed these results to the use of low-intensity FUS and to almost purely electrical charge generation, which did not generate significant thermal or pH changes in the extracellular medium (Supplementary Fig. 10).

For all neural stimulation experiments, we operate in the diagnostic, nonthermal, noncavitational (<100 mW.cm⁻²) spatial peak temporal average intensity levels[37–39]. Moreover, the excitation parameters are within the low-intensity US regime and lower than the observed and proposed intensities in the literature[40] for thermal, cavitation, microtubule resonance, and mechanosensitive stimulation mechanisms in terms of the temporal and spatial average intensity of FUS. To further evaluate these potential thermal and mechanical effects due to FUS excitation, we started our investigations by evaluating possible thermal effects due to FUS. Firstly, we modeled our FUS transducer in COMSOL Multiphysics by the recorded pressure waves using a hydrophone. To realize this, we modeled a water and tissue domain in COMSOL (Supplementary Fig. 11a). By using the pressure acoustics (Supplementary Fig. 11b, c) and bioheat transfer modules, the heating and cooling of the tissue phantom were calculated via Penne's bioheat equation. Thermal simulations were performed in a two-fold process corresponding to a worst-case scenario, propagation in a water medium, and thermal absorption in a brain-mimicking

medium. The following parameters[41] were followed for the propagation medium (water): sound speed, c = 1500 m s⁻¹; volumetric mass, ρ = 1000 kg m⁻³; nonlinearity coefficient, B/A = 5; attenuation coefficient, α = 2.2 × 10⁻³ dB cm⁻¹ MHz⁻ʸ; frequency power law of the attenuation coefficient, y = 2. COMSOL simulations were calibrated by adjusting the input pressure to match the pressure at the focus measured in the water tank by the hydrophone (Supplementary Fig. 11d–f). In the second part of the simulation, we utilized the heat transfer module in the tissue domain with the parameters: brain volumetric mass $\rho_{brain}$ = 1046 kg m⁻³, the brain sound speed $c_{brain}$ = 154 s⁻¹, $K_t$ is the brain thermal conductivity (0.51 W m⁻¹ °C⁻¹) with the initial brain temperature $T_0$ = 37 °C[41]. Once we have the heat generation due to acoustic wave absorption in the tissue domain, we can simulate the transient heating/cooling cycles due to 0.5–200 Hz modulated 2 MHz FUS excitation. First, we utilized single US pulses with 10–1000 ms of 2 MHz pressure waves (Supplementary Fig. 12a). We investigated the single pulse heating (Supplementary Fig. 12b) and utilized this information to simulate 0.5–50 Hz stimulation for 10 s (Supplementary Fig. 12c). The results indicate that increasing the stimulation frequency decreases the overall heating and even for the lowest frequency we did not expect >0.009 °C heating during 10 sec stimulation. To verify our simulations, we carried out local temperature measurements[42,43] by patch pipette resistance changes. First, we obtain the resistance-temperature calibration (Supplementary Fig. 12d). Then, we positioned the same patch pipette to the focal point of the FUS at the bottom of the recording chamber. The same pulse and frequency-dependent analysis was carried out and compared with the simulation results (Supplementary Fig. 12e, f). Simulations were well-matched with the experimental temperature measurements and indicate that the FUS intensity used in this study does not generate significant local heating and is much lower than the reported temperature changes for thermal activation of neurons[13,24,42,44].

We calculated the intensity characteristics of FUS stimulus based on the standards developed by the National Electronics Manufacturers Association[45], The American Institute of Ultrasound Medicine, and the United States Food and Drug Administration (FDA), *Marketing Clearance of Diagnostic Ultrasound Systems and Transducers*. By utilizing the measurements recorded from the calibrated hydrophone, a FEM model was built to simulate mechanical and thermal effects due to FUS. The pulse intensity integral (PII), the spatial-peak pulse-average intensity ($I_{SPPA}$), the spatial-peak, temporal-average intensity ($I_{SPTA}$), and the mechanical index were calculated as $PII = \int \frac{p^2(t)}{Z_0} dt$, where $p$ is the instantaneous peak pressure, $Z_0$ is the characteristic acoustic impedance in Pa s/m defined as ρc where ρ is the density of the medium, and c is the speed of sound in the medium. For the safety considerations, we used the brain tissue as the propagation and focus medium, where the brain volumetric mass $\rho_{brain}$ = 1046 kg.m⁻³, and the brain sound speed $c_{brain}$ = 154 m.s⁻¹. $I_{SPPA} = \frac{PII}{PD}$ where PD is the pulse duration, and $I_{SPTA} = PII*PRF$ where PRF is equal to the pulse repetition frequency in Hz. The mechanical index was defined as $MI = \frac{p_r}{\sqrt{f}}$. Our calculations indicate spatial-peak temporal-average intensities ($I_{SPTA}$) of 3.8–28.7 mW.cm⁻² for a total stimulus duration ranging between 2.5 and 500 ms. FUS waveforms had peak rarefactional pressures ($p_r$) of 0.014–0.108 MPa, pulse intensity integrals (PII) of 0.017–0.095 mJ.cm⁻², and spatial-peak pulse-average intensities ($I_{SPPA}$) of 0.027–0.267 W.cm⁻². The mechanical index (MI) was calculated as 0.028–0.0864 for the lowest and highest rarefactional pressures. These results indicate that FUS intensity values used in this study are not sufficient to evoke significant thermal or mechanical effects that might lead to stimulation of primary hippocampal neurons, especially >10 Hz neural stimulation[40,46–49]. In addition, we calculated the momentarily stress on the cells induced by the movement of the PEMPs on the cellular layer[50] (Supplementary Note 2). The calculations

revealed that the forces due to propulsion of the PEMP and gravitation were <50 pN, and corresponding pressure of <50 Pa with the assumption of 1 μm$^2$ contact area between the PEMP and neurons. The resulting force and pressure values are an order of magnitude smaller than the thresholds found in the literature[51]. Thus, the safety of PEMPs under FUS excitation has been proven by electrical activity, cell viability, and ROS intake tests, as well as by temperature measurements and mechanical calculations. To further evaluate the potential long-term use of our particles, we evaluated the neural stimulation success of PEMPs for 200 days in vitro. The particles were stored either in PBS or in 30 mM $H_2O_2$ added PBS, which accounts for the potential inflammatory response[52], and periodically tested for their neural stimulation performance on cultured primary neurons. These results indicated that PEMPs show high neural stimulation success over 200 days, >60% (Fig. 2k), even considering the effect of 30 mM $H_2O_2$. Therefore, the neural stimulation experiments with high stimulation success at the therapeutic band (50–200 Hz) together with the cell viability and functional stability tests indicate that the PEMPs have the potential for safe and long-term neural stimulation applications.

## PEMPs induce piezoelectric activation of voltage-gated ion channels under low-intensity FUS excitation

Having investigated the neural stimulation performance of PEMPs via patch-clamp electrophysiology experiments, we carried out a comprehensive channel blocker study to understand the mechanism behind the stimulation of primary neurons via PEMPs under FUS. Firstly, we identified the spontaneous baseline responses in neurons that have intracellular calcium shifts (Supplementary Fig. 13). Then, we started the calcium imaging experiments without any channel blocker and compared the area under the curve of spontaneous activity of primary neurons and their activity under FUS, FUS+magnetic Janus particles (Ni/Au coating on bare 20 μm silica microparticles, Supplementary Fig. 14), and FUS+PEMPs (Supplementary Fig. 15a) conditions. We found no significant difference between the spontaneous activity and under FUS, while the area under the curve significantly increased via PEMP-induced effects (Fig. 3a). Secondly, we evaluated the neural response via the application of synaptic blockers (CNQX and AP5), which blocks postsynaptic excitatory receptors, AMPA and NMDA, respectively[53], to understand whether the neural response is autonomous or requires synaptic connections with other neurons[16]. While a reduction in the area under the curve was observed for control samples without PEMPs, we did not observe a statistically significant difference for neural stimulation induced by PEMPs (Supplementary Fig. 15b). This shows that PEMP-based stimulation does not effectively require synaptic transmission between neurons. We then started treating neurons with different channel blockers while the dose of tetrodotoxin (TTX), cadmium ($Cd^{+2}$), gadolinium (III) ($Gd^{+3}$), and ruthenium red (RR) was carefully chosen to avoid blocking other channels or altering cell excitability (Supplementary Fig. 16c–f). We blocked the voltage-gated sodium channels with TTX and observed reduced calcium intensity during activation for both the control groups and the PEMP group. However, the reduce in the magnitude was significant for the PEMP group, indicating the role of voltage-gated sodium channels during neural activation (Fig. 3b). Then, neural responses were recorded with the addition of $Cd^{+2}$ (100, 200, and 400 μM), which prevents the activation of voltage-gated calcium channels[48,54,55]. We observed a significant reduction in the magnitude for both control and PEMP groups for 400 μM $Cd^{+2}$ concentration, for 100 μM $Cd^{+2}$ the reduction was significant for only the PEMP group (Supplementary Fig. 15d and Fig. 3c). The residual calcium transients not blocked by TTX and $Cd^{+2}$, 24% and 19% of the peak magnitude (Supplementary Fig. 15c, d), respectively, could be attributed to the noncomplete block of the voltage-gated sodium and calcium channel population[56], to other calcium sources in primary hippocampal neurons or involvement of other channels, which we investigated next.

While the control experiments in patch-clamp recording and calcium imaging under FUS without PEMPs indicated that, FUS itself did not induce any neural stimulation, we utilized gadolinium (III) to modify the deformability of the lipid bilayer and non-specific inhibition of mechanosensitive channels[16,57,58]. We observed a small <15% decrease in the PEMP group which indicates that the mechanosensitive channel population is not dominant or active in neural stimulation via PEMPs under FUS (Fig. 3d and Supplementary Fig. 15e). Since gadolinium (III) is not specific and changes the overall mechanical properties of neural membranes, Ruthenium Red was applied as a pore blocker of TRPV1, TRPV2, and TRPV4 channels[16,56,59]. We did not observe a significant decrease in the calcium intensity or excitability for the PEMP-induced neuron group (Supplementary Fig. 16f and Fig. 3e). As a result, considering the TTX and $Cd^{+2}$ blocker experiments, these observations indicate that PEMPs under FUS increase the $Na^+$ conductance in primary neurons and trigger $Ca^{+2}$ transients dominantly via voltage-gated sodium and calcium channels. The potential effect of synaptic transmission and mechanical activation were not involved or did not significantly contribute to the neural stimulation. Therefore, the electrochemical characterization of a single PEMP response under FUS, experimental characterization of heating and calculation of mechanical effects, and channel blocker-dependent experiments suggest that the neural stimulation was induced via the opening of voltage-gated sodium and calcium channels.

## Finite element simulations for spatial neural stimulation characteristics as a function of the particle size and orientation

Comprehending the spatial distribution of the generated electric field is essential for optimizing neural stimulation techniques, minimizing side effects, and enabling targeted modulation of specific neural populations. To understand the spatial distribution of the electric potential across the extracellular medium, we utilized finite element modeling (FEM) simulations. As the BTNP-conjugated microparticle surface acts as an electrical terminal under LIFU excitation, by predefined extracellular conductivity and permeability, we constructed the FEM simulation for different orientations and particle sizes of a single PEMP, i.e., top, bottom, and side orientation named for the orientation of the magnetization direction (Supplementary Fig. 16a).

The FEM simulations revealed that the generated electric potential profile highly depends on the PEMP orientation and is confined within proximity of PEMP (Supplementary Fig. 16b). Moreover, this confinement could be further tuned with the PEMP size (Supplementary Fig. 16c), which enables the rational design of PEMPs depending on the target type and space. For instance, applications demanding highly confined electric fields (e.g., for single-cell stimulation near PEMP) require smaller diameter particles. Furthermore, we can extend the simulations using multiple PEMPs. For instance, if there are two PEMPs in the target area (Supplementary Fig. 16d), depending on the distance between the PEMPs, generated electric fields could behave separately or these fields could temporally and spatially interfere (Supplementary Fig. 16e, f). Therefore, our simulations show that, depending on the targeted application, PEMPs could offer (i) broad or spatially-confined electric fields, (ii) spatial stimulation pattern design using their orientation-dependent electric field generation, and (iii) multiple PEMPs could generate various spatially- and temporally-interfered electric field patterns, which could be further investigated for temporal interference (TI) concept[60,61]. Before the population stimulation analysis, we utilized these results to interpret the orientation-dependent stimulation performance and to identify spatiotemporal neural stimulation characteristics of a single PEMP.

## Assessing population response and spatial neural stimulation characteristics

To better understand the population response and spatial neural stimulation performance of a single PEMP, and better interpret the

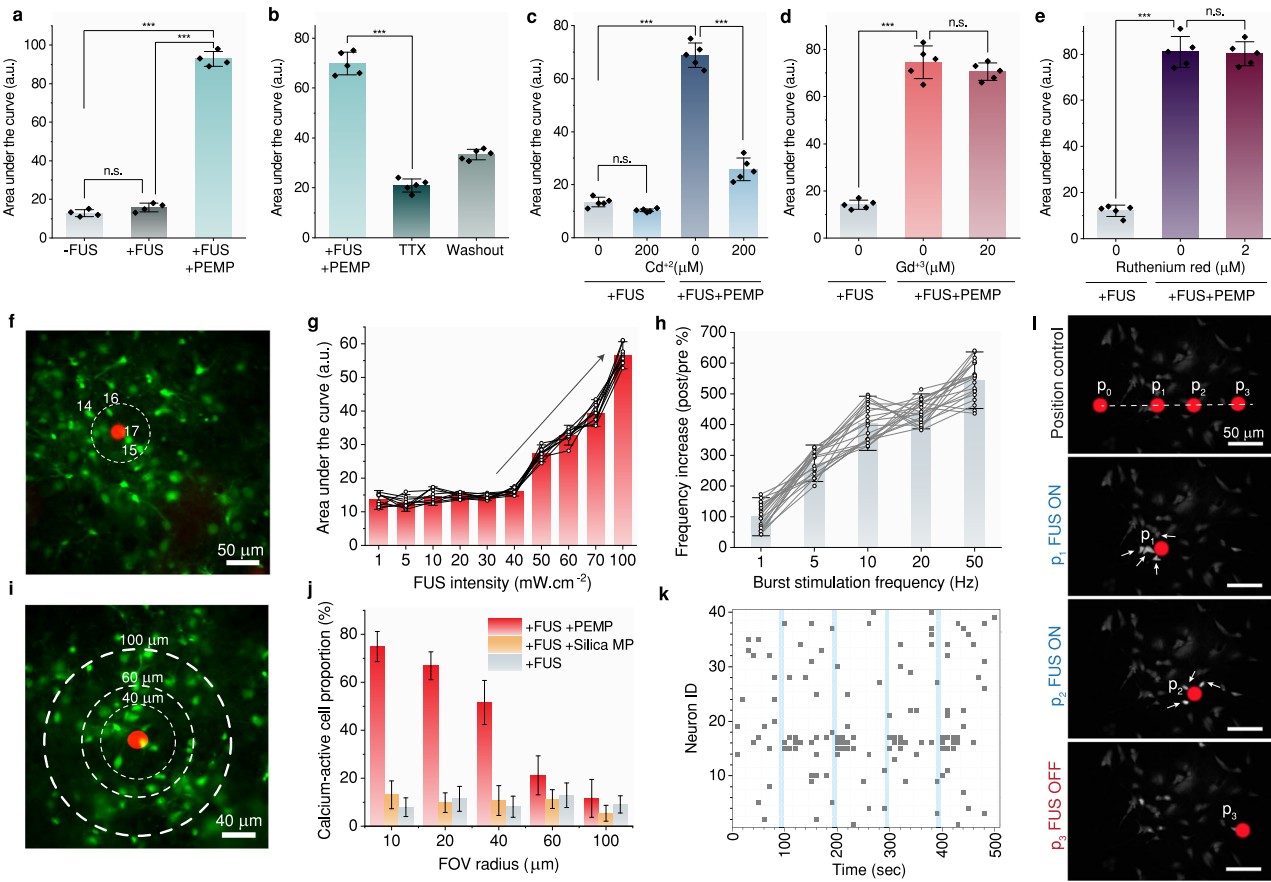

**Fig. 3 | PEMP-induced neural stimulation under LIFU is mediated by the voltage-gated sodium and calcium channels and confined within the PEMP proximity. a** Comparison of area under the curve (a.u.c.) of calcium response of spontaneous activity and neural activity during FUS excitation with and without PEMPs (*n* = 4 independent experiments, two-sided t-test, *p* = 0.0008). **b** Neural activity changes before and after tetrodotoxin (TTX) treatment (two-sided t-test, *p* = 0.0006). Quantification of a.u.c. before and after **c** cadmium (for 0 μM Cd$^{+2}$ *p* = 0.0005, 0 μM vs 200 μM Cd$^{+2}$ *p* = 0.0007), **d** gadolinium (III) (for 0 μM Gd$^{+3}$ *p* = 0.0009), and **e** ruthenium red (RR) treatment (for 0 μM RR *p* = 0.0003). For all experiments in (**b**), (**c**), (**d**), and (**e**), *n* = 5 independent experiments for control and experimental groups. Bar graph values in (**a**), (**b**), (**c**), (**d**), and (**e**) are represented as mean ± s.d. Two-tailed, unpaired, and paired T-tests were utilized for statistical analysis (\**p* < =0.05, \*\**p* < =0.01, \*\*\**p* < =0.001). **f** Representative image of a calcium imaging experiment with stained neurons and PEMPs. The orientation of the BTNP conjugated face of the PEMPs is directed towards the neurons. (neurons: green; PEMP: red). Scale bar, 50 μm. **g** Quantification of the a.u.c. under FUS pulses with different intensities (*n* = 20, 100 ms pulses, 10 Hz burst frequency). Data are presented as mean ± s.d. **h** Quantification of frequency increase under FUS with changing frequencies (*n* = 20, 50 mW.cm$^{-2}$). The increase is calculated by the ratio of post-stimulus and pre-stimulus neural activity. Each dot in (**g**) and (**h**) represents a single cell and its response to changing FUS parameters. Data are presented as mean ± s.d. **i** Image of individual neurons and single PEMP under 50 mW.cm$^{-2}$ FUS with 10 Hz burst frequency. Dashed lines represent the region of interest for the quantification of calcium signal intensity of neurons. **j** The percentage of cells with signal activity in the dashed area for three different conditions. The error bar represents the quantification for particles in different field-of-views (FOVs) (mean ± s.d. for *n* = 12 different FOVs, in each FOV *n* = 20 cells were analyzed). Bar graph values are represented as mean ± s.d. Scale bar, 40 μm. **k** Representative calcium activity of single neurons in the experimental condition of (**a**). Gray dots represent a neural activity exceeding the 5σ threshold. Blue lines represent the FUS excitation periods. **l** Representative images of Fluo-4 stained cultured primary neurons with a single PEMP experimental condition. The top image represents the overlaid images of a single PEMP at different positions. The middle and bottom images represent the calcium activity of neurons with PEMP at different positions, whereas white arrows represent calcium-active neurons with exceeded thresholds. Similar locomotion and neural stimulation performance was observed between the PEMPs. Scale bar, 50 μm.

FEM simulation results, we performed calcium imaging experiments on cultured primary neurons (Fig. 3f). We first analyzed the calcium signal amplitude as a function of the increased FUS intensity (Fig. 3g). The neurons exhibited reproducible responses with monotonically increasing amplitudes for >40 mW.cm$^{-2}$ FUS intensities. Moreover, in comparison with the neurons' spontaneous spiking activity, burst FUS pulses of 50 mW.cm$^{-2}$ increased the spiking frequency up to 5 times (Fig. 3h), indicating that neuron activity followed the burst stimulation pulses. However, analyzing the effects of higher-frequency burst pulses (>50 Hz) is not sufficient due to the slow fluorescent response kinetics of the chemical calcium indicators, which limits the detection of high-frequency spiking events[62].

After evaluating the neural stimulation performance in terms of transient calcium kinetics, we moved on to the spatial evaluation of the stimulation performance of a single PEMP, for the condition where the BTNP-conjugated surface was oriented towards the neurons. We draw circles in the field-of-view (FOV) with various radii centered at the PEMP to quantify the stimulation performance over the distance (Fig. 3i). To uncouple the effect of FUS, PEMP, and MR, we tested all of these conditions to evaluate the stimulation performance. From the center of the PEMP surface, a single PEMP could stimulate neurons >60%, up to 40 μm distance, while the calcium active cell proportion significantly decreased for further away neurons (Fig. 3j).

In comparison with the FEM simulations (Supplementary Figs. 16 and 3j), calcium imaging experiments revealed a longer

stimulation radius of up to 40 µm and a quick decay for longer distances. We attributed this difference to the idealization in FEM simulations, which neglected the electric-field screening effect in the extracellular electrolyte, and more importantly, electrical coupling and neural connections that transfer stimulated signals between neurons. This spatial characterization indicates that stimulation is confined within the PEMP proximity, which holds great promise for stimulating specific target regions. Moreover, we did not observe any statistical difference between bare silica microparticles under FUS excitation and only FUS excitation, which proves that the stimulation was solely due to generated piezoelectric potential (Fig. 3j). The population analysis for all neurons in the FOV also proved that stimulation was highly effective on neuron ID 15, 16, and 17 (Fig. 3f), while FUS excitation did not generate a population response (Fig. 3k). After the spatial characterization of neural stimulation via single PEMP, we demonstrated the actuation and locomotion capability of PEMPs. The experimental condition in Fig. 3l represents a target FOV with a single PEMP. The particle was stationary in $p_0$ and actuated to roll towards positions $p_1$, $p_2$, and $p_3$, using a custom-made five-coiled three-axis electromagnetic coil system. For all of these four positions, electromagnetic actuation stopped for providing stationary conditions and then the FUS of 50 mW.cm$^{-2}$ was applied. We observed that PEMP movement and electromagnetic actuation did not generate neural stimulation (Fig. 3l.i, iv), while, upon FUS excitation, PEMP exhibited stimulation of neurons nearby (Fig. 3l.ii, iii).

## Spatial control of neural stimulation using PEMPs

To further evaluate the spatial resolution of individual PEMPs, we conducted several patch-clamp recordings by measuring the membrane potential of neurons with changing distances from a single PEMP (Fig. 4a). At first, we investigated the effective stimulation distance for a single PEMP, which was magnetically steered to the target cell area and aligned to have BTNP-conjugated surface facing down to the neuron layer (Supplementary Movies 1 and 2). The membrane potential traces from various neurons indicated that a single PEMP with its piezoelectric active surface facing the neurons had an effective stimulation area defined by a circle with a radius of 55 µm. Although, its piezoelectric potential may have generated depolarization in further away neurons (Fig. 4b), either the depolarization was under the neural activation threshold or the neural activation is not reproducible (with a success rate of <40%). Moreover, observed spontaneous depolarization or action potentials for neurons, which are not in the effective stimulation radius, might be attributed to electrically-coupled neurons as we also observed in the calcium imaging experiments (Fig. 3f). Consistent with the FEM simulation results, the distance between two PEMPs determined their electric field and resulting neural stimulation profiles. For instance, FEM simulations suggested that if the distance between two PEMPs was over 100 µm, the generated electric field was decoupled for two microparticles (Supplementary Fig. 16e, f). Similarly, PEMPs could stimulate a neuron if they are separated by ~90 µm, while it was not possible for ~135 µm separation (Fig. 4c).

As a demonstration of the spatial control of neural stimulation, we created three different spatial configurations, specifically, single PEMPs were located within the FOV, then their orientation was flipped via magnetic actuation (Fig. 4d, e, and f and Supplementary Movie 2). By obtaining the primary neuron recordings, we could compare the expected piezoelectric field in FEM simulations with the in vitro neural stimulation characteristics. For the top, bottom, and side alignment (Fig. 4d, e, and f, respectively), at least eight neurons were recorded to reduce experimental error due to interconnected neurons, and we repeated the same experiments for four PEMPs for each alignment. The patch-clamp recordings revealed that for the top magnetic alignment, the effective stimulation radius was ~60 µm, while the bottom alignment generated stimulation success of ~35% for neurons only in close vicinity of the PEMPs (Fig. 4e). On the other hand, for the side

alignment, the effective stimulation radius had a similar profile to the top alignment for BTNP conjugated particle surface, however, the electric field quickly decayed from the non-conjugated particle surface (Fig. 4f). Therefore, we can conclude two main findings: first, the spatial piezoelectric neural stimulation performance reflects similar profiles with the FEM simulations. Second, as the stimulation success drastically changes between the BTNP-conjugated and non-conjugated surface, PEMPs can be utilized for neural stimulation in confined areas, and their steering ability could provide on-demand spatial control on the generated electric field.

## Investigation of the biocompatibility of PEMPs on various cell types

Before demonstrating cell-specific attachment and targeted neural stimulation, we first evaluated the biocompatibility of PEMPs. As we propose our design as a tool for basic research as well as for future clinical neural stimulation applications, we evaluated biosafety both for the LIFU excitation and for the PEMPs. For the former, the LIVE/DEAD™ cell imaging kit was utilized to evaluate the direct effect of FUS excitation for 60 min (Fig. 5a), which we chose as the upper limit considering the experimental and chronic neural stimulation applications. This experiment revealed that 100 mW.cm$^{-2}$ FUS excitation for 60 min did not generate significant effects on the differentiated SH-SY5Y neural cells (Fig. 5b). To evaluate the in vitro biocompatibility of the PEMPs, we conducted mitochondrial activity and membrane integrity-based cell viability tests on the same neural cell line using 3-(4,5-dimethyl-2-thiazolyl)-2,5-diphenyl tetrazolium bromide (MTT) assay and lactate dehydrogenase (LDH) assay. While the MTT assay indicated that for the 10 particle/cell condition, the cell viability decreased (Fig. 5c), we did not observe a significant difference with the control in the LDH assay (Supplementary Fig. 17a). In addition, no significant difference was found in the LDH assay for 1 particle/cell condition up to 96 h (Fig. 5d) with and without FUS excitation (Supplementary Fig. 17b). This variation between the two different cytotoxicity tests could be because of the particle interference to the colorimetric measurements of MTT assay in the high concentrations[63].

Moreover, to critically evaluate the biocompatibility for biomedical applications, we conducted cytotoxicity and immune reactivity tests on astrocytes and microglia to evaluate the innate immune response for potential future clinical applications[64]. For this purpose, we investigated the morphology and the cytokine release from astrocytes and microglia after incubation with PEMPs. We neither measured any statistically significant change of proinflammatory cytokines in the cell culture medium in enzyme-linked immunosorbent assay (ELISA) measurements (Supplementary Fig. 18) nor reactive morphology of glial cells (Fig. 5e). When we analyzed the total number and length of the branching for each cell, we did not observe any significant difference between the frequency distributions of glial cells with and without the PEMPs (Fig. 5f and Supplementary Fig. 19). Thus, these results indicate that our PEMP design and LIFU excitation did not exhibit significant toxicity and immune reactive effects on the cells, providing their potential use in neuroscientific research and future biomedical applications.

## Neurotransmitter-specific neuron targeting

Cell-specific neural stimulation enables researchers to map and characterize the complex neural circuits underlying neurological disorders. Therefore, the ability to selectively target and stimulate disease-specific neurons holds great promise for the basic research and treatment of a range of neurological and psychiatric disorders[65]. To demonstrate the adaptability of PEMPs for cell-specific targeting, the PEMP surface was decorated with antibodies against the G-protein-regulated inward-rectifier potassium channel 2 (GIRK2), which are already expressed on the surface of dopaminergic neurons of substantia nigra and ventral tegmental area, the commonly affected areas

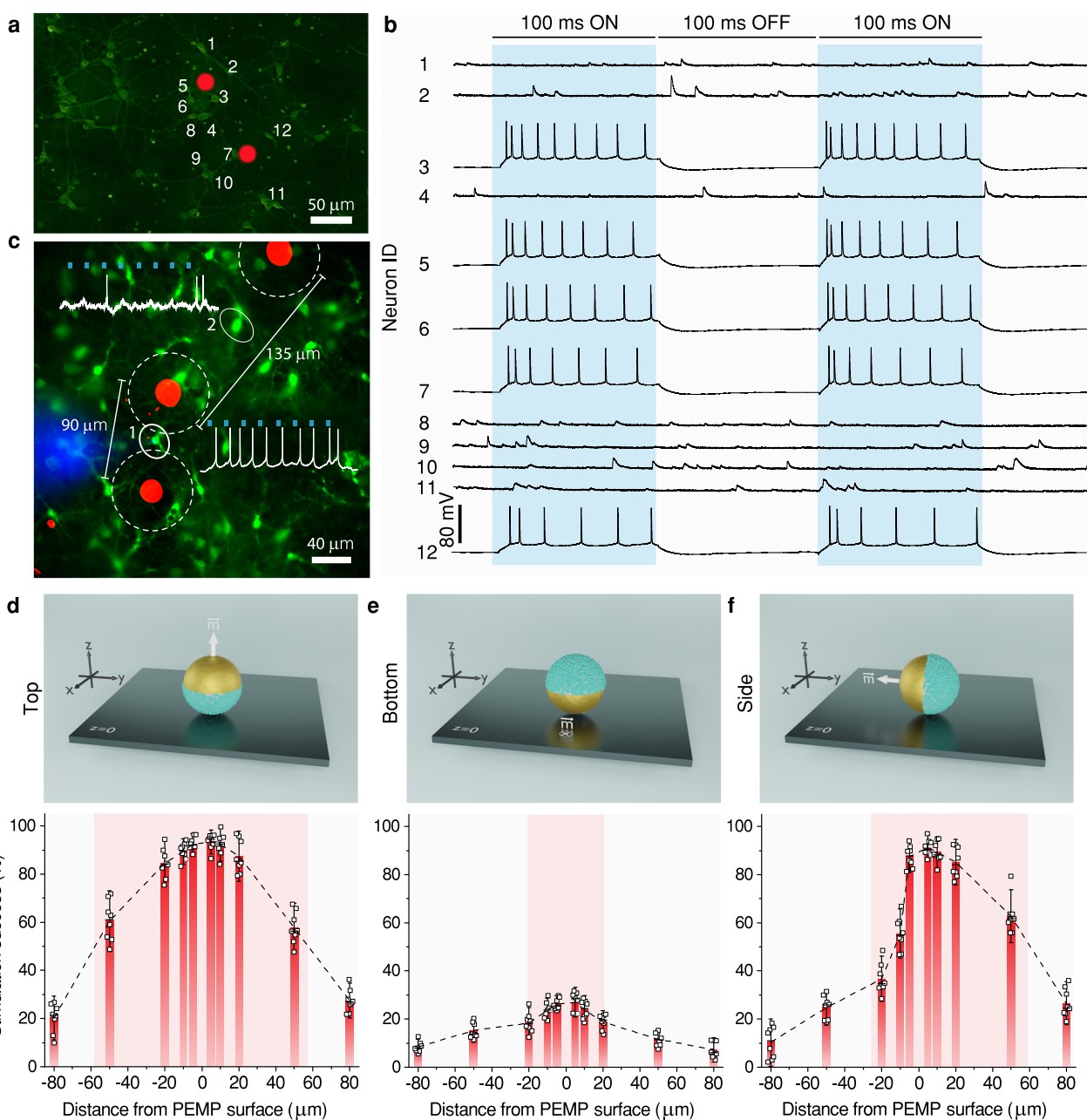

**Fig. 4 | Magnetic actuation-enabled orientational and spatial control over neural stimulation. a** Fluorescence image of primary hippocampal neurons and two piezoelectric magnetic microparticles (PEMPs) stained for spatial neural stimulation experiments. Numbers represent the recorded neurons with defined Neuron IDs. Scale bar, 50 µm. **b** Membrane potential traces for neurons are indicated in Fig. 4a. Each patch-clamped neuron is recorded for 5 min under 50 mW.cm⁻², 2 MHz FUS with 10 Hz burst frequency. The recording is repeated at least three times and a representative trace is plotted for each neuron ID. Blue areas represent focused ultrasound (FUS) excitation ON cycles. **c** Fluorescence image of primary hippocampal neurons in a field-of-view with three PEMPs with different distances. Dashed white lines represent effective stimulation areas for each PEMP. Representative membrane potential traces were presented for neurons 1 and 2 with stimulation ON times as blue lines on top. The patch-pipette is stained with Fluo-4

and pseudo-colored in the final image as blue. The orientation of the BTNP conjugated face of the PEMP is directed towards the neurons for all experiments in (**a**), (**b**), and (**c**). Similar neural stimulation profiles were observed for PEMPs separated by the similar distances for (**a**), (**b**), and (**c**). Scale bar, 40 µm. **d** Above, the orientation of a single PEMP is controlled by the out-of-plane rotating magnetic field of 10 mT, and neurons near the PEMP were recorded at various distances, where the magnetization direction for all conditions was indicated. The top, bottom, and side titles indicate the orientation of magnetization direction in (**d**), (**e**), and (**f**), respectively. Below, the neural stimulation success of a single PEMP for neurons at various distances was calculated ($n = 8$). The pink box in each plot represents the reproducible high-success stimulation area for each given condition. All data is represented as mean ± s.d.

from neurological diseases[66] (Fig. 6a). We utilized immuno-fluorescence staining to image the PEMP attachment to the dopaminergic neurons (Supplementary Movies 3 and 4). In the presence of PEMPs with functionalized targeting antibodies (GIRK2), microparticles were attached to the dopaminergic neurons even after several washing steps (Fig. 6b, c). When we compare the PEMPs with and

without GIRK2 antibody functionalization (Fig. 6d), the PEMPs with antibodies were bound to dopaminergic neurons (0.46 ± 0.06 particles per neuron), whereas PEMPs without antibodies were washed away completely (Fig. 6c). This PEMPs-based dopaminergic neuron-specific stimulation could be functional in neurological disease models as a non-invasive and non-genetic alternative of the optogenetic

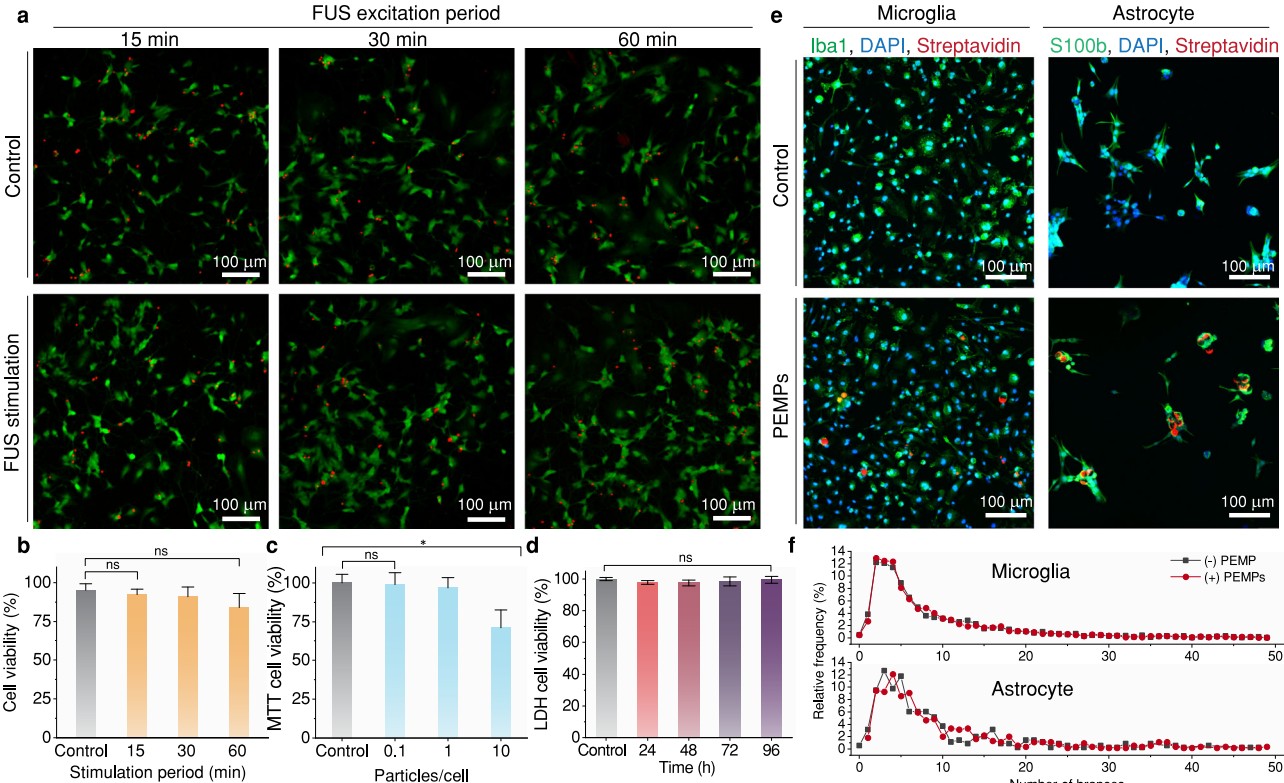

**Fig. 5 | Biocompatibility of the LIFU stimulation and PEMPs for various cell types. a** LIVE/DEAD™ cell imaging of differentiated SH-SY5Y cells under 100 mW.cm⁻², continuous FUS with a center frequency of 2 MHz and no FUS excitation for 15, 30, and 60 min. While the green color indicates live cells, the red color indicates dead cells. Scale bar, 100 µm. **b** Quantification of the cell viability at 15, 30, and 60 min (*n* = 20, from 4 independent experiments, two-sided *t*-test). Each cell viability (%) was calculated in comparison with their corresponding control groups. **c** The effect of the PEMPs on mitochondrial metabolic activity of the differentiated SH-SY5Y cells for 0, 0.1, 1, and 10 particles/cell condition assessed by the MTT assay (*n* = 12, from 3 independent experiments, two-sided *t*-test, control-10 particle/cell:

*p* = 0.0267). **d** The effect of PEMPs on differentiated SH-SY5Y cells under 1 particle/cell condition for 24, 48, 72, and 96 h (*n* = 12, from 3 independent experiments, two-sided *t*-test). **e** Immunofluorescence staining of microglia and astrocytes with cell-specific antibodies, Iba1 and S100b, respectively, after 72 h of incubation with and without PEMPs. Scale bar, 100 µm. **f** Quantification of the branch number per cell for the microglia (top) and astrocytes (bottom) with and without PEMPs. Bar graph values in (**b**), (**c**), and (**d**) are represented as mean ± s.d. Statistical significance is determined by a two-sided *t*-test and *p* < 0.05 was considered statistically significant. ns, not significant.

stimulation[67]. Owing to the modifiability of the streptavidin-biotin bonding, PEMPs can also be used for other types of neurons, such as glutamatergic or GABAergic neurons, to investigate the pathophysiology of various neuropsychiatric disorders[68]. In addition to its potential in basic research, the functionalized PEMPs with GIRK2 antibodies could create more precise and effective stimulation compared to non-targeted nanoparticle-based systems[10,69].

## Discussion

Cell-specific neural stimulation offers a new paradigm, enabling targeted manipulation of distinct cell types remotely and providing a more profound understanding of neural circuits underlying these disorders. Here, we showed that our approach of combining magnetic microparticles with piezoelectric nanoparticles offers three main advantages. First, by exciting the conjugated nanoparticles on a cell-sized microparticle surface, we confined the induced electric field, enabling selective stimulation of desired cell populations while reducing interference with neighboring neurons. Owing to this confinement effect, the PEMP design required lower US intensity thresholds (<100 mW.cm⁻²) for effective neural stimulation, unlike previous methods[24,28,70–72] that relied on the internalization and binding of nanoparticles[21,28], or high-intensity ultrasound[28]. Second, the asymmetry and magnetic steering ability of the PEMPs offer on-demand spatial control of the electric field generation in the extracellular space and precise neuromodulation without interference with off-target

cells. We showcased the orientation-dependent neural stimulation performance and effective stimulation radius on primary neurons, highlighting that swarms of PEMPs could be employed for 2D and 3D spatiotemporal engineering of stimulation patterns. Additionally, our antibody-antigen binding-based cell targeting approach on dopaminergic neurons demonstrated the versatility and future potential of the PEMP design for specific neurotransmitter-releasing neuron populations[68].

The PEMP design serves as a fundamental building block by integrating neural stimulation, cell-targeting, and imaging possibilities into a single microparticle system. Our present PEMP approach has potential applications in at least two distinct fields. First, it could be used as an alternative to optogenetics[73] in fundamental neuroscience research, especially using the emerging wireless recording systems[74–76]. While the locomotion and steering capability offers on-demand stimulation of targeted neurons without binding or damaging the cells, antibody-based cell targeting allows high-avidity binding for robust and cell-specific neural stimulation. Moreover, during the experiments detailed in the present study, we did not observe obvious side effects of the LIFU excitation and PEMPs on primary neurons, astrocytic, and microglial cells. Although the foreign body response varies for each microparticle-based implant, especially for the spherical-shaped implants, the foreign body response is reduced significantly[77]. For larger particles than 10 µm, the cellular clearance mechanisms, including phagocytosis, do not work[78], and they leave

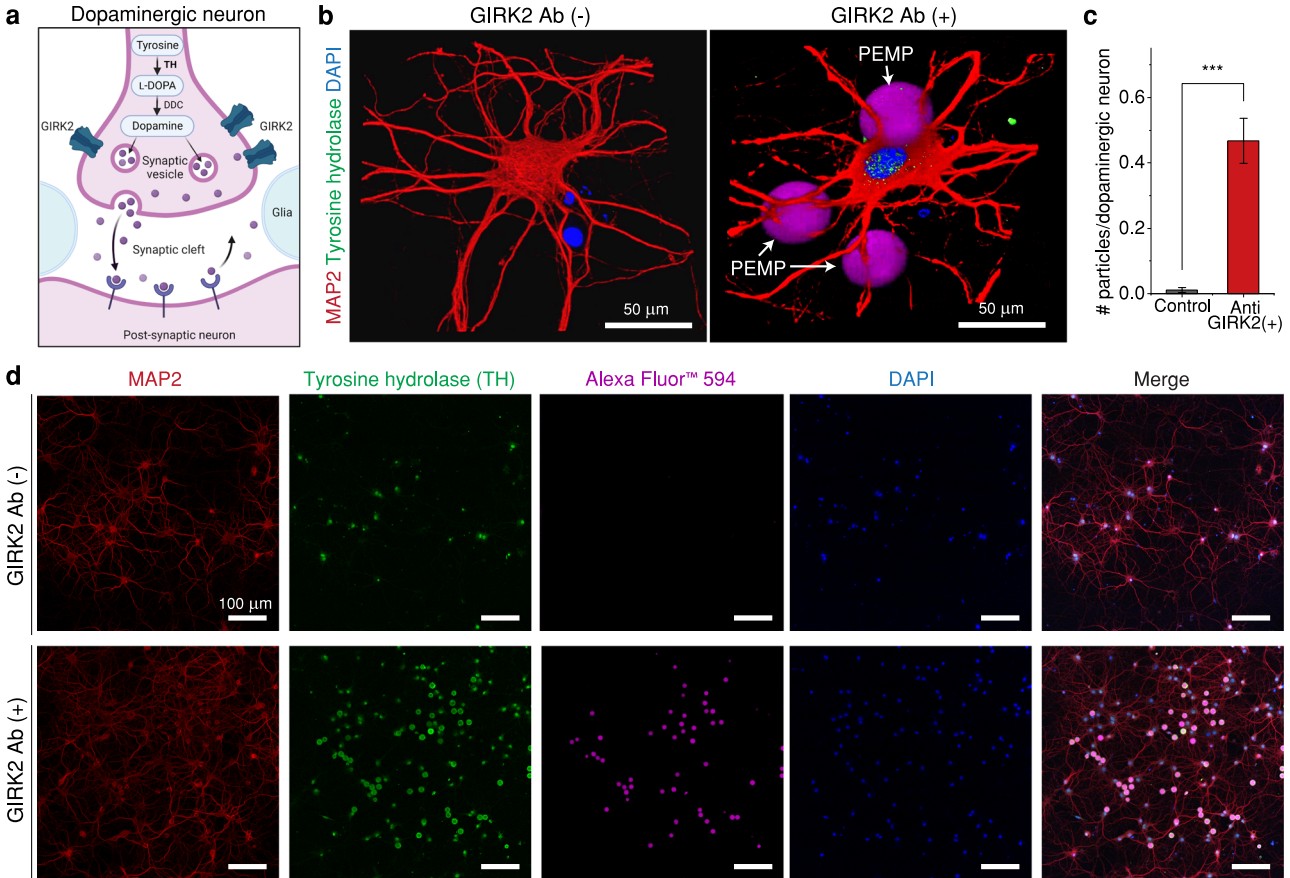

**Fig. 6 | Cell-specific targeting of the PEMPs for dopaminergic neurons.**
**a** Schematic of dopaminergic neuron junction with cell-specific antibody targets, such as GIRK2 and tyrosine hydroxylase. **b** 3D reconstruction (Supplementary Movies 3 and 4) of a single dopaminergic neuron and attached PEMPs with functionalized targeting antibodies (GIRK2). While the magenta color, Alexa Fluor™ 594, indicates PEMPs, the blue color, DAPI, indicates cell nuclei, the red color, MAP2, indicates neural cytoskeleton, and the green color, tyrosine hydroxylase inside

dopaminergic neurons. Scale bar, 50 µm. **c** Quantification of attached PEMPs per dopaminergic neuron, based on immunofluorescence images of primary neurons with and without Anti-GIRK2 antibodies (n = 24 independent field-of-views, two-sided t-test, p = 0.0006). Bar graph values are represented as mean ± s.d.
**d** Immunofluorescence images of primary neurons with the PEMPs functionalized with and without GIRK2 antibody decorations. Scale bar, 100 µm. All data is represented as mean ± s.d. Fig. Panel **a** is created with Biorender.com.

their place to tissue remodeling processes, including fibrosis[77]. While these processes can cover the implanted microparticles on the neural tissue, their effect on the material surface will be significantly lower than macrophage-based cellular clearance mechanisms[79]. Despite these advantages, and although it is fully covered by the Au thin film, the Ni metal film may not be ideal for long-term use in vivo due to potential material delamination and consequent toxicity effects. Alternatively, we would use more biocompatible magnetic films, for example, the L1$_0$ phase of FePt[80], which would provide higher biocompatibility for long-duration applications. Fine-tuning the magnetic and piezoelectric properties, along with exploring alternative materials, holds promise for improving the piezoelectric response and reducing ultrasound intensity thresholds, allowing even more precise control over neural stimulation.

A second field of possible application of the PEMP-based piezoelectric neural stimulation is that of therapeutics. For instance, early evidence suggests that deep brain stimulation (DBS) protects dopaminergic neurons by reducing dopaminergic neuronal cell death[81,82]. However, cell-specific neural stimulation is not possible with the conventional electrodes or requires transfection or genetic modification of the targeted cells[12,15,23,83]. Contrary to conventional electrodes, the PEMP-based targeting mechanism offers a non-genetic approach to improve current clinical treatment options as wireless miniaturized electrodes with specific cell targeting capability. Our design could be deployed by stereotactic injection into the potential target areas, such

as the subthalamic nucleus, hippocampus, and motor cortex for studying and targeting neurological diseases[84] or the ventral tegmental area (VTA) for targeting the reward circuitry[85] in behavioral neuroscience research. If smaller particles were needed for particular purposes, one potential limitation of PEMPs may be the tradeoff between the microparticle size and threshold FUS intensity due to the surface area-dependent conjugation amount of BTNPs. As the microparticle size decreases, the total amount of BTNPs on the particle half surface decreases, which reduces the total piezoelectric charge generation, thus higher FUS intensity is required for neural activation. Fortunately, there have been significant advancements in wireless power transfer systems[86,87] and in piezoelectric nanomaterials, which present the opportunity to enhance piezoelectric charge generation through the utilization of diverse nanoarchitectures for polymeric and synthetic ceramic materials[28,88]. Therefore, we envision that our method would pave the way for the development of particle-based wireless neural stimulation systems with improved efficacy and spatiotemporal control for both basic research and treatment of neurological disorders.

## Methods
### Fabrication and characterization of PEMPs
The fabrication of PEMPs consists of two main stages, namely magnetic Janus microparticle formation[27] and their conjugation with BTNPs. Magnetic Janus microparticles were fabricated by a benchtop

sputtering system (Leica EM ACE600, Leica Microsystem) by sequential deposition of 60-nm-thick Ni and 50-nm-thick Au films, respectively, on porous silica microparticles of 20 μm diameter and ~10 nm pore size. First, silica microparticles were spin-coated on a silicon wafer as a monolayer before sputtering. Second, a vibrating sample magnetometer (VSM) system (MicroSense) was utilized for out-of-plane magnetization of Janus microparticles under a 1.8 T uniform magnetic field. Finally, microparticles were released from the silicon wafer using a bath sonicator in ethanol and dispersed in phosphate-buffered saline (PBS). Then, the second stage provides the conjugation of BTNPs onto the Janus microparticle surface. To achieve this, we facilitate the biotin-streptavidin-biotin interaction between the half-silica surface and BTNPs. At first, the Janus particles were treated with 5% (v/v) (3-aminopropyl) triethoxysilane solution in ethanol and vortexed for 5 h, following the incubation at 65 °C for 3 h. The amino group grafted silica particles were transferred to dimethyl sulfoxide (DMSO) and reacted with N-hydroxysuccinimide–conjugated biotin (EZ-Link NHS-Biotin, Thermo Fisher Scientific, Waltham, MA; 5 mg/ml in DMSO) for 3 h. Then, the particles were treated with fluorescently labeled streptavidin (Alexa Fluor 594 conjugate, 100 μg/ml in 1× PBS, Thermo Fisher Scientific, Waltham, MA) for 1 h. As the Janus microparticles were ready for conjugation, BTNPs were grafted with the amino groups and biotinylated in the same procedure as the Janus microparticles. For the final stage, BTNPs and Janus microparticles were mixed in PBS for 3 h to complete the conjugation. For GIRK2 antibody surface functionalization, the particles were treated with biotinylated GIRK2 antibody (100 μg/ml in PBS, Thermo Fisher Scientific, Waltham, MA) for 1 h. Scanning electron microscopy (SEM) imaging and energy-dispersive X-ray spectroscopy (EDX) mapping of the PEMPs were performed by a Zeiss Ultra 550 Gemini SEM (Carl Zeiss Inc., Oberkochen, Germany). The hydrodynamic size and size distributions of $BaTiO_3$ nanoparticles were determined by Dynamic Light Scattering (DLS) (Möbius, Wyatt Technologies) with backscattering detection at a scattering angle of 163.5° and a wavelength of 532 nm. The solutions were diluted to 0.1 mg/mL in phosphate-buffered saline (1x) before the measurement. Each measurement was repeated three times and the average of the results was reported.

## Ultrasound-driven excitation system
For all patch-clamp recordings and calcium imaging experiments, we utilized a custom-built system, where the ultrasound excitation system and the electromagnetic coil system are integrated into an Olympus upright microscope equipped with a Prime BSI Scientific CMOS (sCMOS) camera and a pE-300 LED illumination system for fluorescence microscopy (Supplementary Fig. 3). As the focused ultrasound (FUS) source, we utilized a SU-101 FUS transducer (Sonic Concepts, WA, USA) with a 2 MHz center frequency and 1.28 mm focal width. We positioned the transducer inside a water tank on an optical table with vibration isolators. For the acoustic energy transfer without significant loss, we use polydimethylsiloxane (PDMS) film for the bottom of the patch-clamp recording chamber. The FUS transducer signal was generated using AFG 31000 series arbitrary function generator (Tektronix, OR, USA) and amplified by a piezo amplifier (Model 2100HF, Trek Inc.). The calibration of the transducer and measurement of the pressure profile was carried out using an HNA-0400 needle hydrophone (Onda Corporation, CA, USA). The position of the hydrophone was controlled by a computer-controlled linear translational stage (LTS300C, Thorlabs Inc., NJ, USA), and the voltage traces were recorded by a digital oscilloscope (Tektronix MDO3024, Tektronix Inc., OR, USA). The acoustic power and intensity were calculated using previously published standards[47]. We calculated the intensity characteristics of FUS stimulus based on the standards developed by the National Electronics Manufacturers Association[45], The American Institute of Ultrasound Medicine, and the United States Food and Drug Administration (FDA), *Marketing Clearance of Diagnostic Ultrasound Systems and*

*Transducers*. By utilizing the measurements recorded from the calibrated hydrophone (Supplementary Fig. 11d), the pulse intensity integral (PII), the spatial-peak pulse-average intensity ($I_{SPPA}$), the spatial-peak, temporal-average intensity ($I_{SPTA}$), and the mechanical index were calculated as $PII = \int \frac{p^2(t)}{Z_0} dt$ where $p$ is the instantaneous peak pressure, $Z_0$ is the characteristic acoustic impedance in Pa s/m defined as ρc where ρ is the density of the medium, and c is the speed of sound in the medium. For the safety considerations, we used the brain tissue as the propagation and focus medium, where the brain volumetric mass $\rho_{brain} = 1046$ kg.m$^{-3}$, and the brain sound speed $c_{brain} = 154$ m.s$^{-1}$. $I_{SPPA} = \frac{PII}{PD}$ where PD is the pulse duration, and $I_{SPTA} = PII*PRF$ where PRF is equal to the pulse repetition frequency in Hz. The mechanical index was defined as $MI = \frac{p_r}{\sqrt{f}}$.

## SH-SY5Y cell culture and differentiation
The human neuroblastoma SH-SY5Y cell line from DSMZ (German Collection of Microorganisms and Cell Cultures, DSMZ No.: ACC 209) was cultured in DMEM/F12 (Thermo Fisher Scientific, Gibco™ 11320033) medium supplemented with 10% heat-inactivated FBS (Thermo Fisher Scientific, Gibco™ 10500064) and 1% penicillin/streptomycin (Thermo Fisher Scientific, Gibco™ 15070063). The culture was maintained in a humidified incubator with temperature control at 37 °C and $CO_2$ control at 5%. SH-SY5Y culture was sub-cultured at the ratio of 1:10 once a week when it reached 75% confluence. Briefly, cells were washed with DPBS without calcium or magnesium (Thermo Fisher Scientific, Gibco™ 14190250) and then trypsinized with 0.05% Trypsin-EDTA (Thermo Fisher Scientific, Gibco™ 25300054). Trypsinization was ceased with four times the volume of the culture medium, and the cell suspension was centrifuged at $125 \times g$ for 5 min. The cell pellet was then re-suspended in a culture medium and cell concentration was counted using a hemocytometer (NanoEntek, DHC-N01). For biocompatibility tests in vitro, cells were plated in cell culture-treated 96-well plates at the density of 20,000 cells/cm². For population calcium imaging analysis, cells were plated at the same density on 10 μg/mL PDL (Sigma-Aldrich, P6407) coated coverslips. One day after plating, cell differentiation was induced with 10 μM retinoic acid (Sigma-Aldrich, R2625) in a differentiation medium consisting of Neurobasal™ Medium (Thermo Fisher Scientific, Gibco™ 21103049) supplemented with B27™ (Thermo Fisher Scientific, Gibco™ 17504044), 20 mM KCl, penicillin/streptomycin, 2 mM GlutaMAX™-I (Thermo Fisher Scientific, Gibco™ 35050061), 50 ng/mL BDNF (Sigma-Aldrich, B3795), 2 mM dibutyryl cyclic AMP (Sigma-Aldrich, D0627)[89]. Retinoic acid was added freshly just before the medium change. The differentiation medium was refreshed every two to three days and experiments were performed six to eight days after differentiation induction.

## Primary hippocampal neuron culture
Primary rat hippocampal neurons (Thermo Fisher Scientific, Gibco™, A1084101) were prepared according to the manufacturer's recommendations. Before plating for experiments, coverslips were coated with 10 μg/mL PDL and 5 μg/mL Laminin (Sigma-Aldrich, L2020) in PBS (Thermo Fisher Scientific, Gibco™ 10010023) for at least one hour in the incubator. Coverslips were then washed thoroughly with distilled water three times. Frozen cells from liquid nitrogen were rapidly thawed by gently swirling in a 37 °C water bath. The cryovial was washed once with 1 mL complete medium consisting of Neurobasal™ Plus Medium (Thermo Fisher Scientific, Gibco™, A3582901) supplemented with 0.5 mM GlutaMAX™-I and B27™ Plus (Thermo Fisher Scientific, Gibco™ A3582801). The neurons were suspended to a total volume of 4 mL complete medium. Viable cell density was determined with a hemocytometer. Cell suspension was diluted accordingly and cells were plated at a density of 50,000 cells/cm² on the pre-coated coverslips. The cells were then maintained in a humidified incubator at 37 °C temperature and 5% $CO_2$. Half of the medium was replaced with

fresh medium 24 h after incubation. From day 4 on, the medium was half refreshed every three days with the above-mentioned medium supplemented with an additional 25 μM L-Glutamate.

## Astrocytic and microglial cell culture

C8-D1A cells (CRL-2541, ATCC) were used as an astrocytic cell model and EOC13.31 cells (CRL-2468, ATCC) were used as a microglial cell model in the cell culture experiments. While Dulbecco's modified Eagle's medium (DMEM, Thermo Fisher Scientific, Gibco™, 11965092) with 10% heat-inactivated FBS was used for the C8-D1A astrocytic cells, a conditioned medium with LADMAC cells (CRL-2420, ATCC) were used for EOC13.31 microglial cells. Both cells were incubated in a humidified incubator at 37 °C temperature and 5% $CO_2$, until further experiments.

## Immunofluorescence staining, imaging, and image processing

The neurons were washed with PBS three times after 30 min of incubation with PEMPs under the FUS excitation period, whereas the glial cells, astrocytes, and microglia, were washed with PBS after 72 h of incubation with PEMPs in the humidified incubator. Then, all cells were fixed with 4% paraformaldehyde and washed three times in PBS with 0.1% Triton X-100 (PBS-T). Cells were blocked in SuperBlock solution (37515, Thermo Scientific) and they were incubated overnight with various primary antibodies (1:200 dilution in Superblock) mentioned below. After three times washing in PBS-T, the samples were incubated for 90 min at 37 °C with corresponding secondary antibodies (1:200 in PBS-T solution) for the multicolor immunofluorescence images. All samples were washed three times with PBS-T, then mounted with a DAPI-supplemented mounting medium (ab104139, Abcam), and the images were collected by using a Leica SP8 confocal fluorescence microscope. The primary antibodies, that were used in the staining; MAP2 (ab5392, Abcam) for neural cytoskeleton, tyrosine hydroxylase (ab137869, Abcam) as a dopaminergic neuron marker, S100b (ab52642, Abcam) as an astrocyte marker, Iba1 (ab178846, Abcam) as a microglia marker. In addition to Alexa Fluor 594-stained PEMPs, the secondary antibodies, that were used in the staining; were anti-chicken Alexa Fluor 555 (A-21437, Invitrogen), anti-rabbit Alexa Fluor 488(A-11008, Invitrogen), and anti-mouse Alexa Fluor 647 (A-21235, Invitrogen) with respective primary antibodies. The numbers of cells, particles, and cellular morphometric parameters were measured with custom codes in MATLAB software (R2021a, MathWorks), according to the protocols from previous literature[64,90]. All MATLAB codes, that were used for the image processing can be found here: https://github.com/erdosty/pemps.

## In vitro cytotoxicity analyses

To analyze the biocompatibility of PEMPs, an MTT cell proliferation kit (Roche, 11 465 007 001) and CyQuant™ LDH cytotoxicity assay kit (Thermo Fisher Scientific, Invitrogen C20300) were used to check the metabolic activity and the membrane integrity on SH-SY5Y cells, respectively. Different concentrations of PEMPs (0.1, 1, and 10 particles/cell) were applied to cells two days after plating. Assays were performed according to the user guide 24 h after the PEMP introduction. For the MTT test, the MTT labeling reagent was added, and the plate was kept in the incubator for 4 h. The solubilization buffer was then added, and the reaction was performed in the incubator overnight. Absorbance at 570 nm with reference at 660 nm was measured with a TECAN Infinite® M Plex microplate reader. For the LDH assay, 50 μl medium from each well was taken out into a new 96-well reading plate and mixed with 50 μl reaction mixture, incubated at room temperature for 30 min protected from light. 50 μl stop solution was then added in and the mixture was mixed by gentle tapping. Absorbance at 490 nm with reference at 680 nm was measured one hour later with the TECAN Infinite® M Plex microplate reader.

After the SH-SY5Y neurons were prepared similarly to the previous biocompatibility experiments, they were also incubated under 100 mW.cm$^{-2}$, continuous FUS with a center frequency of 2 MHz and no FUS excitation for 15, 30, and 60 min. After FUS application, the cell viability was measured using the LIVE/DEAD™ cell imaging kit (Thermo Fisher Scientific, Invitrogen, R37601). The numbers of live and dead cells were counted, similar to the method described for the immunofluorescence image analysis using inverted fluorescence microscopy (Axio Observer Z1, Zeiss). For the long-term cytotoxicity analysis, PEMPs were applied to cells one day after differentiation initiation and the cells were stimulated with focused ultrasound (FUS) four hours later for 10 min. LDH tests were performed 48 h, 72 h, and 96 h after stimulation with a differentiation medium change at 48 h time point.

## Enzyme-linked immunosorbent assays for cytokines

The supernatants of the astrocytic and microglial cells were collected after 48 and 72 h of incubation with and without PEMPs. The concentrations of the proinflammatory cytokines, IL-1β, IL-6, TNF-α, and G-CSF, were measured with commercially available kits (SimpleStep ELISA, Abcam) according to the manufacturer's recommended protocols. The cytokine levels for each experimental group were measured blindly in a triplicate manner.

## In vitro electrophysiology experiments

The patch-clamp electrophysiology experiments on cultured primary neurons were carried out by the patch-clamp amplifier (Axopatch 200B, Molecular Devices, CA, USA) connected to the low-noise data acquisition system (Axon Digidata 1550B, Molecular Devices, CA, USA). Patch pipettes were pulled by a P-2000 micropipette puller (Sutter Instrument, CA, USA) to obtain 10–12 MΩ resistance. On the day of the experiment, the primary neurons without any PEMPs were placed in the measurement chamber, which was filled with the fresh extracellular medium (Invitrogen™ Live Cell Imaging Solution (LCIS), Fisher Scientific + 15 mM D-glucose). We then added PEMPs in 0.1 particle/cell concentration to prevent potential mixed effects of many particles and to reduce interference between the PEMPs. The measurement pipettes were filled with the intracellular solution (Internal KF 110, Nanion, Munich, Germany) for whole-cell patch-clamp configuration. An Olympus microscope with a Prime BSI Scientific CMOS (sCMOS) digital camera was used to monitor the patching procedure. We looked for PEMPs individually standing in the field-of-view in order to understand the interaction between standalone single PEMPs and nearby neurons. The cells were patched prior to the FUS excitation and their health and excitability were monitored. In the orientation-dependent experiments, electromagnetic actuation was provided for reorienting the PEMPs for the desired configuration prior to the patching. For the experiments with FUS excitation, we modified the microscope system by removing the microscope condenser and switching to fluorescence imaging. To reduce vibrations due to FUS excitation, the bottom of the water tank was coated with acoustic isolators. The Ag/AgCl ground electrode was immersed in the extracellular medium as the measurement ground. The temperature of the measurement chamber was monitored during all patch-clamp experiments. The extracellular medium was refreshed in each 30 min by half medium change with LCIS.

## Piezoelectric current measurements

A very low concentration of PEMPs in live cell imaging solution was prepared to transfer single particles inside of pulled glass patch pipettes, which were mounted onto the patch-clamp system. The glass patch pipettes had a tip diameter of <2 μm. The measurement pipette was positioned on the bottom surface of the recording chamber to its position at in vitro experiments on primary neurons. The system

voltage is held at zero in voltage-clamp mode to generate the virtual ground in the system following the published procedure[32]. The FUS intensity of 10, 20, 50, and 100 mW.cm$^{-2}$ was applied to the recording chamber. As a control experiment, the glass recording pipette without any PEMP was recorded under the same FUS intensities. The same filtering settings used in the patch-clamp electrophysiology experiments mentioned in the "Methods" section.

## Intracellular oxidative stress measurements

The intracellular oxidative stress was measured by 2′,7′-dichlorodihydrofluorescein diacetate ($H_2DCFDA$) (D399, Molecular Probes, Invitrogen), which is an intracellular reactive oxygen species indicator[91]. Primary hippocampal neurons were seeded on cover glasses as mentioned above and neurons were excited with 2 MHz continuous FUS for 20 min after PEMPs were introduced. While one experimental group was stimulated, the same amount of PEMPs were added to the experimental group with only PEMPs and 100 μM $H_2O_2$ added to the positive control group. After the different treatments, all neurons were washed once with PBS and then incubated with 20 μM $H_2DCFDA$ in aCSF solution for 45 min in the humidified cell culture incubator before imaging to allow $H_2DCFDA$ to enter the neurons. After the incubation and washing with PBS in dark conditions, immunofluorescence images were taken from the samples in the same light intensity, exposure, time point, and magnification with a live-cell fluorescence microscope (DMi8, Leica), and average fluorescence intensity was measured with a custom code in MATLAB software (R2021a, MathWorks), that can be found in the same repository with previously mentioned codes. In short, the fluorescence images were converted to grayscale images, and the background noise was subtracted. To measure the number and area of cells, the maximum entropy threshold was applied to the image and the particles were analyzed for integrated density. In this way, both areas of neurons and the intensity of fluorescence were measured in the images from randomly selected ten different regions from each sample in three different samples for the experimental group. Relative fluorescence intensity was calculated with negative (neurons without any treatment) and positive (neurons with $H_2O_2$ treatment) controls. The ROS production levels in the negative control group were accepted as 100% in the relative fluorescence intensity comparison.

## Ca$^{+2}$ imaging experiments

Cells were loaded with 1 μM Fluo-4 AM dye (Thermo Fisher Scientific) in Live Cell Imaging Solution (LCIS; Invitrogen) for 30 min at 37 °C. After Fluo-4 loading, cells were washed three times with LCIS, and the PEMPs were loaded into the experimental medium. We utilized the same setup as the patch-clamp experiments for the Ca$^{+2}$ imaging experiments. Fluo-4 was excited using a 460-nm LED and time-lapse images were recorded at ×20 magnification, every 0.1 s and recorded using a Prime BSI Scientific CMOS (sCMOS) digital camera. Time-lapse recordings were analyzed using ImageJ software and EZcalcium toolbox[92]. In channel blocker experiments, chemical blockers were applied directly to the extracellular medium. For each channel blocker, the spontaneous calcium activity and cell excitability were monitored to determine the application concentration. To pharmacologically block voltage-gated sodium channels, tetrodotoxin (TTX) was applied in 0.5 μM. The postsynaptic blockers AP5 and CNQX both in 1 μM concentration were applied to test the synaptic transmission between primary neurons, which blocks postsynaptic excitatory receptors[53]. To block voltage-gated calcium channels, cadmium chloride was added to the extracellular medium to obtain Cd$^{+2}$ in 100, 200, and 400 μM concentrations. Finally, gadolinium (III) in 10, 20, and 30 μM concentrations and ruthenium red (RR) in 1, 2, and 5 μM concentrations were applied to nonspecifically block the mechanosensitive ion channels and to block TRP channels, respectively[16,93].

## Actively controlled locomotion on cell culture

The microrollers were actuated using a custom-made five-coiled electromagnetic coil system, which provides uniform rotating magnetic fields with an amplitude of 10 mT that enables surface rolling and steering control for all experiments.

## COMSOL simulations

COMSOL Multiphysics 5.5 simulation software (COMSOL, Inc.) was utilized to estimate the spatial distribution of the electric potential magnitude in the extracellular medium and over the cell membrane using the "Electric Currents" interface[94]. All simulations were performed in 3D geometry, by assuming a steady-state system with constant electrical potential generation and its distribution in extracellular fluid, particularly in aCSF. A single PEMP was represented by a 20-μm-diameter sphere with half the surface coated with Au and the other half as the potential generation terminal, which models the BTNP conjugated surface. For the spatial electric field simulations, the PEMP diameter swept between 4 and 50 μm, and the electric field intensity was normalized to its maximum value for each case. For the two PEMP cases, the particle diameter was kept at 20 μm and the distance between the particles was swept between 0 to 100 μm. The sphere was put in an electrically conductive medium with 1.25 S/m conductivity, and the simulation space was grounded on all sides.

## Heating simulations and temperature measurements

The FUS transducer was modeled in COMSOL Multiphysics® v6.1 software (COMSOL AB, Sweden) by the recorded pressure waves using a hydrophone (Supplementary Fig. 11a, b). To realize this, the water and tissue domains were modeled in COMSOL. The Pressure Acoustics and Bioheat transfer modules were utilized to investigate acoustic field pressure and to calculate the heating and cooling of the tissue phantom via Penne's bioheat equation, respectively. Thermal simulations were performed in a two-fold process corresponding to a worst-case scenario, propagation in a water medium, and thermal absorption in a brain-mimicking medium. The following parameters[41] were followed for the propagation medium (water): sound speed, c = 1500 m s$^{-1}$; volumetric mass, ρ = 1000 kg m$^{-3}$; nonlinearity coefficient, B/A = 5; attenuation coefficient, α = 2.2 × 10$^{-3}$ dB cm$^{-1}$ MHz$^{-y}$; frequency power law of the attenuation coefficient, y = 2. COMSOL simulations were calibrated by adjusting the input pressure to match the pressure at the focus measured in the water tank by the hydrophone. In the second part of the simulation, we utilized the heat transfer module in the tissue domain with the parameters: brain volumetric mass $\rho_{brain}$ = 1046 kg m$^{-3}$, the brain sound speed $c_{brain}$ = 154 s$^{-1}$, $K_t$ is the brain thermal conductivity (0.51 W m$^{-1}$°C$^{-1}$) with the initial brain temperature $T_0$ = 37 °C. Once the heat generation due to acoustic wave absorption in the tissue domain was calculated, the transient heating/cooling cycles were simulated for 0.5–200 Hz modulated 2 MHz FUS excitation. To verify the numerical calculations, local temperature measurements[42,43] were carried out by patch pipette resistance changes. First, the resistance-temperature calibration was obtained via recording of pipette resistance and the thermocouple while the bath temperature of the extracellular medium was cooled down to the room temperature from 45 °C. Then, the same patch pipette was positioned at the focal point of the FUS at the bottom of the recording chamber. The same pulse and frequency-dependent analysis was carried out and compared with the simulation results.

## Statistical analysis methods

All quantitative values are presented as mean ± s.d. Student's t-tests were performed for two group comparisons and two-way ANOVA tests were performed for multiple comparisons by using the softwares OriginPro 2021b (OriginLab) and GraphPad Prism 6 (Graph Pad Inc.) to assess the statistical significance of differences in the results. *$p$ < 0.05 was considered as statistically significant, and non-significant

differences are presented as "ns". In each figure $P$ values were represented as *$p < 0.05$, **$p < 0.01$, ***$p < 0.001$ and ****$p < 0.0001$. The box-plot elements are the following: center line, mean; box limits, 25th and 75th percentiles; whiskers represent the outliers (coefficient 1.5) of the distribution.

### Reporting summary

Further information on research design is available in the Nature Portfolio Reporting Summary linked to this article.

## Data availability

All data supporting the findings of this study are available within the article and its supplementary files. Any additional requests for information can be directed to, and will be fulfilled by, the corresponding authors. Source data are provided with this paper.

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

## Acknowledgements

The authors thank Muhammad Turab Ali Khan for the discussions and critical reading of the manuscript. We also thank Anitha Shiva for TEM and SEM-EDX imaging. This work was funded by the Max Planck Society and German Research Foundation (Deutsche Forschungsgemeinschaft; DFG) within the Priority Program 2311, grant number 465186293. E.Y. received funding from the European Union's Horizon 2020 research and innovation program under the Marie Skłodowska-Curie grant agreement number: 101059593.

## Author contributions

M.H., E.Y., U.B., and M.S. designed the study. M.H., E.Y., and U.B. performed the fabrication and characterization of PEMPs. M.H. performed neural stimulation experiments. E.Y., A.A., and Y.Y. performed the cell culture and biocompatibility experiments. A. B. performed hydrophone measurements and helped with the simulations. S.K. performed and analyzed DLS measurements. M.H., E.Y., U.B., A.A., S.K., and M.S. critically reviewed and edited the final version of the manuscript. All authors were involved in the data interpretation, discussed the results, and commented on the study. M.H. wrote the manuscript with input from all authors. M.S. initiated and supervised the study.

## Funding

## Competing interests

The authors (M.H., E.Y., U.B., M.S.) declare the following competing interests based on this study: a patent application to the European Patent Office with application number 24158087.7. The remaining authors (A.A., Y.Y., A.B., S.K.) declare no competing interests.
