## [Peer Review File · Nature Communications]

REVIEWER COMMENTS

Reviewer #1 (Remarks to the Author):

Using ultrasound to wirelessly stimulate nerves to treat diseases is a new means of treating neurological diseases. In this paper, silica-based piezoelectric magnetic Janus microparticles (PEMPs) were developed to enable clinically relevant high-frequency nerve stimulation of primary neurons under low-intensity focused ultrasound. This manuscript is well-written and provides very good application value. Therefore, I recommend publishing this manuscript after minor revisions. Here are the suggestions for the authors.

1. Janus nanoparticles are promising materials. As far as we know, the Janus nanoparticles used in this work should belong to inorganic Janus nanoparticles. In the reports of other biological applications, researchers often consider the toxicity and hydrophilicity of inorganic Janus nanoparticles. Your research has been compared with *in vitro* cell experiments, and it seems to be less toxic to cells, which is very gratifying. However, in the injection scenario as you described, should we also consider the subsequent metabolism of Janus nanoparticles? Do cell-sized Janus nanoparticles pose a potential threat to human health as microplastics?

2. Your transmission electron microscopy Figure 1e shows that the size of the Janus nanoparticles prepared by you is about 200 nm, and the silica particles used are about 20 microns (μm). How large is the particle size? Or is the scale of TEM mislabeled? (In addition, TEM does not look uniform in size, which will affect performance.)

3. Does the thickness of gold (20 nm) and nickel (60 nm) be considered (as cell-sized Janus particle size) or supported by previous work? Why choose such a thickness?

Reviewer #2 (Remarks to the Author):

The study focuses on the development of silica-based piezoelectric magnetic Janus microparticles (PEMPs) combined with low-intensity focused ultrasound (LIFU) to enable high frequency neural stimulation of primary neurons. The microparticle designed by the authors is innovative and the applicability is multifaced due to the possibility of antibody targeting functionalization. The manuscript includes the fabrication of the piezoelectric-magnetic microparticles, as well as multiple aspects of *in vitro* response of different neural cells treated with PEMP alone or PEMP in combination with LIFU.

The results reported in the manuscript are well described and the novelty of the work is sufficiently clear, the data interpretation is appropriately described. Despite these positive elements, some important aspects need to be deepened.

To strengthen the work and make it suitable for publication in Nature Communications I strongly suggest the authors to further investigate the following aspects related to the innovative system that they are willing to publish:

- Investigate the colloidal stability of PEMP in water and cell culture media.

The dimension of the final microparticles is reported by the authors to be equal to 20 μm according to TEM measurements, but do the authors have any data about their dimension and their dimension distribution in water or in cellular medium? For how long will the particle structure remain stable in time and in different media? Are there at any point in time some aggregation phenomena? Is it possible that the BaTiO₃ nanoparticles (BTNP) conjugated on one half of the microparticles begin to dissolve and release ions which are toxic for cells?

- Investigate the long-term toxicity of PEMP on neural cells.

The work proposed by the authors investigates cell viability after 24h from PEMP administration or ultrasound stimulation, but the effects on cell viability after 48, 72 and 96 h were not studied. This data are fundamental to effectively evaluate the biocompatibility of PEMP.

Furthermore, do the authors have any information, based on experimental knowledge or previous literature, about the clearance of their microparticles?

Minor editing of English language are required.

Reviewer #3 (Remarks to the Author):

Han et al. report piezoelectric magnetic Janus microparticles (PEMP) for neural stimulation. This is a well-organized, written and interesting study. Especially by looking to the literature using calcium imaging dyes for neurostimulation sensing, the use of patch-clamp electrophysiology, which is a difficult one, is a great contribution to the field that can help future studies with a more quantitative understanding of neurostimulation via piezoelectric nanoparticles.

One surface of the spherical silica microparticles are coated with BaTiO₃ nanoparticles, while the other surface is magnetically responsive. The manuscript provides a detailed explanation of the PEMP and

their capabilities. A wide variety of electrophysiology and cellular experiments were nicely done even by changing the orientation of the PEMP. I recommend the publication of the manuscript after minor revisions.

There are several points that can further improve the study:

- 1) The mechanisms underlying the neural stimulation process and the physical principles governing the behavior of PEMP need to be elucidated in more detail. This would enhance the understanding of the neuron and PEMP interaction.
- 2) The authors can report the voltage and/or current via patch-clamp electrophysiology setup. To enhance the effect, array of PEMP may be assembled on a glass substrate if required.
- 3) How is the procedure applied. Are cells cultured first and then PEMP injected? How many of them were injected? Write the experimental protocol on interfacing of PEMP with membrane step-by-step into the methods (for the patch-clamp experiments).
- 4) Patch-clamp FUS+ metallic side (without piezoelectric nanoparticles) is useful. There may be some local effects triggered by the metal and FUS interactions.
- 5) The authors apply a 20 ms pulse and show an exemplary AP firing in Fig S2, but there is already steady increase in the resting membrane potential before the stimulation. Can authors clarify why there is this increase (almost 7 mV in 60ms)?
- 6) Figure 2e shows that for 50 ms US activation the neuron still continues fires 2-3 times more after switching off the US-activation. This is an interesting result. What is the reason of this?
- 7) Does the PEMP movement damage the neuronal environment (cell soma or neurite) and does moving the particles effect the cells excitability?
- 8) In Fig 4b, authors show the patch clamp results depending on distance from the particles. Here some of the neurons experiences strong depolarization, while others do not because of the distance. If the neurons ids and distances shown in Fig 4a and b are correlated, why does 1, 2, 4, and 8 does not experience any depolarization, while number 12 is having strong firing success?

Other points:

- When explaining figure 2f, the success rate is below 70% and 60%, while it is written as 83% and 61% for 100 Hz and 200 Hz, respectively.
- In line 231, missing closing parentheses.
- Fig 4a, neuron 9 is not indicated.
- Page 9, a small typo: biocompativle

Reviewer #4 (Remarks to the Author):

The manuscript submitted by Sitti and coworkers describes the use of piezoelectric microparticles that can be targeted at the single-cell level and transduce focused ultrasound to stimulate action potentials excitable cells (eg hippocampal neurons) in vitro. The particles additionally have a magnetic coating which allows using an externally applied magnetic field to manipulate the position of the particles. In general I find the approach interesting and clever, however there are, in my estimation, major technical flaws in the experiments and the interpretation. In its present form, this paper falls into the category of one of many where 1) nano/microparticles added to in vitro cell culture 2) energy is applied to the system 3) cells somehow fire action potentials.

Technical problems:

1) The paper claims that the piezoactuated particles stimulate the cells electrically, via action upon voltage-gated channels and subsequent cellular membrane depolarization. The data do not support, and rather contradict this conclusion. There is no evidence for extracellular electrical stimulation, for the following reasons:

1.1 Electrical depolarization of the cell by an extracellular microelectrode results in a clear stimulation artefact in the current-clamp trace of the intracellular recording. The magnitude of this stimulation artefact is of similar magnitude as the action potential signal, or often much larger than it. As can be seen in Fig 2b,e and very clearly in Fig S2, there is NO stimulation artefact corresponding to a capacitive current resulting from the stimulation being turned on and off. In Fig S2, a small and slow depolarization is observable during the ultrasound being on, however the magnitude of this depolarization is very similar to that shown later in the trace between 160 – 200 ms. Actual electrical coupling to the cell membrane of a magnitude that would result in AP generation would be apparent in the current or voltage-clamp trace. The lack of any electrical artefact here discounts an electrical stimulation mechanism.

1.2 Latency is inconsistent with electrical stimulation. The paper reports latency to AP generation between 25 ms to 10 ms, with shorter times with higher FUS intensity. Electrical actuation of the cell membrane would have to be within a few ms, after this time electrical field is screened by the electrolyte and the cell membrane does not “feel” any potential difference from an extracellular particle/electrode. This is why extracellular neurostimulation pulses last between a few hundred microseconds, and maximum a few ms. A latency of 25 ms is not consistent with electrical stimulation, and the notion of

“charge buildup” outside of the cell over such time is, as described already, not possible, due to the kinetics of the electric field screening upon formation of capacitive double layers.

1.3 . Geometry of the particles is not well-suited to extracellular electrical stimulation. In extracellular electrical stimulation, it is important to consider cathode/anode, or so-called primary and return electrode – basically from where the current is flowing and to where it is going. There has to be a potential drop across the cell membrane, included by current flowing around the cell. As a reference, consider papers such as 1. Schoen, I. & Fromherz, P. The mechanism of extracellular stimulation of nerve cells on an electrolyte-oxide-semiconductor capacitor. *Biophys. J.* 92, 1096–1111 (2007); 1. Boinagrov, D., Loudin, J. & Palanker, D. Strength–Duration Relationship for Extracellular Neural Stimulation: Numerical and Analytical Models. *J. Neurophysiol.* 104, 2236–2248 (2010). Here, each piezoelectric nanoparticle can charge on the surface, but where is the plus and minus? The charge and electric field are localized on one particle? Therefore by Gauss Law the electric field is located inside of the particle, not outside of it. How can current flow around the cell and create differences in cell membrane potential, even locally? The COMSOL model assumed that the whole sphere as the Au coated surface as one terminal. This would then make sense – but how can current asymmetrically from the NP into the gold ? and more importantly, how can it happen at MHz frequency to generate any useful currents in solution? Which is the next point:

1.4 As I understand the frequency of the FUS of 2 MHz will cause the piezoelectric particles to charge with this frequency. How can MHz frequency stimulation result in ionic currents to flow in solution and cause action upon voltage gated ion channels? This frequency is way above the relaxation frequency of ions. One can make the argument that then there is an electric field that would not be screened, but again, electric field over what geometric space that effects the cell membrane?

2) There is no control measurement / sample for FUS alone, without particles. This is a major oversight. The authors must compare the patch clamp measurements in the absence of particles in order to claim that the particles are doing anything. The only control sample is for the ROS analysis and viability tests, however there is no control for FUS stimulation. How can the authors exclude that it is not FUS alone that is stimulating cell action potential, as is routinely reported in many other studies?

The generation of action potentials appears to be via a thermal mechanism, consistent with the slow change in membrane potential. This is reminiscent of several studies on using laser light and various nano/microparticles to stimulate cells, where a photothermal mechanism is established. Consider the mechanism rationalization guidelines on page 12 of this protocol paper: Nongenetic optical neuromodulation with silicon-based materials. *Nat. Protoc.* 14, 1339–1376 (2019); or Fig 2 of this paper: Rational design of silicon structures for optically controlled multiscale biointerfaces. *Nat. Biomed. Eng.* 2, 508–521 (2018). Moreover, the actual effects of the functional particles in this paper is not possible to determine considering the lack of a control experiment with FUS alone, which is of course well-known to elicit APs in excitable cells.

Response to Reviewers

article: NCOMMS-23-36212A

Revision summary:

In response to the reviewers' comments, we have meticulously addressed each point and implemented revisions guided by their feedback. The paper has undergone substantial modifications, incorporating two new experimental demonstrations, expanded results, additional explanations, five new figures (one main and ten supplementary), one new supplementary note, and two new methods sections. Key revisions and improvements include:

- 1) In-depth control studies on primary neurons using patch-clamp electrophysiology and calcium imaging experiments, investigating control groups with only ultrasound and magnetic microparticles without piezoelectric nanoparticles (**Supplementary Fig. 5, Supplementary Fig. 14**).
- 2) Elaboration on the thermal, mechanical, and piezoelectric effects to address concerns raised by Reviewer #4 regarding the underlying mechanism of neural stimulation. We provided qualitative and quantitative explanations of how PEMP can stimulate primary neurons at high frequencies (**Fig. 4a-e and Supplementary Fig. 15**).
 - We computationally investigated the thermal effects during ultrasound excitation and revealed that potential thermal effects should be less than 0.009 °C under the FUS intensity and pulse repetition frequency used in our study (**Supplementary Fig. 12**).
 - Experimental validation through resistance-temperature measurements confirmed no observed heating higher than 0.0083 °C (**Supplementary Fig. 12**).
 - We carefully characterized the pressure field and acoustic intensity profile of the FUS transducer utilized in this study (**Supplementary Fig. 11**). We calculated the spatial and temporal peak average intensities, and mechanical index of FUS excitation, which further proves that mechanical effects are not dominant in our system and that the used FUS intensity is lower than the threshold intensity of mechanical stimulation of neurons in the literature.
 - To investigate the neural stimulation mechanism further, we carried out channel blocker-based electrophysiology experiments that show neural stimulation via voltage-gated ion channels but not due to mechanosensitive ion channels (**Supplementary Fig. 15 and Fig. 3**).
- 3) We carried out the electrochemical investigation of a single PEMP, indicating piezoelectric current generation in the extracellular medium, further validating the impact of PEMP on neural stimulation (**Supplementary Fig. 4**).
- 4) We carried out the extended-duration biocompatibility assay to demonstrate the in vitro biocompatibility of the PEMP on differentiated neural cells (**Fig. 5d and Supplementary Fig. 17**).

For the convenience of the editor and reviewers, we have highlighted the added and updated text in yellow in the manuscript.

In the following, we have addressed the reviewer comments point by point in black, where the reviewer comments are shown in blue.

Point-by-Point Reviewer Comment Responses:

Reviewer #1

Using ultrasound to wirelessly stimulate nerves to treat diseases is a new means of treating neurological diseases. In this paper, silica-based piezoelectric magnetic Janus microparticles (PEMPs) were developed to enable clinically relevant high-frequency nerve stimulation of primary neurons under low-intensity focused ultrasound. This manuscript is well-written and provides very good application value. Therefore, I recommend publishing this manuscript after minor revisions. Here are the suggestions for the authors.

1. Janus nanoparticles are promising materials. As far as we know, the Janus nanoparticles used in this work should belong to inorganic Janus nanoparticles. In the reports of other biological applications, researchers often consider the toxicity and hydrophilicity of inorganic Janus nanoparticles. Your research has been compared with in vitro cell experiments, and it seems to be less toxic to cells, which is very gratifying. However, in the injection scenario as you described, should we also consider the subsequent metabolism of Janus nanoparticles? Do cell-sized Janus nanoparticles pose a potential threat to human health as microplastics?

Response: We thank the reviewer for their comments.

The particles used in the study are based on inorganic material such as silica in micron size. Silica-based particles have been widely shown to cause negligible toxicity upon contact with biological cells¹⁻⁴. On the other hand, the degradation of silica-based materials is an open research question and is being investigated. It has been previously shown in the literature that mesoporous silica microparticles, also used in our study, have shown degradation both in simulated body fluid and in vivo mice models². Still, the metabolism of silica-based Janus microparticles is an interesting research subject worth investigating. In particular, thin film degradation/erosion in vivo is currently not known and must be elaborated in future studies. Moreover, the fate of such particles in the body is also related to their interactions with immune cells. The interaction of any particle with immune cells is a function of microparticle size, shape, and material⁵. Recent studies demonstrated that the larger silica microparticle sizes create lower immunoreactivity^{6,7}. From the material side, to fully ensure the applicability of the Janus microparticles, more bio-friendly materials such as L1₀-FePt⁸ and zwitterionic coating⁹ may be used to minimize the interactions with the immune cells to render safer interactions with the body.

Compared to microplastics, silica microparticles contain a single molecular element, SiO₂, which is a material that has been comprehensively investigated. Because of its inert chemical structure, SiO₂ does not create any cytotoxic effect by itself. The main reasons for the cytotoxicity caused by silica-based Janus nanoparticles are the particle size and other inorganic components. In our study, we preferred 20 μm spherical microparticles due to their higher biocompatibility and lower immunoreactivity than other SiO₂ particle shapes and sizes^{6,7}. Because of their size, these microparticles cannot be engulfed by cells, which protects the tissue environment from smaller SiO₂ particles as degradation products. While their non-cytotoxic properties are already demonstrated, they could induce foreign body response as an adverse effect, which is a standing problem even for the state-of-the-art silicone-based neural implants that are currently used for clinics¹⁰. In our study, we demonstrated their biocompatibility as an in vitro neuroscientific tool, which is the main claim of the study. For extensive biocompatibility investigation, systemic immune reactions against silica-based Janus microparticles should be investigated in an animal model, which is off-topic for this article. The related limitation of the article is included in the discussion section.

2. Your transmission electron microscopy Figure 1e shows that the size of the Janus nanoparticles prepared by you is about 200 nm, and the silica particles used are about 20 microns (μm). How large is the particle size? Or is the scale of TEM mislabeled? (In addition, TEM does not look uniform in size, which will affect performance.)

Response: We thank the reviewer for their comments. The TEM image (Fig. 1e) shows only the BaTiO_3 nanoparticles (BTNPs), which were conjugated onto the spherical 20- μm -diameter silica microparticles. To avoid any confusion and give more explanation for Fig. 1f, we added the SEM image of bare 20- μm -diameter silica microparticles before BTNP conjugation (Supplementary Fig. 1a). Likewise, we added the bright-field and fluorescence microscope images for readers' convenience (Supplementary Fig. 2a and b). We also added TEM size distribution analysis of BTNPs prepared in two batches and size distribution is calculated in these two experiments (Supplementary Fig. 1b and c). Additionally, dynamic light scattering (DLS) results for BTNPs (Supplementary Fig. 1d) were included to clarify the size distribution of BTNPs used in the conjugation reaction between the BTNPs and 20 micron-diameter silica microparticles. The TEM size distribution shows the mean ~ 260 nm particle diameter with a standard deviation of ~ 43.8 nm. Moreover, the hydrodynamic radius is calculated from DLS measurement as ~ 432.6 nm with a standard deviation of ~ 66 nm. The reviewer's comment on non-uniform size distribution and its effect on the PEMP performance requires attention. In our system, we hypothesize that we leverage the macroscopic piezoelectric effect of all barium titanate nanoparticles on the silica surface. Therefore, the size distribution and piezoelectric performance of individual BTNPs might not have a dominant effect on the overall PEMP performance.

We calculated the maximum potential difference on a single BTNP according to the electro-elastic model of Marino, Suzuki, and Ciofani et al.¹¹. If we consider the same calculation for BTNPs used in our study, we could calculate the maximum potential difference:

$$\varphi_R \equiv - \frac{R(s * e_{rr} + 2e_{r\theta})}{s * \epsilon_{rr}} * \frac{p_{US}}{s * \gamma + 2\alpha}$$

$$\varphi_R \equiv R * p_{US} * k,$$

$$\text{where } k \equiv - \frac{(s * e_{rr} + 2e_{r\theta})}{s * \epsilon_{rr}} * \frac{1}{s * \gamma + 2\alpha} \approx 7.6 \cdot 10^{-3} \left(\frac{\text{V.m}}{\text{N}} \right)$$

where $p_{US} = 40 \text{ kPa}$ and the other parameters such as the water density, sound speed in water¹², viscoelastic, and piezoelectric properties for BTNPs were found from the literature¹¹. If the low-intensity case is considered (50 mW.cm^{-2} FUS intensity), the maximum potential difference on a single BTNP can be calculated as $\sim 0.042 \text{ mV}$ - 0.1412 mV depending on the different dielectric constants found in the literature. Based on the estimation that half of the 20 μm microparticle surface is conjugated with BTNPs, we consider that half of this amount ~ 1100 BTNPs could be found on the microparticle surface. This calculation is based on a simplified model and all individual BTNP potentials on the surface cannot be directly summed up. On the other hand, the single BTNP potential generation of $\sim 0.042 \text{ mV}$ - 0.1412 mV and >1000 BTNPs on the PEMP surface could provide insights that PEMPs could induce sufficient piezoelectric charge to evoke membrane depolarization, which was already observed in patch-clamp and calcium imaging experiments. While the uniform-sized and carefully polled piezoelectric particles would generate better performance, we leave the performance optimization

and size dependent analysis (both the piezoelectric particle size and the silica particle size) for the future study. Moreover, we observed no significant variation between the fabricated PEMPs in terms of neural stimulation performance. According to the reviewer's comments, we included the figures and given information above into the revised manuscript:

Supplementary Fig. 1. **a** SEM image of the unmodified silica microparticles. **b** Example TEM image of BTNPs. A larger field of view could be seen in the inset. **c** Particle size distribution analysis of three different TEM imaging experiments. **d** Dynamic light scattering (DLS) was used to characterize MENP hydrodynamic properties in artificial cerebrospinal fluid (aCSF).

Supplementary Fig. 2. **a** Bright field microscope image of the silica microparticles before the PEMP fabrication. **b** Example fluorescence image of PEMPs in three orientations indicated by dashed circles. The side view represents the condition of the magnetization direction of PEMPs in the xy plane, similarly, the magnetization direction is in +z and -z direction for top and bottom orientations, respectively.

3. Does the thickness of gold (20 nm) and nickel (60 nm) be considered (as cell-sized Janus particle size) or supported by previous work? Why choose such a thickness?

Response: We thank the reviewer for their comments. The thickness for the magnetic nickel layer and gold coating is based on and supported by the previous work from our group^{13,14}. Based on our previous studies, we picked a relatively thinner Ni film to prevent magnetic aggregation of Janus particles⁸ for robust operation. In fact, 60 nm Ni film is more than sufficient to align, and actuate the Janus particles for rotation frequencies up to 100 Hz¹³ at low amplitude rotating magnetic fields (~10 mT) as evidenced in our previous studies^{8,13}. We used 5 Hz as a maximum rotation frequency in our study. Moreover, gold thickness is sufficient to build a barrier between the extracellular medium and nickel film, which promotes biocompatibility by isolating the nickel from the wet and ionic environment. While larger nickel thickness could provide more robust actuation in more viscous and challenging environments, in this study, we concentrated on the neural stimulation performance of PEMPs rather than the magnetic steering and locomotion capabilities. The later would be particularly important for our next study to optimize the locomotion capabilities by the magnetic material, thickness, and coating material choices.

Reviewer #2

The study focuses on the development of silica-based piezoelectric magnetic Janus microparticles (PEMPs) combined with low-intensity focused ultrasound (LIFU) to enable high-frequency neural stimulation of primary neurons. The microparticle designed by the authors is innovative and the applicability is multifaceted due to the possibility of antibody targeting functionalization. The manuscript includes the fabrication of the piezoelectric-magnetic microparticles, as well as multiple aspects of in vitro response of different neural cells treated with PEMP alone or PEMP in combination with LIFU. The results reported in the manuscript are well described and the novelty of the work is sufficiently clear, the data interpretation is appropriately described. Despite these positive elements, some important aspects need to be deepened.

To strengthen the work and make it suitable for publication in Nature Communications I strongly suggest the authors to further investigate the following aspects related to the innovative system that they are willing to publish:

- Investigate the colloidal stability of PEMP in water and cell culture media. The dimension of the final microparticles is reported by the authors to be equal to 20 μm according to TEM measurements, but do the authors have any data about their dimension and their dimension distribution in water or in cellular medium? For how long will the particle structure remain stable in time and in different media? Are there at any point in time some aggregation phenomena? Is it possible that the BaTiO₃ nanoparticles (BTNP) conjugated on one half of the microparticles begin to dissolve and release ions which are toxic for cells?

Response: We thank the reviewer for their comments. We report 20 μm -diameter silica particles, which have half of their surface coated with magnetic thin film and the other half of their surface conjugated with ~ 260 nm BTNPs. The size of the silica particles was verified via SEM imaging (Fig. 1f and Supplementary Fig. 1a) while the size of the BTNPs was verified via TEM imaging (Fig. 1e and Supplementary Fig. 1b). To answer the reviewer's question, we did dynamic light scattering (DLS) measurements of BTNPs, which resulted in ~ 432.6 nm with a standard deviation of ~ 66 nm hydrodynamic radius of BTNPs (Supplementary Fig. 1d).

During our experiments, we investigated structural and functional PEMP stability in several ways. Firstly, we did a functional stability test for PEMP stored at 37 °C in PBS and 30 mM H₂O₂ added to PBS. This test evaluates the neural stimulation success of PEMP and indicates robust performance up to 200 days with $\sim 22.6\%$ and $\sim 28.2\%$ decrease in the success rate in PBS and PBS+H₂O₂, respectively. Since the neural stimulation success depends on the piezoelectric field generated by the BTNPs on the silica particle surface, we may claim that most BTNPs were still on the silica surface and induced similar performance as of the initial fabrication. Also, we did not observe any significant toxic effect of PEMP under no ultrasound and ultrasound conditions on differentiated SH-SY5Y neural cells for 96h (Fig. 5d and Supplementary Fig. 17b). Moreover, the PEMP are stable for months in DI water and aCSF. While we experienced aggregation of magnetic Janus microparticles without BTNPs for months of storage at +4 °C, this is due to the magnetically induced attachment of particles. Once the magnetic microparticles were vortexed and freshly washed, magnetic clustering could be eliminated. Likewise, inducing the rotational magnetic field to the particles also disaggregates them upon electromagnetic actuation. On the other hand, BTNP conjugation on the magnetic microparticles increases the monodispersity of PEMP.

While we focused on the PEMP as an in vitro neurostimulation platform in this study, we plan to evaluate long-term toxicity and functional stability for potential in vivo applications in our next studies. Other than that, the conjugation method used in the study includes stable chemical bondings and interactions. Other than stable chemical bondings used in the study as NHS-Amine coupling, streptavidin-avidin interactions are the strongest known non-covalent interactions¹⁵, widely used in a plethora of applications. Also, it has been shown that the bonding is unaffected by extreme pH, temperature, organic solvents, or denaturing agents¹⁵⁻¹⁷; even, this interaction cannot be disturbed by many detergents. According to the reviewer's comments, we have now included the figures and text shown below in the revised manuscript:

Supplementary Fig. 1. **a** SEM image of the unmodified silica microparticles. **b** Example TEM image of BTNPs. A larger field of view could be seen in the inset. **c** Particle size distribution analysis of three different TEM imaging experiments. **d** Dynamic light scattering (DLS) was used to characterize MENP hydrodynamic properties in artificial cerebrospinal fluid (aCSF).

The following text was added to the methods section for DLS measurements:

“The hydrodynamic size and size distributions of BaTiO₃ nanoparticles were determined by Dynamic Light Scattering (DLS) (Möbius, Wyatt Technologies) with backscattering detection at a scattering angle of 163.5° and a wavelength of 532 nm. The solutions were diluted to 0.1 mg/mL in phosphate-buffered saline (1x) before the measurement. Each measurement was repeated three times and the average of the results was reported.”

-Investigate the long-term toxicity of PEMP on neural cells. The work proposed by the authors investigates cell viability after 24h from PEMP administration or ultrasound stimulation, but the effects on cell viability after 48, 72, and 96 h were not studied. This data is fundamental to effectively evaluate the biocompatibility of PEMP.

Response: We thank the reviewer for their comments. We extended the cell viability test from 48 hours to 96 hours with LDH cell membrane integrity assay. We investigated the cell viability with and without the PEMP and focused ultrasound stimulation on SH-SY5Y differentiated neurons. LDH assay indicated no statistically significant difference between the control and experimental groups (Fig. 5d and Supplementary Fig. 17). We also included these experiments in the main figure with a detailed explanation in the figure legend and the related part of the text.

Fig. 1d The effect of PEMP on differentiated SH-SY5Y cells under 1 particle/cell condition for 24, 48, 72, and 96h.

Supplementary Fig. 1 Investigation of long-term neural cell viability with LDH assay. **a** The effect of the PEMP on the cellular membrane integrity of the differentiated SH-SY5Y cells for 0, 0.1, 1, and 10 particles/cell conditions assessed by the LDH assays. **b** LDH cell viability experiments of three experimental groups on differentiated SH-SY5Y cells for 96h in 1 particle/cell condition; under the excitation of FUS for 30 min with and without the PEMP, and only PEMP without the FUS excitation. All data is represented as mean \pm s.d. Statistical significance is determined by the two-sided Student's t-test and * $p < 0.05$ was considered as statistically significant. ns, not significant.

Furthermore, do the authors have any information, based on experimental knowledge or previous literature, about the clearance of their microparticles?

Response: We thank the reviewer for their comments.

We thank the reviewer for their comments. The clearance of the microparticles depends on several factors, including surface charge, material, size, shape, tissue, and delivery method^{18,19}. Because of that, each microparticle should be investigated in its specific biomedical application.

During the nontoxic and nonreactive material implantations into the central nervous system, brain tissue induces minimal foreign body response with negligible inflammation, minimal fibrosis, and limited resorption¹⁸. Especially for the spherical-shaped implants, this foreign body response is reduced significantly²⁰. For larger particles than 10 μm , the cellular clearance mechanisms, including opsonization and phagocytosis, do not work²¹, and they leave their place to tissue remodeling processes, including fibrosis²⁰. While these processes can cover the implanted microparticles, their

effect on the material surface will be significantly lower than macrophage-based cellular clearance mechanisms. Both cellular clearance and tissue remodeling are coordinated by the inflammatory response of the resident microglia of the central nervous system²². While we investigated the biocompatibility and immunoreactivity of PEMP with microglia and astrocytes, we also examined their morphological and cytokine responses to these particles. Thanks to the inert nature of the SiO₂, the PEMPs do not induce any reactive or toxic effects on astrocytes and microglia, the immune-responsive components of the central nervous system (Fig. 5e and f, and Supplementary Fig. 18 and Fig. 19). This result correlates with previous literature on silica-based microparticles^{6,7}. Although the long-term tissue-level clearance mechanisms in the central nervous can be investigated, these investigations will cause the study to lose its focus on spatially controllable neurostimulation. The related discussion is included in the manuscript.

We added the explanations above to the discussion part of the manuscript:

“Moreover, during the experiments detailed in the present study, we did not observe obvious side effects of the LIFU excitation and PEMP on primary neurons, astrocytic and microglial cells. Especially for the spherical-shaped implants, the foreign body response is reduced significantly⁷⁷. For larger particles than 10 μm, the cellular clearance mechanisms, including opsonization and phagocytosis, do not work⁷⁸, and they leave their place to tissue remodeling processes, including fibrosis⁷⁷. While these processes can cover the implanted microparticles, their effect on the material surface will be significantly lower than macrophage-based cellular clearance mechanisms. Despite these advantages and although it is fully covered by the Au thin film, the Ni metal film may not be ideal for long-term use in vivo due to potential material delamination and consequent toxicity effects. Alternatively, we would use more biocompatible magnetic films, such as L1₀ phase of FePt⁷⁹, which would provide higher biocompatibility for long-duration applications. Fine-tuning the magnetic and piezoelectric properties, along with exploring alternative materials, holds promise for improving the piezoelectric response and reducing ultrasound intensity thresholds, allowing even more precise control over neural stimulation.”

Minor editing of English language are required.

Response: We checked and revised the manuscript for language mistakes.

Reviewer #3

Han et al. report piezoelectric magnetic Janus microparticles (PEMPs) for neural stimulation. This is a well-organized, written and interesting study. Especially by looking to the literature using calcium imaging dyes for neurostimulation sensing, the use of patch-clamp electrophysiology, which is a difficult one, is a great contribution to the field that can help future studies with a more quantitative understanding of neurostimulation via piezoelectric nanoparticles.

One surface of the spherical silica microparticles are coated with BaTiO₃ nanoparticles, while the other surface is magnetically responsive. The manuscript provides a detailed explanation of the PEMP and their capabilities. A wide variety of electrophysiology and cellular experiments were nicely done even by changing the orientation of the PEMP. I recommend the publication of the manuscript after minor revisions.

There are several points that can further improve the study:

1) The mechanisms underlying the neural stimulation process and the physical principles governing the behavior of PEMP need to be elucidated in more detail. This would enhance the understanding of the neuron and PEMP interaction.

Response: We thank the reviewer for their comments and recommendation for publication. We agree with the reviewer that the mechanism behind neural stimulation should be investigated in more detail. To do that, we started our investigations by evaluating possible thermal effects due to FUS. Firstly, we modeled our FUS transducer in COMSOL Multiphysics® v6.1 software (COMSOL AB, Sweden) by the recorded pressure waves using a needle hydrophone. To realize this, we modeled a water and tissue domain in COMSOL. We utilize Pressure Acoustics in the frequency domain to obtain the acoustic pressure field (Supplementary Fig. 11a, b, and c) and integrated Bioheat Transfer module to calculate the heating and cooling of the tissue phantom via Penné's bioheat equation. Thermal simulations were performed in a two-fold process corresponding to a worst-case scenario, propagation in a water medium, and thermal absorption in a brain-mimicking medium. The following parameters²³ were followed for the propagation medium (water): sound speed, $c = 1,500 \text{ m s}^{-1}$; volumetric mass, $\rho = 1,000 \text{ kg m}^{-3}$; nonlinearity coefficient, $B/A = 5$; attenuation coefficient, $\alpha = 2.2 \times 10^{-3} \text{ dB cm}^{-1} \text{ MHz}^{-\gamma}$; frequency power law of the attenuation coefficient, $\gamma = 2$. COMSOL simulations were calibrated by adjusting the input pressure to match the pressure at the focus measured in the water tank by the hydrophone (Supplementary Fig. 11d, e, and f). In the second part of the simulation, we utilized the heat transfer module in the tissue domain with the parameters: brain volumetric mass $\rho_{\text{brain}} = 1046 \text{ kg m}^{-3}$, the brain sound speed $c_{\text{brain}} = 154 \text{ s}^{-1}$, K_t is the brain thermal conductivity ($0.51 \text{ W m}^{-1} \text{ }^\circ\text{C}^{-1}$) with the initial brain temperature $T_0 = 37 \text{ }^\circ\text{C}$ ²³. Once we have the heat generation due to acoustic wave absorption in the tissue domain, we can simulate the transient heating/cooling cycles due to 0.5-200 Hz modulated 2 MHz FUS excitation. Supplementary Fig. 12 presents the simulation and experimental heating investigations. First, we utilized single US pulses with 10-1000 ms of 2 MHz pressure waves and please note the rise/fall regions of the pulse (Supplementary Fig. 12a). We investigated the single pulse heating (Supplementary Fig. 12b) and utilized this information to simulate 0.5-50 Hz stimulation for 10 seconds (Supplementary Fig. 12c). The results indicate that increasing the stimulation frequency decreases the overall heating and even for the lowest frequency we did not expect $>0.009 \text{ }^\circ\text{C}$ heating during 10-sec stimulation. To verify our simulations, we carried out local temperature measurements^{24,25} by patch pipette resistance changes. First, we obtained the resistance-temperature calibration (Supplementary Fig. 12d). Then, we positioned the same patch pipette to the focal point of the FUS at the bottom of the recording chamber. The same pulse and frequency-dependent analysis was carried out and compared with the simulation results (Supplementary Fig. 12e and f). Simulations were well-matched with the experimental temperature measurements and indicate that the FUS intensity used in this study does not generate significant local heating and is much lower than the reported temperature changes for thermal activation of neurons^{24,26-28}.

Supplementary Fig. 11. Experimental and numerical investigation of pressure waves generated by the FUS transducer. **a** 2D representation of ultrasound transducer/water/tissue phantom system. Model parts are marked in the figure. The model possesses the axial symmetry that enables three-dimensional reconstruction of the 2D computations. Numerical results and corresponding heat maps of **b** absolute acoustic pressure in xy plane at the center of the FUS focus and **c** acoustic intensity magnitude in zr plane. **d** Experimental and simulation results of transient pressure waves for a single burst pulse of FUS at 2 MHz. Acoustic pressure profile along **e** the symmetry axis and **f** radial direction in the focal plane.

Supplementary Fig. 2 Evaluating thermal effects by numerical simulations and resistance-temperature measurements. **a** FUS driving signal for temperature measurements. The driving signal is at 2 MHz modulated with a single pulse with a rise and fall time of 10 ms. Inset shows the rise and fall time. **b** Heating calculations of brain phantom under continuous $100 \text{ mW} \cdot \text{cm}^{-2}$ FUS for changing pulse durations. The inset shows 2 MHz heating/cooling cycles. **c** Heating calculations of brain phantom under $2 \text{ MHz } 100 \text{ mW} \cdot \text{cm}^{-2}$ FUS for changing pulse frequencies. **d** Calibration curves using a linear fit of recorded pipette

resistances and corresponding extracellular medium temperatures. The linear fit shows the relationship between the pipette resistance and temperature. This calibration was utilized to measure the temperature increase under 100 mW.cm⁻² FUS for **e** changing pulse frequency and **f** duration. The bar plots in **e** and **f** demonstrate the numerical calculations and experimental results for $n = 7$ and $n = 5$ independent experiments for pulse frequency and duration, respectively.

On the other hand, we operate in the diagnostic, nonthermal, noncavitational (<100 mW.cm⁻²) spatial peak temporal average intensity levels^{29–31}. Our excitation parameters are within the low-intensity US regime and lower than the observed and proposed intensities in the literature³² for thermal, cavitation, microtubule resonance, and mechanosensitive stimulation mechanisms in terms of the temporal and spatial average intensity of FUS. We calculated the intensity characteristics of FUS stimulus based on the standards developed by the National Electronics Manufacturers Association³³, The American Institute of Ultrasound Medicine, and the United States Food and Drug Administration (FDA), *Marketing Clearance of Diagnostic Ultrasound Systems and Transducers*. By utilizing the measurements recorded from the calibrated needle hydrophone, the pulse intensity integral (PII), the spatial-peak pulse-average intensity (I_{SPPA}), the spatial-peak, temporal-average intensity (I_{SPTA}), and the mechanical index were calculated as

$$PII = \int \frac{p^2(t)}{Z_0} dt$$

where p is the instantaneous peak pressure, Z_0 is the characteristic acoustic impedance in Pa s/m defined as ρc where ρ is the density of the medium, and c is the speed of sound in the medium. For the safety considerations, we used the brain tissue as the propagation and focus medium, where the brain volumetric mass $\rho_{\text{brain}} = 1,046 \text{ kg.m}^{-3}$, and the brain sound speed $c_{\text{brain}} = 154 \text{ m.s}^{-1}$.

$$I_{SPPA} = \frac{PII}{PD}$$

where PD is the pulse duration, and

$$I_{SPTA} = PII * PRF$$

where PRF is equal to the pulse repetition frequency in Hz. The mechanical index was defined as

$$MI = \frac{p_r}{\sqrt{f}}$$

Our calculations indicate spatial-peak temporal-average intensities (I_{SPTA}) of 3.8–28.7 mW.cm⁻² for a total stimulus duration ranging between 2.5 and 500 ms. FUS waveforms had peak rarefactional pressures (p_r) of 0.014–0.108 MPa, pulse intensity integrals (PII) of 0.017–0.095 mJ.cm⁻², and spatial-peak pulse-average intensities (I_{SPPA}) of 0.027–0.267 W.cm⁻². The mechanical index (MI) was calculated as 0.028–0.0864 for the lowest and highest rarefactional pressures. These results indicate that FUS intensity values used in this study are not sufficient to evoke significant thermal or mechanical effects that might lead to stimulation of primary hippocampal neurons, especially >10 Hz neural stimulation^{32,34–37}. In another hypothesis, Krasovitski et al.²⁹ developed a model mechanism, called intramembrane cavitation, for ultrasound-induced neural stimulation at the noncavitational low-intensity regime. They claim and model that in these low intensities ultrasonically induced bilayer membrane motion, which does not require the preexistence of air voids in the tissue, could induce neural stimulation. In their extended model on neural cells in mouse motor cortex³⁸, they provide threshold intensity and ultrasound duration to generate a single action potential. If we compare the intensity and energy thresholds, PEMP could elicit action potentials at 50 Hz with a high success rate under an order of magnitude less intensity and about two orders of magnitude less energy than

required for neural stimulation by direct ultrasound effects³⁸. Therefore, we may also rule out the proposed intramembrane cavitation effects, which supports our claim that PEMP's under FUS excitation induce neural stimulation via piezoelectric charge generation.

Having ruled out the potential heating and cavitation effects via experimental data and numerical calculations, we followed a systematic approach to determine the dominant electrophysiological basis of the neural response to PEMP's excited with LIFU. As suggested by the reviewer, we started the control experiments with FUS stimulation without any particles and FUS + magnetic Janus particles (without BTNPs) by utilizing both current-clamp recordings (Supplementary Fig. 5) and calcium imaging experiments (Supplementary Fig. 14). We did not observe recording artefacts due to FUS or the presence of magnetic Janus microparticles (Supplementary Fig. 5) under $100 \text{ mW}\cdot\text{cm}^{-2}$ FUS intensity. Moreover, calcium imaging experiments showed no significant difference in the area under the curve for control groups, while the FUS+PEMP group showed significantly higher calcium intensities (Supplementary Fig. 14). Therefore, we ruled out the neural stimulation due to FUS excitation or due to magnetic Janus microparticles, and we can claim that barium titanate nanoparticles are the responsible transducers for neural stimulation.

Supplementary Fig. 3 Control experiments on primary neurons under LIFU with and without magnetic Janus particles. **a** Representative membrane potential traces of primary neurons excited under LIFU (two pulses of 5 ms $100 \text{ mW}\cdot\text{cm}^{-2}$ FUS) with and without magnetic Janus microparticles.

Supplementary Fig. 4 Control experiments on primary neurons under LIFU with and without magnetic Janus particles. Calcium responses of primary neurons without any particles, with magnetic Janus particles, and with PEMPs under 50 ms, 50, and 100 mW.cm⁻² FUS pulses ($n = 4$ independent experiments, each dot represents the mean of 10 individual neuron responses).

Additionally, we carried out a comprehensive channel blocker study using calcium imaging to understand the mechanism behind the stimulation of primary neurons via PEMPs under FUS. We started the study without any channel blocker and compared the area under the curve of spontaneous activity of primary neurons, and their activity under FUS and FUS+PEMPs. We found no significant difference between the spontaneous activity and under FUS (Supplementary Fig. 15a and Fig. 3a). Secondly, we evaluated the neural response via the application of synaptic blockers (CNQX and AP5), which blocks postsynaptic excitatory receptors, AMPA and NMDA, respectively³⁹, to understand whether the neural response is autonomous or requires synaptic connections with other neurons⁴⁰. While we observed a reduction in the area under the curve for control samples without PEMPs, we did not observe a statistically significant difference for neural stimulation induced by PEMPs (Supplementary Fig. 15b). This shows that PEMP-based stimulation does not effectively require synaptic transmission between neurons.

Then, we started treating neurons with different channel blockers while the dose of tetrodotoxin (TTX), cadmium (Cd⁺²), gadolinium (III) (Gd⁺³), and ruthenium red (RR) was carefully chosen to avoid blocking other channels or altering cell excitability (Supplementary Fig. 15). We blocked the voltage-gated sodium channels with TTX and observed reduced calcium intensity during activation for both the control groups and the PEMP group. However, the reduction in the magnitude was significant for the PEMP group, indicating the role of voltage-gated sodium channels during neural activation (Supplementary Fig. 15c and Fig. 3b). Then, neural responses were recorded with the addition of Cd⁺² (100, 200, and 400 μ M), which prevents the activation of voltage-gated calcium channels^{36,41,42}. We observed a significant reduction in the magnitude for both control and PEMP groups for 400 μ M Cd⁺² concentration, for 100 μ M Cd⁺² the reduction was significant for only the PEMP group (Supplementary Fig. 15d and Fig. 3c). The residual calcium transients not blocked by TTX and Cd⁺², 24% and 19% of the peak magnitude (Supplementary Fig. 15c and d), respectively, could be attributed to the noncomplete block of the voltage-gated sodium and calcium channel population⁴³, to other calcium sources in primary hippocampal neurons or involvement of other channels, which we investigated next. While the control experiments in patch-clamp recording and calcium imaging under FUS without PEMPs

indicated that in the FUS intensity regime of interest, FUS itself did not induce any neural stimulation, we utilized gadolinium (III) to modify the deformability of the lipid bilayer and non-specific inhibition of mechanosensitive channels^{40,44,45}. We observed a small <15% decrease in the PEMP group which indicates that the mechanosensitive channel population is not dominant or active in neural stimulation via PEMPs under FUS (Supplementary Fig. 15e and Fig. 3d). Since gadolinium (III) is not specific and changes the overall mechanical properties of neural membranes, Ruthenium Red was applied as a pore blocker of TRPV1, TRPV2, and TRPV4 channels^{40,43,46}. We did not observe a significant decrease in the calcium signal intensity or excitability for the PEMP-induced neuron group (Supplementary Fig. 15f and Fig. 3e). As a result, considering the TTX and Cd²⁺ blocker experiments, these observations indicate that PEMPs under FUS increase the Na⁺ conductance in primary neurons and trigger Ca²⁺ transients dominantly via voltage-gated sodium and calcium channels. The potential effect of synaptic transmission and mechanical activation were not involved or significantly contributed to the neural stimulation. Therefore, the electrochemical characterization of a single PEMP response under FUS, experimental characterization of heating and calculation of mechanical effects, and channel blocker-dependent experiments suggest that the neural stimulation was induced via the opening of voltage-gated sodium and calcium channels. We added the information provided here to the results section in a new subsection, while the experimental procedures were added to the methods section.

Supplementary Fig. 5 PEMP-induced neural stimulation under LIFU is mediated by the voltage-gated sodium and calcium channels. **a** Calcium response under different LIFU intensities with and without the PEMPs ($n = 4$ independent experiments for control and experimental groups.). **b** Calcium responses before and after treatment with synaptic blockers, AP5 and CNQX. **c** Calcium responses before, after, and washout treatment with TTX. **d** Calcium responses before and after treatment with Cd²⁺. **e** Calcium responses before and after treatment with Gd³⁺. **f** Calcium responses before and after treatment with Ruthenium Red. For all experiments in b, c, d, e, and f, $n = 5$ independent experiments for control and experimental groups. Bar graph values are represented as mean \pm SD. Two-tailed, unpaired and paired T-tests were utilized for statistical analysis (* $p < 0.05$, ** $p < 0.01$, *** $p < 0.001$).

Fig. 3. PEMP-induced neural stimulation under LIFU is mediated by the voltage-gated sodium and calcium channels. **a** Comparison of area under the curve of calcium response of spontaneous activity and neural activity during FUS excitation with and without PEMPs ($n = 4$ independent experiments). **b** Neural activity changes before and after tetrodotoxin (TTX) treatment to block sodium channels. Quantification of area under the curve before and after **c** cadmium, **d** gadolinium (III), and **e** ruthenium red (RR) treatment to block calcium channels, global mechanosensitive channels nonspecifically, and TRPV channels, respectively. For all experiments in **b**, **c**, **d**, and **e**, $n = 5$ independent experiments for control and experimental groups. Bar graph values are represented as mean \pm SD. Two-tailed, unpaired and paired T-tests were utilized for statistical analysis (* $p < 0.05$, ** $p < 0.01$, *** $p < 0.001$).

We added the following text to the main manuscript:

“PEMPs induce piezoelectric activation of voltage-gated ion channels under low-intensity FUS excitation

Having investigated the neural stimulation performance of PEMPs via patch-clamp electrophysiology experiments, we carried out a comprehensive channel blocker study to understand the mechanism behind the stimulation of primary neurons via PEMPs under FUS. We started the calcium imaging experiments without any channel blocker and compared the area under the curve of spontaneous activity of primary neurons and their activity under FUS, FUS+magnetic Janus particles (Ni/Au coating on bare 20 μm silica microparticles, Supplementary Fig. 14), and FUS+PEMPs (Supplementary Fig. 15a) conditions. We found no significant difference between the spontaneous activity and under FUS, while the area under the curve significantly increased via PEMP-induced effects (Fig. 3a). Secondly, we evaluated the neural response via the application of synaptic blockers (CNQX and AP5), which blocks postsynaptic excitatory receptors, AMPA and NMDA, respectively³⁹, to understand whether the neural response is autonomous or requires synaptic connections with other neurons⁴⁰. While a reduction in the area under the curve was observed for control samples without PEMPs, we did not observe a statistically significant difference for neural stimulation induced by PEMPs (Supplementary Fig. 15b). This shows that PEMP-based stimulation does not effectively require synaptic transmission between neurons. We then started treating neurons with different channel blockers while the dose of tetrodotoxin (TTX), cadmium (Cd^{+2}), gadolinium (III) (Gd^{+3}), and ruthenium red (RR) was carefully chosen to avoid blocking other channels or altering cell excitability (Supplementary Fig. 16c, d, e, and f). We blocked the voltage-gated sodium channels with TTX and observed reduced calcium intensity during activation for both the control groups and the PEMP group. However, the reduce in the magnitude was significant for the PEMP group, indicating the role of voltage-gated sodium channels during neural activation (Fig. 3b). Then, neural responses were recorded with the addition of Ca^{+2} (100, 200, and 400 μM), which prevents the activation of voltage-gated calcium channels^{36,41,42}. We observed a significant reduction in the magnitude for both control and PEMP groups for 400 μM Ca^{+2} concentration, for 100 μM Ca^{+2} the reduction was

significant for only the PEMP group (Supplementary Fig. 15d and Fig. 3c). The residual calcium transients not blocked by TTX and Cd^{+2} , 24% and 19% of the peak magnitude (Supplementary Fig. 15c and d), respectively, could be attributed to the noncomplete block of the voltage-gated sodium and calcium channel population⁴³, to other calcium sources in primary hippocampal neurons or involvement of other channels, which we investigated next. While the control experiments in patch-clamp recording and calcium imaging under FUS without PEMPs indicated that, FUS itself did not induce any neural stimulation, we utilized gadolinium (III) to modify the deformability of the lipid bilayer and non-specific inhibition of mechanosensitive channels^{40,44,45}. We observed a small <15% decrease in the PEMP group which indicates that the mechanosensitive channel population is not dominant or active in neural stimulation via PEMPs under FUS (Fig. 3d and Supplementary Fig. 15e). Since gadolinium (III) is not specific and changes the overall mechanical properties of neural membranes, Ruthenium Red was applied as a pore blocker of TRPV1, TRPV2, and TRPV4 channels^{40,43,46}. We did not observe a significant decrease in the calcium intensity or excitability for the PEMP-induced neuron group (Supplementary Fig. 16f and Fig. 3e). As a result, considering the TTX and Cd^{+2} blocker experiments, these observations indicate that PEMPs under FUS increase the Na^+ conductance in primary neurons and trigger Ca^{+2} transients dominantly via voltage-gated sodium and calcium channels. The potential effect of synaptic transmission and mechanical activation were not involved or did not significantly contribute to the neural stimulation. Therefore, the electrochemical characterization of a single PEMP response under FUS, experimental characterization of heating and calculation of mechanical effects, and channel blocker-dependent experiments suggest that the neural stimulation was induced via the opening of voltage-gated sodium and calcium channels.”

2) The authors can report the voltage and/or current via patch-clamp electrophysiology setup. To enhance the effect, array of PEMPs may be assembled on a glass substrate if required.

Response: We thank the reviewer for their comments. We followed the method in previously described photo-⁴⁷ and sono-electrochemical⁴⁸ measurement procedures for single particles. We first prepared a very low concentration of PEMPs in live cell imaging solution put single particles inside of pulled glass patch pipettes, and mounted them onto the patch-clamp system (Supplementary Fig. 4a). The glass patch pipettes had tips with a diameter of <2 μm . The measurement pipette was positioned on the bottom surface of the recording chamber to its position at in vitro experiments on primary neurons. The system voltage is held at zero in voltage-clamp mode to generate the virtual ground in the system. We used the same filtering settings used in the patch-clamp electrophysiology experiments mentioned in the methods section. When the FUS pulses were applied to the recording chamber without any particles, we did not observe any current deviation from the baseline signal (Supplementary Fig. 4b). Later, we enclosed a single PEMP as described above and applied ultrasound pulse intensities of 10, 20, 50, and 100 $\text{mW}\cdot\text{cm}^{-2}$ FUS while keeping the excitation period at 10 ms (Supplementary Fig. 4c). Although piezoelectric currents are in sinusoidal form for sinusoidal pressure transients, we observed a rising period and sustained current generation, particularly under >20 $\text{mW}\cdot\text{cm}^{-2}$ FUS. Following the end of the FUS pulse, the generated current decays and reaches the baseline before the stimulation pulse. While this experiment proves the piezoelectric current generation of single PEMPs via BTNPs on its surface, it is hard to comment on the exact electrochemical process for the piezoelectric current generation. The recorded current waveforms show a monophasic behavior and no negative polarity after the FUS excitation. Therefore, we can hypothesize that the piezoelectric current generation suggests faradaic processes, which could be due

to the electron transfer between BTNPs and the ionic extracellular medium. As we did not observe any current generation under FUS without PEMP and observed an increase in the sustained current levels with increasing FUS intensity, we could claim that recorded current transients represent piezoelectric current generation. It is also important to note that, current transients may not fully represent the actual piezoelectric currents in the ionic media since the low amplitude current peaks might be screened in the extracellular medium and fast current peaks will be inevitably filtered out via low pass filters during signal amplification. The explanations above were added to the manuscript and the corresponding experimental procedure was added to the methods section.

Supplementary Fig. 4. Single PEMP recordings to investigate piezoelectric current generation under FUS. **a** Experimental schematic of piezoelectric current measurement system. **b** Current trace from glass patch pipette with no PEMP under two pulses of 10 ms 100 $\text{mW}\cdot\text{cm}^{-2}$ FUS. This is a representative trace from $n = 18$ measured traces. Blue rectangles represent the time points of FUS excitation. **c** Piezoelectric current traces from a single PEMP under 10 ms 10, 20, 50, and 100 $\text{mW}\cdot\text{cm}^{-2}$ FUS. The potential was held at zero in voltage-clamp mode.

We added the following text to the main manuscript to discuss the reviewer's comments:

“Estimation of single PEMP behavior under LIFU

We started our investigations by measuring the single PEMP response in an interconnect-free configuration^{47,48} using a patch clamp system under LIFU. To accomplish that, we built a measurement system consisting of a patch-clamp amplifier system integrated into an upright fluorescence microscope, a water tank, and a focused ultrasound probe with a 2 MHz center frequency (Supplementary Fig. 3). We first prepared a very low concentration of PEMP in live cell imaging solution put single particles inside of pulled glass patch pipettes, and mounted them onto the patch-clamp system (Supplementary Fig. 4a). The measurement pipette was positioned on the bottom surface of the recording chamber to its position to be used in vitro experiments on primary neurons. The system voltage is held at zero in voltage-clamp mode to generate the virtual ground in the system. When the FUS pulses were applied to the recording

chamber without any particles, we did not observe any current deviation from the baseline signal (Supplementary Fig. 4b). Later, a single PEMP was enclosed and LIFU intensity of 10, 20, 50, and 100 mW.cm⁻² was applied while keeping the excitation period at 10 ms (Supplementary Fig. 4c). We observed unipolar piezo-electrochemical current with a rising period and sustained current generation, particularly under LIFU intensity of >20 mW.cm⁻². Following the end of the LIFU pulse, the generated current decays and reaches the baseline before the stimulation pulse. Therefore, we could hypothesize that the piezoelectric current generation suggests Faradaic processes⁴⁹, which could be due to the electron transfer between BTNPs and the ionic extracellular medium. As no current generation under LIFU without PEMPs was observed and the sustained current levels increased with higher LIFU intensity, we could claim that recorded current transients represent piezoelectric current generation. In addition, current transients may not fully represent the actual piezoelectric currents in the ionic media since the low amplitude current peaks might be screened in the extracellular medium and fast current peaks will be inevitably filtered out due to the system bandwidth and via low pass filters during signal amplification. Once the piezoelectric current generation of PEMPs was explored, we proceeded to electrophysiology experiments to evaluate the neural modulation potential of PEMPs.”

3) How is the procedure applied? Are cells cultured first and then PEMPs injected? How many of them were injected? Write the experimental protocol on interfacing of PEMPs with membrane step-by-step into the methods (for the patch-clamp experiments).

Response: We thank the reviewer for their comments. The procedure starts with the primary neuron culture without any PEMPs present. On the day of the experiment for electrophysiological studies, we first changed the cell medium with the Live Cell Imaging Solution (LCIS) + 15 mM D-glucose (since LCIS does not contain any form of glucose) and added the PEMPs into the extracellular medium in <0.1 particles/cell condition. The particle concentration is kept at these levels to eliminate many particle problems and reduce potential interference between particles. Particularly, for patch-clamp and orientation-dependent experiments, we looked for PEMPs individually standing in the field of view to understand the interaction between single PEMPs and nearby neurons. We patched the neurons before the FUS excitation and monitored their health and excitability. In the orientation-dependent experiments, electromagnetic actuation was provided for reorienting the PEMPs for the desired configuration before the patching. We provided more details and step-by-step procedures in the methods section:

“In vitro electrophysiology experiments

The patch-clamp electrophysiology experiments on cultured primary neurons were carried out by the patch-clamp amplifier (Axopatch 200B, Molecular Devices, CA, USA) connected to the low-noise data acquisition system (Axon Digidata 1550B, Molecular Devices, CA, USA). Patch pipettes were pulled by a P-2000 micropipette puller (Sutter Instrument, CA, USA) to obtain 10-12 MΩ resistance. On the day of the experiment, the primary neurons without any PEMPs were placed in the measurement chamber, which was filled with the fresh extracellular medium (Invitrogen™ Live Cell Imaging Solution (LCIS), Fisher Scientific + 15 mM D-glucose). We then added PEMPs in 0.1 particle/cell concentration to prevent potential mixed effects of many particles and to reduce interference between the PEMPs. The measurement pipettes were filled with the intracellular solution (Internal KF 110, Nanion, Munich, Germany) for whole-cell patch-clamp configuration. An Olympus microscope with a Prime BSI Scientific CMOS (sCMOS) digital camera was used to monitor the patching

procedure. We looked for PEMPs individually standing in the field of view to understand the interaction between standalone single PEMPs and nearby neurons. The cells were patched prior to the FUS excitation and their health and excitability were monitored. In the orientation-dependent experiments, electromagnetic actuation was provided for reorienting the PEMPs for the desired configuration prior to the patching. For the experiments with FUS excitation, we modified the microscope system by removing the microscope condenser and switching to fluorescence imaging. To reduce vibrations due to FUS excitation, the bottom of the water tank was coated with acoustic isolators. The Ag/AgCl ground electrode was immersed in the extracellular medium as the measurement ground. The temperature of the measurement chamber was monitored during all patch-clamp experiments. The extracellular medium was refreshed in each 30 min by half medium change with LCIS.”

4) Patch-clamp FUS+ metallic side (without piezoelectric nanoparticles) is useful. There may be some local effects triggered by the metal and FUS interactions.

Response: We thank the reviewer for their suggestions. We did additional control experiments for only FUS and FUS + magnetic Janus particles (denoted as MP in the figure) (without BTNPs) using the patch-clamp and calcium imaging. Supplementary Fig. 5 shows individual membrane potential traces and Supplementary Fig. 14 shows calcium imaging results of primary neurons without any particles and with magnetic Janus particles. These two experiments indicate no significant response in primary neurons due to FUS stimulation and magnetic Janus microparticles without any barium titanate nanoparticles. The following figures were added to the supplementary information and corresponding experimental procedures were added to the methods section.

Supplementary Fig. 6 Control experiments on primary neurons under LIFU with and without magnetic Janus particles. **a** Representative membrane potential traces of primary neurons excited under LIFU (two pulses of 5 ms $100 \text{ mW}\cdot\text{cm}^{-2}$ FUS) with and without magnetic Janus microparticles.

Supplementary Fig. 7 Control experiments on primary neurons under LIFU with and without magnetic Janus particles. Calcium responses of primary neurons without any particles, with magnetic Janus particles, and with PEMPs under 50 ms, 50, and 100 $\text{mW}\cdot\text{cm}^{-2}$ FUS pulses ($n = 4$ independent experiments, each dot represents the mean of 10 individual neuron responses).

5) The authors apply a 20 ms pulse and show an exemplary AP firing in Fig S2, but there is already steady increase in the resting membrane potential before the stimulation. Can authors clarify why there is this increase (almost 7 mV in 60ms)?

Response: We thank the reviewer for their comments. During our experiments, we occasionally observed oscillations in the membrane potential and highlighted by the reviewer's comment, we would like to investigate more about the depolarization and latency of action potential generation upon application of FUS on PEMP in the extracellular medium. To do that, we recorded primary neuron membrane potentials (Supplementary Fig. 6) and plotted all individual traces for each stimulation cycle on the same neuron, while it was a single representative trace in the original submission. For each trial, we synchronized the FUS driving signal with the patch-clamp recording system. To better differentiate the long oscillations in the membrane potential (10s of milliseconds) from the PEMP-induced depolarizations, we applied 50 mW.cm^{-2} , brief 5 ms FUS pulses. In addition, PEMP was closely located near the recorded neuron $<5 \mu\text{m}$ to observe clearer depolarizations. We also averaged the individual traces and plotted the mean signal (dashed black line) for better visualization. The mean latency and induced depolarization were calculated as 7.16 ms and $\sim 13.8 \text{ mV}$, respectively.

Supplementary Fig. 6. Membrane potential change during PEMP-induced stimulation. Individual patch-clamp electrophysiology recording of primary neurons excited with 50 mW.cm^{-2} , 5 ms, 2 MHz LIFU while PEMP was located near the recorded neuron $<5 \mu\text{m}$ ($n = 29$ pulses). The mean latency was calculated as $\sim 7.16 \text{ ms}$.

6) Figure 2e shows that for 50 ms US activation, the neuron still continues fires 2-3 times more after switching off the US-activation. This is an interesting result. What is the reason of this?

Response: We thank the reviewer for their comments. We attribute this behavior to the charging effect in the extracellular medium near the recorded neuron. Particularly, during the electrochemical characterization of a single PEMP (Supplementary Fig. 4c), we observed sustained current generation, particularly under $>20 \text{ mW.cm}^{-2}$ FUS. The sustained current was monophasic and had faradaic components⁴⁹, which was a substantial contributor to the current injection. As we observe the continuing firing after the switch off the FUS for $>20 \text{ ms}$ FUS pulses, this repetitive (tonic) spike

responses could be attributed to the faradaic charge transfer, which could have lasting effects even after the switch off⁴⁹. While this effect has not been studied in the literature on primary neurons and piezoelectric stimulation, there are examples of this tonic or burst behavior in different neural cells, including retinal ganglion cells (RGCs)⁵⁰, human induced pluripotent stem cells (hiPSCs)⁵¹, hippocampal⁵², and dorsal root ganglion (DRG) neurons⁵³ depending on the duration of the irradiation or excitation to different stimuli. This tonic spike induction could be explored in further studies and could be utilized in the stimulation of neural populations.

7) Does the PEPS movement damage the neuronal environment (cell soma or neurite) and does moving the particles effect the cells excitability?

Response: We thank the reviewer for this important consideration. We added the following calculation and discussion as Supplementary Note 2.

“To quantify the induced force generated by the steering and locomotion of a single PEMP, we utilized the basic force balance of a microroller based on previous reports^{13,14,54}. When a microroller moves on a planar surface, it creates a propulsion force to its translational direction, (F_P), and gravitational force (F_G) on the bottom wall^{13,14,54}. If all the movement and forces were considered, the basic force balance of a microroller for F_P and F_G could be drawn as in Fig. R1.

Fig. R1. Force balance on a microroller.

The F_P and F_G are expressed as:

$$F_P = \pi\mu a^2 \omega \left(\frac{4}{5} \ln \frac{a}{\delta} - 1.516 \right) \quad (1)$$

$$F_G = \frac{4}{3} \pi (\rho_p - \rho_f) a^3 g \quad (2)$$

where μ dynamic viscosity of the fluid, a is the particle radius, ω is the angular velocity, δ is lubrication or separation distance, ρ_p the density of the particle, ρ_f the density of the fluid, and g is the gravitational constant. For a 20 μm microroller, which is our case for the PEMPs, the F_G forces were calculated as ~ 46.73 pN, and F_P for different lubrication distances were approximated as follows^{14,55}:

Fig. R2. F_P produced by 20 μm microroller for different lubrication distances.

As a result, both the propulsion and gravitational forces are in the order of pN. With an exaggerated assumption of contact area is $1 \mu\text{m}^2$, which would be much higher in the real case, the momentary stress applied by the microroller would be $<50 \text{ Pa}$. In comparison, mechanical neural stimulation requires threshold force $>200 \text{ nN}$ and pressure value $>5\text{kPa}$ via direct contact with atomic force microscopy cantilever⁵⁶. Therefore, it is reasonable to assume that the movement of microrollers on biological cells would not induce potentially harmful effects or induce changes in the excitability of the neurons.”

We also added the corresponding discussion to the manuscript:

“In addition, we calculated the momentarily stress on the cells induced by the movement of the PEMP on the cellular layer⁵⁵ (Supplementary Note 2). The calculations revealed that the forces due to propulsion of the PEMP and gravitation were $<50 \text{ pN}$, and corresponding pressure of $<50 \text{ Pa}$ with the assumption of 1 mm^2 contact area between the PEMP and neurons. Resulting force and pressure values are order of magnitude smaller than the thresholds found in the literature⁵⁶.”

Moreover, in our previous study⁵⁵, we have tested the effect of microparticle rolling on the cells and the cell viability for 50 μm -in-diameter Janus microparticles and did not observe any significant effect. We also did not observe any significant change in the neural excitability after the PEMP movement near the target neuron, and even after the rotational movement shown in Supplementary Video 1 and 2, respectively.

8) In Fig 4b, authors show the patch clamp results depending on distance from the particles. Here some of the neurons experiences strong depolarization, while others do not because of the distance. If the neurons ids and distances shown in Fig 4a and b are correlated, why does 1, 2, 4, and 8 does not experience any depolarization, while number 12 is having strong firing success?

Response: We thank the reviewer for their comments. The neuron IDs in Fig 4a and b are correlated and the piezoelectric part of PEMPs faced down towards the neurons for this experiment. The neurons 1, 2, 4, and 8 did not experience any depolarization for the PEMP in the upper side of Fig 4a and effective stimulation distance matches with the distance vs stimulation success results in Fig 4d. The PEMP in the bottom side of Fig 4a induced depolarization for neurons 7 and 12 but not for 10 and 11. While this is expected for neuron 11, which is located further than the effective stimulation distance, neuron 10 did not show depolarization although it is close to the stimulation distance. This might be

due to a specific case for neuron 10, which may have lower electrical activity, or due to the potential electric field screening because of neuron 7. As neuron 7 is in between neuron 10 and the PEMP, it could screen the electric field and reduce the effective field seen by neuron 10. On the other hand, for neuron 12 we observed a significant response to the stimulation. We believe that a small deviation in the magnetization direction could cause a misorientation towards neuron 12, leading to unequal electric field distribution of the bottom PEMP. This was actually the reason why we conducted additional experiments under different PEMP orientations and collected more data to obtain the statistical analysis for orientation-dependent neural stimulation performance. While Fig 4a and 4b represent one example case, the statistical analysis with increased data set in Fig 4d, e, and f, provides a better understanding of effective stimulation distance.

Other points:

- When explaining figure 2f, the success rate is below 70% and 60%, while it is written as 83% and 61% for 100 Hz and 200 Hz, respectively.

Response: We thank the reviewer for their comments. We corrected the text as 69% and 59% for 100 Hz and 200 Hz, respectively.

- In line 231, missing closing parentheses.

Response: We corrected the parentheses.

- Fig 4a, neuron 9 is not indicated.

Response: We thank the reviewer for their comments. We indicated neuron 9 in Fig 4b, also in Fig 4a, and revised the figure.

- Page 9, a small typo: biocompativle

Response: We corrected the typo.

Reviewer #4

The manuscript submitted by Sitti and coworkers describes the use of piezoelectric microparticles that can be targeted at the single-cell level and transduce focused ultrasound to stimulate action potentials excitable cells (eg hippocampal neurons) in vitro. The particles additionally have a magnetic coating which allows the use of an externally applied magnetic field to manipulate the position of the particles. In general, I find the approach interesting and clever, however, there are, in my estimation, major technical flaws in the experiments and the interpretation. In its present form, this paper falls into the category of one of many where 1) nano/microparticles added to in vitro cell culture 2) energy is applied to the system 3) cells somehow fire action potentials.

Response: We thank the reviewer for their comments. We appreciate the reviewer's critical and valuable comments which have certainly improved the scientific rigor of the study with additional control experiments. We have performed a series of in vitro experiments and calculations to address the reviewer's concerns.

Technical problems:

- 1) The paper claims that the piezoactuated particles stimulate the cells electrically, via action upon voltage-gated channels and subsequent cellular membrane depolarization. The data do not support,

and rather contradict this conclusion. There is no evidence for extracellular electrical stimulation, for the following reasons:

1.1 Electrical depolarization of the cell by an extracellular microelectrode results in a clear stimulation artefact in the current-clamp trace of the intracellular recording. The magnitude of this stimulation artefact is of similar magnitude as the action potential signal, or often much larger than it. As can be seen in Fig 2b, e and very clearly in Fig S2, there is NO stimulation artefact corresponding to a capacitive current resulting from the stimulation being turned on and off. In Fig S2, a small and slow depolarization is observable during the ultrasound being on, however the magnitude of this depolarization is very similar to that shown later in the trace between 160 – 200 ms. Actual electrical coupling to the cell membrane of a magnitude that would result in AP generation would be apparent in the current or voltage-clamp trace. The lack of any electrical artefact here discounts an electrical stimulation mechanism.

Response: We thank the reviewer for their comments. We would like to answer the comments from several different perspectives and with additional experiments. Firstly, while the reviewer's comment on stimulation artefacts is valid and inevitable for direct electrical stimulation using metallic biointerfaces, dominantly capacitive silicon, and organic polymer-based optoelectronic biointerfaces, the artefact could be minimal or not traceable at all for plasmonic, dominantly faradaic neural stimulation mechanisms. Moreover, even in the case of full capacitive biointerfaces, recordings could be made without strong and fast stimulation artefacts depending on the device characteristics and filtering during the amplification and digitization of the recorded signal. There are several studies^{48,57-61} with no significant stimulation artefacts owing to the aforementioned methods. Secondly, the stimulatory waveform is different for US-based piezoelectric neural stimulation studies. In direct electrical and optoelectronic stimulation studies, researchers utilize square wave signals with fast rise and fall times (<10 ns in general and ~5.4 ns for our function generator), which generate spike-like artefacts due to either the signal response of recording systems and/or due to the electrochemical response of the biointerfaces. While in US-based piezoelectric stimulation, several previous studies and we utilize sinusoidal driving signals. Although the sinusoidal driving signal is modulated with lower frequency square wave signals^{35,62}, the sinusoidal nature of the driving signal also reduces and/or eliminates the artefacts. For instance, in our study, FUS transducer is driven via 2 MHz sinusoidal signal modulated with square waves in 0.5-200 Hz. Moreover, we utilize modified square waves with longer rise and fall times of ~10 μ s, respectively (Supplementary Fig. 12a) due to two main reasons. The first is to maintain the whole-cell patch-clamp conditions despite the ultrasound pressure waves, while the second is to eliminate the instantaneous high-pressure field during the spike-like rise period of the US. The latter is also beneficial for reducing any electrical or mechanical noise within the recording chamber. We added this figure for the beginning and the end of the driving square wave signal to properly explain this to the reader. Moreover, neural responses were amplified in current clamp mode using a 1 kHz low-pass Bessel filter (80 dB/decade), low-pass filtered at 10 kHz, and digitized at 50 kHz. We believe that these strategies also eliminate the stimulation artefact.

We followed the method in previously described photo-⁴⁷ and sono-electrochemical⁴⁸ measurement procedures for single particles. We first prepared a very low concentration of PEMP in live cell imaging solution put single particles inside of pulled glass patch pipettes, and mounted them onto the patch-clamp system (Supplementary Fig. 4a). The glass patch pipettes had a tip diameter of <2 μ m. The measurement pipette was positioned on the bottom surface of the recording chamber to its position at in vitro experiments on primary neurons. The system voltage is held at zero in voltage-clamp mode to generate the virtual ground in the system. We used the same filtering settings used in the patch-clamp electrophysiology experiments mentioned in the methods section. When the FUS

pulses were applied to the recording chamber without any particles, we did not observe any current deviation from the baseline signal (Supplementary Fig. 4b). Later, we enclosed a single PEMP as described above and applied ultrasound pulse intensities of 10, 20, 50, and 100 $\text{mW}\cdot\text{cm}^{-2}$ FUS while keeping the excitation period at 10 ms (Supplementary Fig. 4c). Although piezoelectric currents are in sinusoidal form for sinusoidal pressure transients, we observed a rising period and sustained current generation, particularly under $>20 \text{ mW}\cdot\text{cm}^{-2}$ FUS. Following the end of the FUS pulse, the generated current decays and reaches the baseline before the stimulation pulse. While this experiment proves the piezoelectric current generation of single PEMPs via BTNPs on its surface, it is hard to comment on the exact electrochemical process for the piezoelectric current generation. The recorded current waveforms show a monophasic behavior and no negative polarity after the FUS excitation. Therefore, we can hypothesize that the piezoelectric current generation suggests faradaic processes, which could be due to the electron transfer between BTNPs and the ionic extracellular medium. As we did not observe any current generation under FUS without PEMPs and observed an increase in the sustained current levels with increasing FUS intensity, we could claim that recorded current transients represent piezoelectric current generation. It is also important to note that, current transients may not fully represent the actual piezoelectric currents in the ionic media since the low amplitude current peaks might be screened in the extracellular medium and fast current peaks will be inevitably filtered out via low pass filters during signal amplification. Additionally, we may claim that the faradaic nature of the piezoelectric current generation with $>1 \text{ ms}$ rise time does not induce significant artefacts during neural recording since the faradaic process does not generate immediate and capacitive high peak artefact.

Supplementary Fig. 4. Single PEMP recordings to investigate piezoelectric current generation under FUS. **a** Experimental schematic of piezoelectric current measurement system. **b** Current trace from glass patch pipette with no PEMPs under two pulses of 10 ms $100 \text{ mW}\cdot\text{cm}^{-2}$ FUS. This is a representative trace from $n = 18$ measured traces. Blue rectangles represent the time points of FUS excitation. **c** Piezoelectric current traces from a single PEMP under 10 ms 10, 20, 50, and $100 \text{ mW}\cdot\text{cm}^{-2}$ FUS. The potential was held at zero in voltage-clamp mode.

We added the following text to the main manuscript to discuss the reviewer's comments:

“Estimation of single PEMP behavior under LIFU

We started our investigations by measuring the single PEMP response in an interconnect-free configuration^{47,48} using a patch clamp system under LIFU. To accomplish that, we built a measurement system consisting of a patch-clamp amplifier system integrated into an upright fluorescence microscope, a water tank, and a focused ultrasound probe with a 2 MHz center frequency (Supplementary Fig. 3). We first prepared a very low concentration of PEMPs in live cell imaging solution put single particles inside of pulled glass patch pipettes, and mounted them onto the patch-clamp system (Supplementary Fig. 4a). The measurement pipette was positioned on the bottom surface of the recording chamber to its position to be used in vitro experiments on primary neurons. The system voltage is held at zero in voltage-clamp mode to generate the virtual ground in the system. When the FUS pulses were applied to the recording chamber without any particles, we did not observe any current deviation from the baseline signal (Supplementary Fig. 4b). Later, a single PEMP was enclosed and LIFU intensity of 10, 20, 50, and 100 mW.cm⁻² was applied while keeping the excitation period at 10 ms (Supplementary Fig. 4c). We observed unipolar piezo-electrochemical current with a rising period and sustained current generation, particularly under LIFU intensity of >20 mW.cm⁻². Following the end of the LIFU pulse, the generated current decays and reaches the baseline before the stimulation pulse. Therefore, we could hypothesize that the piezoelectric current generation suggests Faradaic processes⁴⁹, which could be due to the electron transfer between BTNPs and the ionic extracellular medium. As no current generation under LIFU without PEMPs was observed and the sustained current levels increased with higher LIFU intensity, we could claim that recorded current transients represent piezoelectric current generation. In addition, current transients may not fully represent the actual piezoelectric currents in the ionic media since the low amplitude current peaks might be screened in the extracellular medium and fast current peaks will be inevitably filtered out due to the system bandwidth and via low pass filters during signal amplification. Once the piezoelectric current generation of PEMPs was explored, we proceeded to electrophysiology experiments to evaluate the neural modulation potential of PEMPs.”

1.2 Latency is inconsistent with electrical stimulation. The paper reports latency to AP generation between 25 ms to 10 ms, with shorter times with higher FUS intensity. Electrical actuation of the cell membrane would have to be within a few ms, after this time electrical field is screened by the electrolyte and the cell membrane does not “feel” any potential difference from an extracellular particle/electrode. This is why extracellular neurostimulation pulses last between a few hundred microseconds, and maximum a few ms. A latency of 25 ms is not consistent with electrical stimulation, and the notion of “charge buildup” outside of the cell over such time is, as described already, not possible, due to the kinetics of the electric field screening upon formation of capacitive double layers.

Response: We thank the reviewer for their comments. We agree with the reviewer that the mechanism of stimulation and its nature should be elaborated comprehensively. If we consider the literature, there are several studies in optoelectronic biointerfaces^{53,57,58,61,63–65} with stimulation latencies higher than a few ms of latency found in direct electrical stimulation and generally 10s of ms for biointerfaces converting one excitation source to electric charges. Particularly, biointerfaces that show capacitive electrochemical behavior under external stimuli induce neural activation with low

latency values close to values found in direct-contact extracellular electrical stimulation or intracellular stimulation. On the other hand, biointerfaces with mixed or dominantly faradaic electrochemical responses resulted in relatively higher latency values during neural stimulation due to slower kinetics of ion exchange and faradaic reactions^{66,67}. This difference could also be explained by comparing the current profile of a capacitive versus faradaic biointerface. For capacitive biointerfaces, there is a fast peak response and a following exponential decay in the current waveform during the onset of the excitation pulse. However, for faradaic processes, the current slowly reaches its final amplitude. As the total charge generated during the excitation pulse is the integral of the current waveform, the injected charge is lower during the onset of the stimulation for faradaic processes. If we define a charge threshold for sufficient depolarization for neural stimulation, faradaic processes slowly generate the same amount of charge than the capacitive biointerfaces. Therefore, it is expected to have long latencies for faradaic biointerfaces and it has been observed and studied in the literature^{53,57,58,61,63–65}.

The second aspect to consider is the excitation waveform. In most of the studies with developed biointerfaces, square waveforms have been utilized. Square stimulation waveforms have very fast rise and fall times, which induces a fast and strong current generation via proposed biointerfaces. On the other hand, in our study, the FUS transducer was driven via a 2 MHz sinusoidal signal modulated with square waves in 0.5–200 Hz. Moreover, we utilize modified square waves with longer rise and fall times of $\sim 10 \mu\text{s}$ (in comparison, ordinary square waves have fast rise and fall times $< 10 \text{ ns}$ in general) (Supplementary Fig. 12a) for two main reasons. The first is to maintain the whole-cell patch-clamp conditions despite the ultrasound pressure waves, while the second is to eliminate the instantaneous high-pressure field during the spike-like rise period of the US. Additionally, sinusoidal electrical stimulation itself generates late neural responses due to slow charging times. For instance, we numerically simulated sinusoidal electrical stimulation and extracted corresponding membrane potential transients using the Hodgkin–Huxley (HH) model. Fig. R3 and Fig. R4 show the frequency-dependent trend for neural stimulation for the same amplitude of sinusoidal excitations. While the frequency increases, the latency of the first action potential decreases since the sinusoidal pulse generates faster charging on the neural membrane^{68,69}. Considering the capacitive-faradaic charge generation process and the waveform of the excitation, we hypothesize that it is expected to have a longer latency than direct electrical stimulation.

Moreover, the latency depends on the excitation source intensity (such as the laser/LED power) and the distance between the stimulation source and the neuron. For instance, the work by Bezanilla and Tian et al.⁴⁷ showed the dependency between the excitation power and duration to elicit action potentials (Fig. 4c of the mentioned article). Their lowest laser power of interest to excite their material elicits action potentials in $\sim 9 \text{ ms}$. They have also found that decreasing excitation power increases the latency since the energy to induce threshold depolarization is delivered in a longer time for low excitation powers^{47,70}. Similar to these findings, we observed reduced latency due to increased FUS intensity, which increases the piezoelectric charge generation within shorter time windows (Fig 2c and Supplementary Fig. 4). On the other hand, considering the reviewer's comment and having realized a mismatch in the original submission between Fig. 2b and Fig. 2c (latency values were higher in Fig. 2c than its real value seen in the membrane potential traces in Fig. 2b), we again analyzed all of our patch-clamp data. The latency values in Fig. 2c were not always in agreement with the traces in Fig. 2b. We realized the problem due to our event detection template search in the analysis program Clampfit. During the patch-clamp experiment, we synchronized the patch-clamp digitizer with the signal generator for ultrasound excitation. This event detection template search considers the beginning time of 1st pulse generated by the signal generator and generates the time points automatically. However, we realized that the time frames of generated pulse time points did not

match with the electrophysiology data. As a result, for higher FUS intensities the ON time of the US pulses and corresponding neural response mix with each other since for the higher intensities action potentials are not single but in the form of spike trains.

To resolve this problem, we manually analyzed the data on MATLAB by using the recorded ultrasound and patch-clamp signals, without using Clampfit's automatic search algorithm. The corrected latency plot below (Fig 2c) indicates that there is no significant difference from the previous calculation for 10 and 20 $\text{mW}\cdot\text{cm}^{-2}$ FUS intensities but the correct latency is lower than the previous calculation since the spike trains induced in these intensities falsified the automatic algorithm. We thank the reviewer for highlighting the importance of latency and would like to answer the reviewer's comment based on the correct latency data. We observed the reduction in latency while we increased the FUS intensity (Fig. 2c), down to ~ 10.6 ms and ~ 7.4 ms for 50 and 200 $\text{mW}\cdot\text{cm}^{-2}$ intensity, respectively. While the latency values are still higher than the direct electrical stimulation, latency reduces once we increase the intensity to >200 $\text{mW}\cdot\text{cm}^{-2}$. The reduction in latency as a function of FUS intensity could be explained by the increase in the amount of charge injection, which is explored in Supplementary Fig. 4c.

In conclusion, our hypothesis is based on a piezoelectric faradaic charge-injection mechanism. Firstly, the transient positive and negative pressure field generated by the US is never fully symmetric²³ (also observed in Supplementary Fig. 11d). This biased sinusoidal pressure wave results in an asymmetric charge generation in each sinusoidal cycle. The electrochemical characterization of single PEMP (Supplementary Fig. 4c) supports this claim since the current waveform is unipolar and not symmetric. Therefore, what we intend to mean by charge build-up is this phenomenon of asymmetric charge generation in each pressure cycle and its macroscopic effect in thousands of cycles. Although several studies have utilized piezoelectric nanoparticles for neural stimulation, particularly the pioneering works of Marino, Suzuki, and Ciofani¹¹ with BTNPs indicating activation of voltage-sensitive channels, there are no benchmark values for latency or patch-clamp data for our comparison with the literature.

Fig. 2c. The mean latency of the action potential peaks and the jitter as the standard deviation of latencies of all measured neurons under changing FUS excitation intensities ($n = 20$).

Fig. R3. Numerical simulation of membrane potential change in response to sinusoidal electrical stimulation at 1, 2, and 5 Hz based on the Hodgkin–Huxley (HH) model. Each plot for a specific frequency shows the applied sinusoidal electrical current in the top part and the corresponding membrane potential transient in the bottom part.

Fig. R3. Numerical simulation of membrane potential change in response to sinusoidal electrical stimulation at 10, 20, 50, and 100 Hz based on the Hodgkin–Huxley (HH) model. Each plot for a specific frequency shows the applied sinusoidal electrical current in the top part and the corresponding membrane potential transient in the bottom part.

1.3. The geometry of the particles is not well-suited to extracellular electrical stimulation. In extracellular electrical stimulation, it is important to consider cathode/anode, or so-called primary and return electrode – basically from where the current is flowing and to where it is going. There has to be a potential drop across the cell membrane, included by current flowing around the cell. As a reference, consider papers such as 1. Schoen, I. & Fromherz, P. The mechanism of extracellular stimulation of nerve cells on an electrolyte-oxide-semiconductor capacitor. *Biophys. J.* 92, 1096–1111 (2007); 1. Boinagrov, D., Loudin, J. & Palanker, D. Strength–Duration Relationship for Extracellular Neural Stimulation: Numerical and Analytical Models. *J. Neurophysiol.* 104, 2236–2248 (2010). Here, each piezoelectric nanoparticle can charge on the surface, but where is the plus and minus? The charge and electric field are localized on one particle? Therefore, by Gauss Law the electric field is located inside of the particle, not outside of it. How can current flow around the cell and create differences in cell membrane potential, even locally? The COMSOL model assumed that the whole sphere as the Au-coated surface as one terminal. This would then make sense – but how can current asymmetrically from the NP into the gold? and more importantly, how can it happen at MHz frequency to generate any useful currents in solution? Which is the next point:

Response: We thank the reviewer for their comments. We believe this is a critical question for all micro/nanoparticle studies. Our PEMP design consists of three elements; silica microparticle of 20 μm diameter, barium titanate nanoparticles (BTNPs), and Ni/Au metallic thin film layer. The silica microparticle is a low-conductor by nature and builds the main base part of PEMPs. While the charge is generated by individual BTNPs on the half surface, the other half with Ni/Au thin film does not generate charge under ultrasound excitation. We also want to emphasize that BTNPs are not present on the Ni/Au thin film coating and are only conjugated on the half-silica surface. If a single BTNP on the silica surface is considered, one part of the nanoparticle is conjugated to a non-conductive material (silica microparticle), while the other parts are in contact with the conductive ionic media. Therefore, considering this symmetry-breaking conditions, we hypothesize that the piezoelectric charging effect on BTNPs induces electric field lines from BTNPs towards the Ni/Au coating, like an individual, virtual ground, since there is no other electrical ground present near the PEMPs. In the COMSOL model, we build the same architecture with a silica microparticle core, one terminal representing the BTNP layer on the half silica surface, and one floating terminal on the other half representing the Ni/Au coating. The only ground is defined at the distant boundaries of the simulation space for charge conservation, which is a cube with 200 μm edges. In addition, we numerically calculated the maximum potential difference on a single BTNP according to the electro-elastic model of Marino, Suzuki, and Ciofani et al.¹¹. If we consider the same calculation for BTNPs used in our study, we could calculate the maximum potential difference:

$$\varphi_R \equiv - \frac{R(s * e_{rr} + 2e_{r\theta})}{s * \epsilon_{rr}} * \frac{p_{US}}{s * \gamma + 2\alpha}$$

$$\varphi_R \equiv R * p_{US} * k,$$

$$\text{where } k \equiv -\frac{(s \cdot e_{rr} + 2e_{r\theta})}{s \cdot \epsilon_{rr}} * \frac{1}{s \cdot \gamma + 2\alpha} \approx 7.6 \cdot 10^{-3} \left(\frac{V \cdot m}{N}\right)$$

where $p_{US} = 40 \text{ kPa}$ and the other parameters such as the water density, sound speed in water¹², viscoelastic, and piezoelectric properties for BTNPs were found from the literature¹¹. If the low-intensity case is considered ($50 \text{ mW} \cdot \text{cm}^{-2}$ FUS intensity), the maximum potential difference on a single BTNP can be calculated as $\sim 0.042 \text{ mV} - 0.1412 \text{ mV}$ depending on the different dielectric constants found in the literature. Based on the estimation that half of the $20 \text{ }\mu\text{m}$ microparticle surface is conjugated with BTNPs, we consider that half of this amount ~ 1100 BTNPs could be found on the microparticle surface. This calculation is based on a simplified model and we acknowledge that all individual BTNP potentials on the surface cannot be directly summed up. On the other hand, the single BTNP potential generation of $\sim 0.042 \text{ mV} - 0.1412 \text{ mV}$ and >1000 BTNPs on the PEMP surface could provide insights that PEMPs could induce sufficient piezoelectric charge to evoke membrane depolarization, which was already observed in patch-clamp and calcium imaging experiments. Moreover, the piezoelectric current generation of a single PEMP was recorded and presented in Supplementary Fig. 4, which also indicates a detectable and distant current generation. The last part of the reviewer's question is answered in the following section 1.4.

1.4 As I understand the frequency of the FUS of 2 MHz will cause the piezoelectric particles to charge with this frequency. How can MHz frequency stimulation result in ionic currents to flow in solution and cause action upon voltage-gated ion channels? This frequency is way above the relaxation frequency of ions. One can make the argument that then there is an electric field that would not be screened, but again, electric field over what geometric space that effects the cell membrane?

Response: We thank the reviewer for their comments. Before diving into our hypothesis, we would like to mention that the PEMP design consists of three main parts: silica microparticle of $20 \text{ }\mu\text{m}$ diameter, barium titanate nanoparticles (BTNPs), and Ni/Au metallic thin film layer. The silica microparticle has been intentionally chosen as a non-conductive base for the PEMP design. While the half-silica surface is conjugated with barium titanate nanoparticles (BTNPs), the opposite half-surface is coated with a Ni/Au thin film. The main reason for such a design with a non-conductive inner part and two different surfaces is to break the electrical symmetry across the whole particle. During the ultrasound excitation state under 2 MHz FUS, BTNPs piezoelectrically charge at 2 MHz . However, we agree with the reviewer that 2 MHz is the frequency of ultrasound excitation of piezoelectric nanoparticles but not the frequency of ionic flux in the extracellular medium. During the excitation, the charge transfer should be towards the Au layer since it is more conductive than any other neighboring material in the electrolyte. This creates an asymmetric charge distribution on the PEMP colloid. The resulting effect is the flux of ion and ionic currents away from the PEMP surface. Therefore, we agree with the reviewer that ionic currents occur at a lower frequency than 2 MHz ultrasound frequency due to limited ion kinetics. Our hypothesis is based on the asymmetry of the electrochemically active surfaces on PEMP, where the electric field lines occur between the BTNP conjugated surface and Ni/Au coated surface without any electron transfer within the silica core of the PEMPs. Additionally, the orientation-dependent neural stimulation experiments (Fig. 4d, e, and f) support our hypothesis on the electric field generated between the BTNP conjugated surface and Ni/Au coated surface. The neural stimulation performance difference between the top and bottom configuration (Fig. 4d and e) indicates higher electric field intensity on the BTNP-conjugated surface. On the other hand, the asymmetry in the neural stimulation profile in Fig. 4f also shows the aforementioned effect.

In addition, if we consider the electrochemical characterization of a single PEMP (Supplementary Fig. 4), we could compare these results with the particular example work by R. Parameswaran et al.⁴⁷, where they investigated the photo-electrochemical effect of the coaxial p-type/intrinsic/n-type (PIN) Si nanowires. They positioned the Si nanowire $\sim 10\text{--}30\ \mu\text{m}$ from the tip of the measurement pipettes and measured currents in the order of pA. While measurement at a distance larger from the Debye length of the corresponding system (in the order of nm) is also counter-intuitive due to the screening in electrolyte, their mechanism also depends on the faradaic reactions so that they can measure the current at further distances due to ionic reactions within the extracellular medium. We believe this is the most similar case to our system, while the materials and excitation mechanisms are different, and further proves the validity of our measurements and supports our hypothesis on piezoelectric neural stimulation. Therefore, the reviewer's questions and comments substantially improved our understanding of the piezo-electrochemical process and we conducted all of the analysis within our capabilities; however further analysis of the electrochemical processes requires in-depth investigations, which is beyond the scope of this study and will be the subject of a separate study. We combined the information obtained through answering this question and started evaluating the mechanism behind the neural stimulation in the next question, 2.

Supplementary Fig. 4. Single PEMP recordings to investigate piezoelectric current generation under FUS. **a** Experimental schematic of piezoelectric current measurement system. **b** Current trace from glass patch pipette with no PEMP under two pulses of 10 ms $100\ \text{mW}\cdot\text{cm}^{-2}$ FUS. This is a representative trace from $n = 18$ measured traces. Blue rectangles represent the time points of FUS excitation. **c** Piezoelectric current traces from a single PEMP under 10 ms 10, 20, 50, and $100\ \text{mW}\cdot\text{cm}^{-2}$ FUS. The potential was held at zero in voltage-clamp mode.

We added the following text to the main manuscript to discuss the reviewer's comments:

“Estimation of single PEMP behavior under LIFU

We started our investigations by measuring the single PEMP response in an interconnect-free configuration^{47,48} using a patch clamp system under LIFU. To accomplish that, we built a measurement system consisting of a patch-clamp amplifier system integrated into an upright fluorescence microscope, a water tank, and a focused

ultrasound probe with a 2 MHz center frequency (Supplementary Fig. 3). We first prepared a very low concentration of PEMP in live cell imaging solution put single particles inside of pulled glass patch pipettes, and mounted them onto the patch-clamp system (Supplementary Fig. 4a). The measurement pipette was positioned on the bottom surface of the recording chamber to its position to be used in vitro experiments on primary neurons. The system voltage is held at zero in voltage-clamp mode to generate the virtual ground in the system. When the FUS pulses were applied to the recording chamber without any particles, we did not observe any current deviation from the baseline signal (Supplementary Fig. 4b). Later, a single PEMP was enclosed and LIFU intensity of 10, 20, 50, and 100 mW.cm⁻² was applied while keeping the excitation period at 10 ms (Supplementary Fig. 4c). We observed unipolar piezo-electrochemical current with a rising period and sustained current generation, particularly under LIFU intensity of >20 mW.cm⁻². Following the end of the LIFU pulse, the generated current decays and reaches the baseline before the stimulation pulse. Therefore, we could hypothesize that the piezoelectric current generation suggests Faradaic processes⁴⁹, which could be due to the electron transfer between BTNPs and the ionic extracellular medium. As no current generation under LIFU without PEMP was observed and the sustained current levels increased with higher LIFU intensity, we could claim that recorded current transients represent piezoelectric current generation. In addition, current transients may not fully represent the actual piezoelectric currents in the ionic media since the low amplitude current peaks might be screened in the extracellular medium and fast current peaks will be inevitably filtered out due to the system bandwidth and via low pass filters during signal amplification. Once the piezoelectric current generation of PEMP was explored, we proceeded to electrophysiology experiments to evaluate the neural modulation potential of PEMP.”

2) There is no control measurement / sample for FUS alone, without particles. This is a major oversight. The authors must compare the patch clamp measurements in the absence of particles in order to claim that the particles are doing anything. The only control sample is for the ROS analysis and viability tests, however there is no control for FUS stimulation. How can the authors exclude that it is not FUS alone that is stimulating cell action potential, as is routinely reported in many other studies? The generation of action potentials appears to be via a thermal mechanism, consistent with the slow change in membrane potential. This is reminiscent of several studies on using laser light and various nano/microparticles to stimulate cells, where a photothermal mechanism is established. Consider the mechanism rationalization guidelines on page 12 of this protocol paper: Nongenetic optical neuromodulation with silicon-based materials. Nat. Protoc. 14, 1339–1376 (2019); or Fig 2 of this paper: Rational design of silicon structures for optically controlled multiscale biointerfaces. Nat. Biomed. Eng. 2, 508–521 (2018). Moreover, the actual effects of the functional particles in this paper is not possible to determine considering the lack of a control experiment with FUS alone, which is of course well-known to elicit APs in excitable cells.

Response: We thank the reviewer for their comments.

We utilized the suggested mechanism rationalization guidelines²⁵. We started our investigations by evaluating possible thermal effects due to FUS. Firstly, we modeled our FUS transducer in COMSOL Multiphysics® v6.1 software (COMSOL AB, Sweden) by the recorded pressure waves using a needle hydrophone. To realize this, we modeled a water and tissue domain in COMSOL. We utilize Pressure Acoustics in the frequency domain to obtain the acoustic pressure field (Supplementary Fig. 11a, b, and c) and integrated Bioheat Transfer module to calculate the heating and cooling of the tissue phantom via Pennes’ bioheat equation. Thermal simulations were performed in a two-fold process

corresponding to a worst-case scenario, propagation in a water medium, and thermal absorption in a brain-mimicking medium. The following parameters²³ were followed for the propagation medium (water): sound speed, $c = 1,500 \text{ m s}^{-1}$; volumetric mass, $\rho = 1,000 \text{ kg m}^{-3}$; nonlinearity coefficient, $B/A = 5$; attenuation coefficient, $\alpha = 2.2 \times 10^{-3} \text{ dB cm}^{-1} \text{ MHz}^{-\gamma}$; frequency power law of the attenuation coefficient, $\gamma = 2$. COMSOL simulations were calibrated by adjusting the input pressure to match the pressure at the focus measured in the water tank by the hydrophone (Supplementary Fig. 11d, e, and f). In the second part of the simulation, we utilized the heat transfer module in the tissue domain with the parameters: brain volumetric mass $\rho_{\text{brain}} = 1046 \text{ kg m}^{-3}$, the brain sound speed $c_{\text{brain}} = 154 \text{ s}^{-1}$, K_t is the brain thermal conductivity ($0.51 \text{ W m}^{-1} \text{ }^\circ\text{C}^{-1}$) with the initial brain temperature $T_0 = 37 \text{ }^\circ\text{C}$ ²³. Once we have the heat generation due to acoustic wave absorption in the tissue domain, we can simulate the transient heating/cooling cycles due to 0.5-200 Hz modulated 2 MHz FUS excitation. Supplementary Fig. 12 presents the simulation and experimental heating investigations. First, we utilized single US pulses with 10-1000 ms of 2 MHz pressure waves and please note the rise/fall regions of the pulse (Supplementary Fig. 12a). We investigated the single pulse heating (Supplementary Fig. 12b) and utilized this information to simulate 0.5-50 Hz stimulation for 10 seconds (Supplementary Fig. 12c). The results indicate that increasing the stimulation frequency decreases the overall heating and even for the lowest frequency we did not expect $>0.009 \text{ }^\circ\text{C}$ heating during 10-sec stimulation. To verify our simulations, we carried out local temperature measurements^{24,25} by patch pipette resistance changes. First, we obtained the resistance-temperature calibration (Supplementary Fig. 12d). Then, we positioned the same patch pipette to the focal point of the FUS at the bottom of the recording chamber. The same pulse and frequency-dependent analysis was carried out and compared with the simulation results (Supplementary Fig. 12e and f). Simulations were well-matched with the experimental temperature measurements and indicate that the FUS intensity used in this study does not generate significant local heating and is much lower than the reported temperature changes for thermal activation of neurons^{24,26-28}.

Supplementary Fig. 11. Experimental and numerical investigation of pressure waves generated by the FUS transducer. **a** 2D representation of ultrasound transducer/water/tissue phantom system. Model parts are marked in the figure. The model possesses the axial symmetry that enables three-dimensional reconstruction of the 2D computations. Numerical results and corresponding heat maps of **b** absolute acoustic pressure in xy plane at the center of the FUS focus and **c** acoustic

intensity magnitude in zr plane. **d** Experimental and simulation results of transient pressure waves for a single burst pulse of FUS at 2 MHz. Acoustic pressure profile along **e** the symmetry axis and **f** radial direction in the focal plane.

Supplementary Fig. 8 Evaluating thermal effects by numerical simulations and resistance-temperature measurements. **a** FUS driving signal for temperature measurements. The driving signal is at 2 MHz modulated with a single pulse with a rise and fall time of 10 ms. Inset shows the rise and fall time. **b** Heating calculations of brain phantom under continuous 100 mW.cm⁻² FUS for changing pulse durations. The inset shows 2 MHz heating/cooling cycles. **c** Heating calculations of brain phantom under 2 MHz 100 mW.cm⁻² FUS for changing pulse frequencies. **d** Calibration curves using a linear fit of recorded pipette resistances and corresponding extracellular medium temperatures. The linear fit shows the relationship between the pipette resistance and temperature. This calibration was utilized to measure the temperature increase under 100 mW.cm⁻² FUS for **e** changing pulse frequency and **f** duration. The bar plots in **e** and **f** demonstrate the numerical calculations and experimental results for $n = 7$ and $n = 5$ independent experiments for pulse frequency and duration, respectively.

Additionally, we operate in the diagnostic, nonthermal, noncavitational (<100 mW.cm⁻²) spatial peak temporal average intensity levels²⁹⁻³¹. Our excitation parameters are within the low-intensity US regime and lower than the observed and proposed intensities in the literature³² for thermal, cavitation, microtubule resonance, and mechanosensitive stimulation mechanisms in terms of the temporal and spatial average intensity of FUS. We calculated the intensity characteristics of FUS stimulus based on the standards developed by the National Electronics Manufacturers Association³³, The American Institute of Ultrasound Medicine, and the United States Food and Drug Administration (FDA), *Marketing Clearance of Diagnostic Ultrasound Systems and Transducers*. By utilizing the measurements recorded from the calibrated needle hydrophone, the pulse intensity integral (PII), the spatial-peak pulse-average intensity (I_{SPPA}), the spatial-peak, temporal-average intensity (I_{SPTA}), and the mechanical index were calculated as

$$PII = \int \frac{p^2(t)}{Z_0} dt$$

where p is the instantaneous peak pressure, Z_0 is the characteristic acoustic impedance in Pa s/m defined as ρc where ρ is the density of the medium, and c is the speed of sound in the medium. For the safety considerations, we used the brain tissue as the propagation and focus medium, where the brain volumetric mass $\rho_{\text{brain}} = 1,046 \text{ kg}\cdot\text{m}^{-3}$, and the brain sound speed $c_{\text{brain}} = 154 \text{ m}\cdot\text{s}^{-1}$.

$$I_{\text{SPPA}} = \frac{PII}{PD}$$

where PD is the pulse duration, and

$$I_{\text{SPTA}} = PII * PRF$$

where PRF is equal to the pulse repetition frequency in Hz. The mechanical index was defined as

$$MI = \frac{p_r}{\sqrt{f}}$$

Our calculations indicate spatial-peak temporal-average intensities (I_{SPTA}) of 3.8–28.7 $\text{mW}\cdot\text{cm}^{-2}$ for a total stimulus duration ranging between 2.5 and 500 ms. FUS waveforms had peak rarefactional pressures (p_r) of 0.014–0.108 MPa, pulse intensity integrals (PII) of 0.017–0.095 $\text{mJ}\cdot\text{cm}^{-2}$, and spatial-peak pulse-average intensities (I_{SPPA}) of 0.027–0.267 $\text{W}\cdot\text{cm}^{-2}$. The mechanical index (MI) was calculated as 0.028–0.0864 for the lowest and highest rarefactional pressures. These results indicate that FUS intensity values used in this study are not sufficient to evoke significant thermal or mechanical effects that might lead to stimulation of primary hippocampal neurons, especially >10 Hz neural stimulation^{32,34–37}. In another hypothesis, Krasovitski et al.²⁹ developed a model mechanism, called intramembrane cavitation, for ultrasound-induced neural stimulation at noncavitation low intensity regime. They claim and model that in these low intensities ultrasonically induced bilayer membrane motion, which does not require the preexistence of air voids in the tissue, could induce neural stimulation. In their extended model on neural cells in mouse motor cortex³⁸, they provide threshold intensity and ultrasound duration to generate a single action potential. If we compare the intensity and energy thresholds, PEMP could elicit action potentials at 50 Hz with a high success rate under an order of magnitude less intensity and about two orders of magnitude less energy than required for neural stimulation by direct ultrasound effects³⁸. Therefore, we may also rule out the proposed intramembrane cavitation effects, which supports our claim that PEMP under FUS excitation induce neural stimulation via piezoelectric charge generation. We added the calculation and discussion of these mechanical and thermal effects to the main text:

“For all neural stimulation experiments, we operate in the diagnostic, nonthermal, noncavitation (<100 $\text{mW}\cdot\text{cm}^{-2}$) spatial peak temporal average intensity levels^{29–31}. Moreover, the excitation parameters are within the low-intensity US regime and lower than the observed and proposed intensities in the literature³² for thermal, cavitation, microtubule resonance, and mechanosensitive stimulation mechanisms in terms of the temporal and spatial average intensity of FUS. To further evaluate these potential thermal and mechanical effects due to FUS excitation, we started our investigations by evaluating possible thermal effects due to FUS. Firstly, we modeled our FUS transducer in COMSOL Multiphysics by the recorded pressure waves using a hydrophone. To realize this, we modeled a water and tissue domain in COMSOL (Supplementary Fig 11a). By using the pressure acoustics (Supplementary Fig 11b and c) and bioheat transfer modules, the heating and cooling of the tissue phantom were calculated via Pennes’s bioheat equation. Thermal simulations were performed in a two-fold process corresponding to a worst-case scenario, propagation in a water medium, and thermal absorption in a brain-mimicking medium. The following parameters²³ were followed for the propagation medium (water): sound speed, $c = 1,500 \text{ m}\cdot\text{s}^{-1}$; volumetric mass, $\rho = 1,000 \text{ kg}\cdot\text{m}^{-3}$; nonlinearity coefficient, $B/A = 5$;

attenuation coefficient, $\alpha = 2.2 \times 10^{-3} \text{ dB cm}^{-1} \text{ MHz}^{-y}$; frequency power law of the attenuation coefficient, $y = 2$. COMSOL simulations were calibrated by adjusting the input pressure to match the pressure at the focus measured in the water tank by the hydrophone (Supplementary Fig 11d, e, and f). In the second part of the simulation, we utilized the heat transfer module in the tissue domain with the parameters: brain volumetric mass $\rho_{\text{brain}} = 1046 \text{ kg m}^{-3}$, the brain sound speed $c_{\text{brain}} = 154 \text{ s}^{-1}$, K_t is the brain thermal conductivity ($0.51 \text{ W m}^{-1} \text{ }^\circ\text{C}^{-1}$) with the initial brain temperature $T_0 = 37 \text{ }^\circ\text{C}$ ²³. Once we have the heat generation due to acoustic wave absorption in the tissue domain, we can simulate the transient heating/cooling cycles due to 0.5-200 Hz modulated 2 MHz FUS excitation. First, we utilized single US pulses with 10-1000 ms of 2 MHz pressure waves (Supplementary Fig 12a). We investigated the single pulse heating (Supplementary Fig 12b) and utilized this information to simulate 0.5-50 Hz stimulation for 10 seconds (Supplementary Fig 12c). The results indicate that increasing the stimulation frequency decreases the overall heating and even for the lowest frequency we did not expect $>0.009 \text{ }^\circ\text{C}$ heating during 10 sec stimulation. To verify our simulations, we carried out local temperature measurements^{24,25} by patch pipette resistance changes. First, we obtain the resistance-temperature calibration (Supplementary Fig 12d). Then, we positioned the same patch pipette to the focal point of the FUS at the bottom of the recording chamber. The same pulse and frequency-dependent analysis was carried out and compared with the simulation results (Supplementary Fig 12e and f). Simulations were well-matched with the experimental temperature measurements and indicate that the FUS intensity used in this study does not generate significant local heating and is much lower than the reported temperature changes for thermal activation of neurons^{24,26-28}.

We calculated the intensity characteristics of FUS stimulus based on the standards developed by the National Electronics Manufacturers Association³³, The American Institute of Ultrasound Medicine, and the United States Food and Drug Administration (FDA), Marketing Clearance of Diagnostic Ultrasound Systems and Transducers. By utilizing the measurements recorded from the calibrated hydrophone, a FEM model was built to simulate mechanical and thermal effects due to FUS (Supplementary Fig. 16). The pulse intensity integral (PII), the spatial-peak pulse-average intensity (I_{SPPA}), the spatial-peak, temporal-average intensity (I_{SPTA}), and the mechanical index were calculated as $\text{PII} = \int \frac{p^2(t)}{Z_0} dt$, where p is the instantaneous peak pressure, Z_0 is the characteristic acoustic impedance in Pa s/m defined as pc where ρ is the density of the medium, and c is the speed of sound in the medium. For the safety considerations, we used the brain tissue as the propagation and focus medium, where the brain volumetric mass $\rho_{\text{brain}} = 1,046 \text{ kg.m}^{-3}$, and the brain sound speed $c_{\text{brain}} = 154 \text{ m.s}^{-1}$. $I_{\text{SPPA}} = \frac{\text{PII}}{\text{PD}}$ where PD is the pulse duration, and $I_{\text{SPTA}} = \text{PII} * \text{PRF}$ where PRF is equal to the pulse repetition frequency in Hz. The mechanical index was defined as $\text{MI} = \frac{p_r}{\sqrt{f}}$. Our calculations indicate spatial-peak temporal-average intensities (I_{SPTA}) of 3.8–28.7 mW.cm^{-2} for a total stimulus duration ranging between 2.5 and 500 ms. FUS waveforms had peak rarefactional pressures (p_r) of 0.014–0.108 MPa, pulse intensity integrals (PII) of 0.017–0.095 mJ.cm^{-2} , and spatial-peak pulse-average intensities (I_{SPPA}) of 0.027–0.267 W.cm^{-2} . The mechanical index (MI) was calculated as 0.028-0.0864 for the lowest and highest rarefactional pressures. These results indicate that FUS intensity values used in this study are not sufficient to evoke significant thermal or mechanical effects that might lead to stimulation of primary hippocampal neurons, especially $>10 \text{ Hz}$ neural stimulation^{32,34–37}.

To eliminate any other mechanical effects induced by the PEMP, we calculated the momentarily stress on the cells induced by the movement of the PEMP on the cellular layer (Supplementary Note 2). The calculations revealed that the forces due to propulsion of the PEMP and gravitation were $<50 \text{ pN}$, and corresponding pressure of $<50 \text{ Pa}$ with the assumption of $1 \text{ } \mu\text{m}^2$ contact area between the

PEMP and neurons. The resulting force and pressure values are an order of magnitude smaller than the thresholds found in the literature⁵⁶. We added the following as a supplementary note:

“To quantify the induced force generated by the steering and locomotion of a single PEMP, we utilized the basic force balance of a microroller based on previous reports^{13,14,54}. When a microroller moves on a planar surface, it creates a propulsion force to its translational direction, (F_P), and gravitational force (F_G) on the bottom wall^{13,14,54}. If all the movement and forces were considered, the basic force balance of a microroller for F_P and F_G could be drawn as in Fig. R1.

Fig. R1. Force balance on a microroller.

The F_P and F_G are expressed as:

$$F_P = \pi\mu a^2 \omega \left(\frac{4}{5} \ln \frac{a}{\delta} - 1.516 \right) \quad (1)$$

$$F_G = \frac{4}{3} \pi (\rho_p - \rho_f) a^3 g \quad (2)$$

where μ dynamic viscosity of the fluid, a is the particle radius, ω is angular velocity, δ is lubrication or separation distance, ρ_p the density of the particle, ρ_f the density of the fluid, and g is the gravitational constant. For a $20 \mu\text{m}$ microroller, which is our case for the PEMPs, the F_G forces were calculated as $\sim 46.73 \text{ pN}$, and F_P for different lubrication distances were approximated as follows^{14,55}:

Fig. R2. F_P produced by $20 \mu\text{m}$ microroller for different lubrication distances.

As a result, both the propulsion and gravitational forces are in the order of pN. With an exaggerated assumption of contact area is $1 \mu\text{m}^2$, which would be much higher in real cases, the momentary stress applied by the microroller would be $<50 \text{ Pa}$. In comparison, mechanical neural stimulation requires threshold force $>200 \text{ nN}$ and pressure value $>5 \text{ kPa}$ via direct contact with atomic force microscopy cantilever⁵⁶. Therefore, it is reasonable to assume that the movement of microrollers on biological cells would not induce potentially harmful effects or induce changes in the excitability of the neurons."

We also added the corresponding discussion to the manuscript:

"In addition, we calculated the momentarily stress on the cells induced by the movement of the PEMP on the cellular layer⁵⁵ (Supplementary Note 2). The calculations revealed that the forces due to propulsion of the PEMP and gravitation were $<50 \text{ pN}$, and corresponding pressure of $<50 \text{ Pa}$ with the assumption of 1 mm^2 contact area between the PEMP and neurons. Resulting force and pressure values are an order of magnitude smaller than the thresholds found in the literature⁵⁶."

Having ruled out the potential heating and cavitation effects via experimental data and numerical calculations, we followed a systematic approach to determine the dominant electrophysiological basis of the neural response to PEMP excited with LIFU. As suggested by the reviewer, we started the control experiments with FUS stimulation without any particles and FUS + magnetic Janus particles (without BTNPs) by utilizing both current-clamp recordings (Supplementary Fig. 5) and calcium imaging experiments (Supplementary Fig. 14). We did not observe recording artefacts due to FUS or the presence of magnetic Janus microparticles (Supplementary Fig. 5) under 100 mW.cm^{-2} FUS intensity. Moreover, calcium imaging experiments showed no significant difference in the area under the curve for control groups, while the FUS+PEMP group showed significantly higher calcium intensities (Supplementary Fig. 14). Therefore, we ruled out the neural stimulation due to FUS excitation or due to magnetic Janus microparticles, and we can claim that barium titanate nanoparticles are the responsible transducers for neural stimulation.

Supplementary Fig. 9 Control experiments on primary neurons under LIFU with and without magnetic Janus particles. **a** Representative membrane potential traces of primary neurons excited under LIFU (two pulses of 5 ms 100 mW.cm^{-2} FUS) with and without magnetic Janus microparticles.

Supplementary Fig. 10 Control experiments on primary neurons under LIFU with and without magnetic Janus particles. Calcium responses of primary neurons without any particles, with magnetic Janus particles, and with PEMP under 50 ms, 50, and 100 mW.cm^{-2} FUS pulses ($n = 4$ independent experiments, each dot represents the mean of 10 individual neuron responses).

Additionally, we carried out a comprehensive channel blocker study using calcium imaging to understand the mechanism behind the stimulation of primary neurons via PEMP under FUS. We started the study without any channel blocker and compared the area under the curve of spontaneous activity of primary neurons, and their activity under FUS and FUS+PEMPs. We found no significant difference between the spontaneous activity and under FUS (Supplementary Fig. 15a and Fig. 3a). Secondly, we evaluated the neural response via the application of synaptic blockers (CNQX and AP5), which blocks postsynaptic excitatory receptors, AMPA and NMDA, respectively³⁹, to understand whether the neural response is autonomous or requires synaptic connections with other neurons⁴⁰. While we observed a reduction in the area under the curve for control samples without PEMP, we did not observe a statistically significant difference for neural stimulation induced by PEMP (Supplementary Fig. 15b). This shows that PEMP-based stimulation does not effectively require synaptic transmission between neurons.

Then, we started treating neurons with different channel blockers while the dose of tetrodotoxin (TTX), cadmium (Cd^{+2}), gadolinium (III) (Gd^{+3}), and ruthenium red (RR) was carefully chosen to avoid blocking other channels or altering cell excitability (Supplementary Fig. 15). We blocked the voltage-gated sodium channels with TTX and observed reduced calcium intensity during activation for both the control groups and the PEMP group. However, the reduction in the magnitude was significant for the PEMP group, indicating the role of voltage-gated sodium channels during neural activation (Supplementary Fig. 15c and Fig. 3b). Then, neural responses were recorded with the addition of Cd^{+2} (100, 200, and 400 μM), which prevents the activation of voltage-gated calcium channels^{36,41,42}. We observed a significant reduction in the magnitude for both control and PEMP groups for 400 μM Cd^{+2} concentration, for 100 μM Cd^{+2} the reduction was significant for only the PEMP group (Supplementary Fig. 15d and Fig. 3c). The residual calcium transients not blocked by TTX and Cd^{+2} , 24% and 19% of the peak magnitude (Supplementary Fig. 15c and d), respectively, could be attributed to the noncomplete block of the voltage-gated sodium and calcium channel population⁴³, to other calcium sources in primary hippocampal neurons or involvement of other channels, which we investigated next. While the control experiments in patch-clamp recording and calcium imaging under FUS without PEMP indicated that in the FUS intensity regime of interest, FUS itself did not induce any neural stimulation, we utilized gadolinium (III) to modify the deformability of the lipid bilayer and non-specific inhibition of mechanosensitive channels^{40,44,45}. We observed a small <15% decrease in the PEMP group which indicates that the mechanosensitive channel population is not dominant or active in neural stimulation via PEMP under FUS (Supplementary Fig. 15e and Fig. 3d). Since gadolinium (III) is not specific and changes the overall mechanical properties of neural membranes, Ruthenium Red was applied as a pore blocker of TRPV1, TRPV2, and TRPV4 channels^{40,43,46}. We did not observe a significant decrease in the calcium intensity or excitability for the PEMP-induced neuron group (Supplementary Fig. 15f and Fig. 3e). As a result, considering the TTX and Cd^{+2} blocker experiments, these observations indicate that PEMP under FUS increase the Na^{+} conductance in primary neurons and trigger Ca^{+2} transients dominantly via voltage-gated sodium and calcium channels. The potential effect of synaptic transmission and mechanical activation were not involved or significantly contributed to the neural stimulation. Therefore, the electrochemical characterization of a single PEMP response under FUS, experimental characterization of heating and calculation of mechanical effects, and channel blocker-dependent experiments suggest that the neural stimulation was induced via the opening of voltage-gated sodium and calcium channels. We added the information provided here to the results section in a new subsection, while the experimental procedures were added to the methods section.

Supplementary Fig. 11 PEMP-induced neural stimulation under LIFU is mediated by the voltage-gated sodium and calcium channels. **a** Calcium response under different LIFU intensities with and without the PEMPs ($n = 4$ independent experiments for control and experimental groups.). **b** Calcium responses before and after treatment with synaptic blockers, AP5 and CNQX. **c** Calcium responses before, after, and washout treatment with TTX. **d** Calcium responses before and after treatment with Cd^{2+} . **e** Calcium responses before and after treatment with Gd^{3+} . **f** Calcium responses before and after treatment with Ruthenium Red. For all experiments in **b**, **c**, **d**, **e**, and **f**, $n = 5$ independent experiments for control and experimental groups. Bar graph values are represented as mean \pm SD. Two-tailed, unpaired and paired T-tests were utilized for statistical analysis (* $p < 0.05$, ** $p < 0.01$, *** $p < 0.001$).

Fig. 3. PEMP-induced neural stimulation under LIFU is mediated by the voltage-gated sodium and calcium channels. **a** Comparison of area under the curve of calcium response of spontaneous activity and neural activity during FUS excitation with and without PEMPs ($n = 4$ independent experiments). **b** Neural activity changes before and after tetrodotoxin (TTX) treatment to block sodium channels. Quantification of area under the curve before and after **c** cadmium, **d** gadolinium (III), and **e** ruthenium red (RR) treatment to block calcium channels, global mechanosensitive channels nonspecifically, and TRPV channels, respectively. For all experiments in **b**, **c**, **d**, and **e**, $n = 5$ independent experiments for control and experimental groups. Bar graph values are represented as mean \pm SD. Two-tailed, unpaired and paired T-tests were utilized for statistical analysis (* $p < 0.05$, ** $p < 0.01$, *** $p < 0.001$).

We added the following text to the main manuscript:

“PEMPs induce piezoelectric activation of voltage-gated ion channels under low-intensity FUS excitation

Having investigated the neural stimulation performance of PEMP_s via patch-clamp electrophysiology experiments, we carried out a comprehensive channel blocker study to understand the mechanism behind the stimulation of primary neurons via PEMP_s under FUS. We started the calcium imaging experiments without any channel blocker and compared the area under the curve of spontaneous activity of primary neurons and their activity under FUS, FUS+magnetic Janus particles (Ni/Au coating on bare 20 μm silica microparticles, Supplementary Fig. 14), and FUS+PEMP_s (Supplementary Fig. 15a) conditions. We found no significant difference between the spontaneous activity and under FUS, while the area under the curve significantly increased via PEMP-induced effects (Fig. 3a). Secondly, we evaluated the neural response via the application of synaptic blockers (CNQX and AP5), which blocks postsynaptic excitatory receptors, AMPA and NMDA, respectively³⁹, to understand whether the neural response is autonomous or requires synaptic connections with other neurons⁴⁰. While a reduction in the area under the curve was observed for control samples without PEMP_s, we did not observe a statistically significant difference for neural stimulation induced by PEMP_s (Supplementary Fig. 15b). This shows that PEMP-based stimulation does not effectively require synaptic transmission between neurons. We then started treating neurons with different channel blockers while the dose of tetrodotoxin (TTX), cadmium (Cd⁺²), gadolinium (III) (Gd⁺³), and ruthenium red (RR) was carefully chosen to avoid blocking other channels or altering cell excitability (Supplementary Fig. 16c, d, e, and f). We blocked the voltage-gated sodium channels with TTX and observed reduced calcium intensity during activation for both the control groups and the PEMP group. However, the reduction in the magnitude was significant for the PEMP group, indicating the role of voltage-gated sodium channels during neural activation (Fig. 3b). Then, neural responses were recorded with the addition of Cd⁺² (100, 200, and 400 μM), which prevents the activation of voltage-gated calcium channels^{36,41,42}. We observed a significant reduction in the magnitude for both control and PEMP groups for 400 μM Cd⁺² concentration, for 100 μM Cd⁺² the reduction was significant for only the PEMP group (Supplementary Fig. 15d and Fig. 3c). The residual calcium transients not blocked by TTX and Cd⁺², 24% and 19% of the peak magnitude (Supplementary Fig. 15c and d), respectively, could be attributed to the noncomplete block of the voltage-gated sodium and calcium channel population⁴³, to other calcium sources in primary hippocampal neurons or involvement of other channels, which we investigated next. While the control experiments in patch-clamp recording and calcium imaging under FUS without PEMP_s indicated that, FUS itself did not induce any neural stimulation, we utilized gadolinium (III) to modify the deformability of the lipid bilayer and non-specific inhibition of mechanosensitive channels^{40,44,45}. We observed a small <15% decrease in the PEMP group which indicates that the mechanosensitive channel population is not dominant or active in neural stimulation via PEMP_s under FUS (Fig. 3d and Supplementary Fig. 15e). Since gadolinium (III) is not specific and changes the overall mechanical properties of neural membranes, Ruthenium Red was applied as a pore blocker of TRPV1, TRPV2, and TRPV4 channels^{40,43,46}. We did not observe a significant decrease in the calcium intensity or excitability for the PEMP-induced neuron group (Supplementary Fig. 16f and Fig. 3e). As a result, considering the TTX and Cd⁺² blocker experiments, these observations indicate that PEMP_s under FUS increase the Na⁺ conductance in primary neurons and trigger Ca⁺² transients dominantly via voltage-gated sodium and calcium channels. The potential effect of synaptic transmission and mechanical activation were not involved or not significantly contribute to the neural stimulation. Therefore, the electrochemical characterization of a single PEMP response under FUS, experimental

characterization of heating and calculation of mechanical effects, and channel blocker-dependent experiments suggest that the neural stimulation was induced via the opening of voltage-gated sodium and calcium channels.”

References

1. Jaganathan, H. & Godin, B. Biocompatibility assessment of Si-based nano- and micro-particles. *Adv. Drug Deliv. Rev.* **64**, 1800–1819 (2012).
2. Serda, R. E. *et al.* Cellular Association and Assembly of a Multistage Delivery System. *Small* **6**, 1329–1340 (2010).
3. Yu, T., Malugin, A. & Ghandehari, H. Impact of Silica Nanoparticle Design on Cellular Toxicity and Hemolytic Activity. *ACS Nano* **5**, 5717–5728 (2011).
4. Serda, R. E. *et al.* Mitotic trafficking of silicon microparticles. *Nanoscale* **1**, 250–259 (2009).
5. Moon, J. J., Huang, B. & Irvine, D. J. Engineering Nano- and Microparticles to Tune Immunity. *Adv. Mater.* **24**, 3724–3746 (2012).
6. Kersting, M. *et al.* Subtoxic cell responses to silica particles with different size and shape. *Sci. Rep.* **10**, 1–17 (2020).
7. Kusaka, T. *et al.* Effect of Silica Particle Size on Macrophage Inflammatory Responses. *PLoS One* **9**, e92634 (2014).
8. Bozuyuk, U. *et al.* High-Performance Magnetic FePt (L10) Surface Microrollers Towards Medical Imaging-Guided Endovascular Delivery Applications. *Adv. Funct. Mater.* **32**, (2022).
9. Cabanach, P. *et al.* Zwitterionic 3D-Printed Non-Immunogenic Stealth Microrobots. *Adv. Mater.* **32**, 1–11 (2020).
10. Palanker, D., Le Mer, Y., Mohand-Said, S., Muqit, M. & Sahel, J. A. Photovoltaic Restoration of Central Vision in Atrophic Age-Related Macular Degeneration. in *Ophthalmology* vol. 127 1097–1104 (Elsevier Inc., 2020).
11. Marino, A. *et al.* Piezoelectric Nanoparticle-Assisted Wireless Neuronal Stimulation. *ACS Nano* **9**, 7678–7689 (2015).
12. Raymond, J. L. *et al.* Broadband attenuation measurements of phospholipid-shelled ultrasound contrast agents. *Ultrasound Med. Biol.* **40**, 410–421 (2014).
13. Alapan, Y., Bozuyuk, U., Erkoc, P., Karacakol, A. C. & Sitti, M. Multifunctional surface microrollers for targeted cargo delivery in physiological blood flow. *Sci. Robot.* **5**, 1–11 (2020).
14. Bozuyuk, U. *et al.* Reduced rotational flows enable the translation of surface-rolling microrobots in confined spaces. *Nat. Commun.* **13**, 6289 (2022).
15. Savage, M. D. *A Laboratory Guide to Biotin-Labeling in Biomolecule Analysis. A Laboratory Guide to Biotin-Labeling in Biomolecule Analysis* vol. 7 (Birkhäuser Basel, 1995).
16. Donovan, J. W. & Ross, K. D. Increase in the stability of avidin produced by binding of biotin. Differential scanning calorimetric study of denaturation by heat. *Biochemistry* **12**, 512–517 (1973).

17. Ross, S. E., Carson, S. D. & Fink, L. M. Effects of detergents on avidin-biotin interaction. *Biotechniques* **4**, 350–354 (1986).
18. Baranov, M. V, Kumar, M., Sacanna, S., Thutupalli, S. & van den Bogaart, G. Modulation of Immune Responses by Particle Size and Shape . *Frontiers in Immunology* vol. 11 (2021).
19. O’Shea, T. M. *et al.* Foreign body responses in mouse central nervous system mimic natural wound responses and alter biomaterial functions. *Nat. Commun.* **11**, 6203 (2020).
20. Veiseh, O. *et al.* Size- and shape-dependent foreign body immune response to materials implanted in rodents and non-human primates. *Nat. Mater.* **14**, 643–651 (2015).
21. Anderson, J. M., Rodriguez, A. & Chang, D. T. Foreign body reaction to biomaterials. *Semin. Immunol.* **20**, 86–100 (2008).
22. Ravikumar, M. *et al.* The roles of blood-derived macrophages and resident microglia in the neuroinflammatory response to implanted Intracortical microelectrodes. *Biomaterials* **35**, 8049–8064 (2014).
23. Cadoni, S. *et al.* Ectopic expression of a mechanosensitive channel confers spatiotemporal resolution to ultrasound stimulations of neurons for visual restoration. *Nat. Nanotechnol.* **18**, 667–676 (2023).
24. Shapiro, M. G., Homma, K., Villarreal, S., Richter, C. P. & Bezanilla, F. Infrared light excites cells by changing their electrical capacitance. *Nat. Commun.* **3**, (2012).
25. Jiang, Y. *et al.* *Nongenetic optical neuromodulation with silicon-based materials.* *Nature Protocols* vol. 14 (Springer US, 2019).
26. Jiang, Y. *et al.* Rational design of silicon structures for optically controlled multiscale biointerfaces. *Nat. Biomed. Eng.* **2**, 508–521 (2018).
27. Yoo, S., Park, J. H. & Nam, Y. Single-Cell Photothermal Neuromodulation for Functional Mapping of Neural Networks. *ACS Nano* **13**, 544–551 (2019).
28. Colombo, E., Feyen, P., Antognazza, M. R., Lanzani, G. & Benfenati, F. Nanoparticles: A Challenging Vehicle for Neural Stimulation . *Frontiers in Neuroscience* vol. 10 (2016).
29. Krasovitski, B., Frenkel, V., Shoham, S. & Kimmel, E. Intramembrane cavitation as a unifying mechanism for ultrasound-induced bioeffects. *Proc. Natl. Acad. Sci. U. S. A.* **108**, 3258–3263 (2011).
30. Dalecki, D. Mechanical bioeffects of ultrasound. *Annu. Rev. Biomed. Eng.* **6**, 229–248 (2004).
31. O’Brien, W. D. Ultrasound-biophysics mechanisms. *Prog. Biophys. Mol. Biol.* **93**, 212–255 (2007).
32. Dell’Italia, J., Sanguinetti, J. L., Monti, M. M., Bystritsky, A. & Reggente, N. Current State of Potential Mechanisms Supporting Low Intensity Focused Ultrasound for Neuromodulation. *Front. Hum. Neurosci.* **16**, 1–23 (2022).
33. Med, A. I. U. *et al.* *Acoustic output measurement and labeling standard for diagnostic ultrasound equipment.* *Am. institute of ultrasound in medicine* (American Institute of Ultrasound in Medicine, 1992).
34. Rabut, C. *et al.* Ultrasound Technologies for Imaging and Modulating Neural Activity. *Neuron* **108**, 93–110 (2020).
35. Tufail, Y. *et al.* Transcranial Pulsed Ultrasound Stimulates Intact Brain Circuits. *Neuron* **66**,

- 681–694 (2010).
36. Tyler, W. J. *et al.* Remote excitation of neuronal circuits using low-intensity, low-frequency ultrasound. *PLoS One* **3**, (2008).
 37. Qiu, Z. *et al.* The Mechanosensitive Ion Channel Piezo1 Significantly Mediates In Vitro Ultrasonic Stimulation of Neurons. *iScience* **21**, 448–457 (2019).
 38. Plaksin, M., Shoham, S. & Kimmel, E. Intramembrane cavitation as a predictive bio-piezoelectric mechanism for ultrasonic brain stimulation. *Phys. Rev. X* **4**, 1–10 (2014).
 39. Yoo, S., Hong, S., Choi, Y., Park, J.-H. & Nam, Y. Photothermal Inhibition of Neural Activity with Near-Infrared-Sensitive Nanotransducers. *ACS Nano* **8**, 8040–8049 (2014).
 40. Yoo, S., Mittelstein, D. R., Hurt, R. C., Lacroix, J. & Shapiro, M. G. Focused ultrasound excites cortical neurons via mechanosensitive calcium accumulation and ion channel amplification. *Nat. Commun.* **13**, 493 (2022).
 41. Swandulla, D. & Armstrong, C. M. Calcium channel block by cadmium in chicken sensory neurons. *Proc. Natl. Acad. Sci. U. S. A.* **86**, 1736–1740 (1989).
 42. Chow, R. H. Cadmium block of squid calcium currents: Macroscopic data and a kinetic model. *J. Gen. Physiol.* **98**, 751–770 (1991).
 43. Albert, E. S. *et al.* TRPV4 channels mediate the infrared laser-evoked response in sensory neurons. *J. Neurophysiol.* **107**, 3227–3234 (2012).
 44. Yang, X. C. & Sachs, F. Block of stretch-activated ion channels in xenopus oocytes by gadolinium and calcium ions. *Science (80-.).* **243**, 1068–1071 (1989).
 45. Jerusalem, A. *et al.* Electrophysiological-mechanical coupling in the neuronal membrane and its role in ultrasound neuromodulation and general anaesthesia. *Acta Biomater.* **97**, 116–140 (2019).
 46. Wang, Y., Garg, R., Cohen-Karni, D. & Cohen-Karni, T. Neural modulation with photothermally active nanomaterials. *Nat. Rev. Bioeng.* **1**, 193–207 (2023).
 47. Parameswaran, R. *et al.* Photoelectrochemical modulation of neuronal activity with free-standing coaxial silicon nanowires. *Nat. Nanotechnol.* **13**, 260–266 (2018).
 48. Kim, T. *et al.* Deep brain stimulation by blood–brain-barrier-crossing piezoelectric nanoparticles generating current and nitric oxide under focused ultrasound. *Nat. Biomed. Eng.* (2022) doi:10.1038/s41551-022-00965-4.
 49. Boehler, C., Carli, S., Fadiga, L., Stieglitz, T. & Asplund, M. Tutorial: guidelines for standardized performance tests for electrodes intended for neural interfaces and bioelectronics. *Nat. Protoc.* **15**, 3557–3578 (2020).
 50. Jiang, Q. *et al.* Temporal neuromodulation of retinal ganglion cells by low-frequency focused ultrasound stimulation. *IEEE Trans. Neural Syst. Rehabil. Eng.* **26**, 969–976 (2018).
 51. Amin, H. *et al.* Electrical responses and spontaneous activity of human iPSC-derived neuronal networks characterized for 3-month culture with 4096-electrode arrays. *Front. Neurosci.* **10**, (2016).
 52. Lowet, E. *et al.* Deep brain stimulation creates informational lesion through membrane depolarization in mouse hippocampus. *Nat. Commun.* **13**, 7709 (2022).
 53. Huang, Y. *et al.* Bioresorbable thin-film silicon diodes for the optoelectronic excitation and

- inhibition of neural activities. *Nat. Biomed. Eng.* **7**, 486–498 (2023).
54. Bozuyuk, U., Alapan, Y., Aghakhani, A., Yunusa, M. & Sitti, M. Shape anisotropy-governed locomotion of surface microrollers on vessel-like microtopographies against physiological flows. *Proc. Natl. Acad. Sci. U. S. A.* **118**, 1–10 (2021).
 55. Bozuyuk, U., Yildiz, E., Han, M., Demir, S. O. & Sitti, M. Size-Dependent Locomotion Ability of Surface Microrollers on Physiologically Relevant Microtopographical Surfaces. *Small* **2303396**, 1–12 (2023).
 56. Gaub, B. M. *et al.* Neurons differentiate magnitude and location of mechanical stimuli. *Proc. Natl. Acad. Sci. U. S. A.* **117**, 848–856 (2020).
 57. Ghezzi, D. *et al.* A polymer optoelectronic interface restores light sensitivity in blind rat retinas. *Nat. Photonics* **7**, 400–406 (2013).
 58. Han, M. *et al.* Photovoltaic neurointerface based on aluminum antimonide nanocrystals. *Commun. Mater.* **2**, 19 (2021).
 59. Damjanovic, R., Bazard, P., Frisina, R. D. & Bhethanabotla, V. R. Hybrid Electro-Plasmonic Neural Stimulation with Visible-Light-Sensitive Gold Nanoparticles. *ACS Nano* **14**, 10917–10928 (2020).
 60. Jiang, Y. *et al.* Nongenetic optical neuromodulation with silicon-based materials. *Nature Protocols* vol. 14 (2019).
 61. Rand, D. *et al.* Direct Electrical Neurostimulation with Organic Pigment Photocapacitors. *Adv. Mater.* **30**, 1707292 (2018).
 62. Tufail, Y., Yoshihiro, A., Pati, S., Li, M. M. & Tyler, W. J. Ultrasonic neuromodulation by brain stimulation with transcranial ultrasound. *Nat. Protoc.* **6**, 1453–1470 (2011).
 63. Ghezzi, D. *et al.* A hybrid bioorganic interface for neuronal photoactivation. *Nat. Commun.* **2**, 166 (2011).
 64. Karatum, O. *et al.* RuO₂ Supercapacitor Enables Flexible, Safe, and Efficient Optoelectronic Neural Interface. *Adv. Funct. Mater.* **32**, 2109365 (2022).
 65. Han, M. *et al.* Tissue-Like Optoelectronic Neural Interface Enabled by PEDOT:PSS Hydrogel for Cardiac and Neural Stimulation. *Adv. Healthc. Mater.* **11**, 2102160 (2022).
 66. Schiavone, G. *et al.* Guidelines to Study and Develop Soft Electrode Systems for Neural Stimulation. *Neuron* **108**, 238–258 (2020).
 67. Cogan, S. F. Neural stimulation and recording electrodes. *Annu. Rev. Biomed. Eng.* **10**, 275–309 (2008).
 68. Nakano, Y. *et al.* Sinusoidal electrical pulse more efficiently evokes retinal excitation than rectangular electrical pulse in retinal prostheses. *Sensors Mater.* **29**, 1667–1677 (2017).
 69. Twyford, P. & Fried, S. The Retinal Response to Sinusoidal Electrical Stimulation. *IEEE Trans. Neural Syst. Rehabil. Eng.* **24**, 413–423 (2016).
 70. McIntyre, C. C. & Grill, W. M. Extracellular stimulation of central neurons: Influence of stimulus waveform and frequency on neuronal output. *J. Neurophysiol.* **88**, 1592–1604 (2002).

REVIEWERS' COMMENTS

Reviewer #1 (Remarks to the Author):

This paper is acceptable now.

Reviewer #3 (Remarks to the Author):

The authors did many further experiments/simulations that further improved the manuscript. I suggest acceptance of the manuscript.

As a minor check, the authors might look at the color matching for legends in Figure S7.

Reviewer #4 (Remarks to the Author):

The authors have done an impressive job in their revision to address my concerns, I read the new results with great interest. I think this revised version makes important strides in the direction of helping to explain nano/microparticle mediated neurostimulation in general.

The key concern I expressed regarding control experiments with FUS alone has been unequivocally resolved.

The results concerning apparent net current generation (via a Faradaic mechanism) are convincing, though of course the deeper mechanism remains unresolved. These results make me wonder about old ideas of the "summation effect" proposed by Gildemeister in the 1930s and 1940s of high-frequency stimulation on cell membranes, where an AC signal results in net depolarization of excitable cells at time-scales up to a few seconds:

Gildemeister M. Zur theorie des elektrischen reizes, V: polarisation durch wechselstro"me. Ber Sachs Ges Wiss. 1930;81:303–313.

Gildemeister M. Untersuchungen u"ber die wirkung der mittelfrequenzstro"me auf den menschen. Pflugers Arch. 1944;247: 366–404.

In the present form, I enthusiastically support publication and commend the authors on a strong effort to greatly improve their paper.

Response to Reviewers

article: NCOMMS-23-36212A

Revision summary:

We would like to thank all reviewers for their constructive comments, questions and the suggestions for publication.

In the following, we have addressed the reviewer comments point by point in black, where the reviewer comments are shown in blue.

REVIEWERS' COMMENTS

Reviewer #1 (Remarks to the Author):

This paper is acceptable now.

Response: We thank the reviewer for their suggestion for publication.

Reviewer #3 (Remarks to the Author):

The authors did many further experiments/simulations that further improved the manuscript. I suggest acceptance of the manuscript.

As a minor check, the authors might look at the color matching for legends in Figure S7.

Response: We thank the reviewer for their suggestion for publication. We checked and corrected the color matching for the mentioned figure and all other figures in the manuscript.

Reviewer #4 (Remarks to the Author):

The authors have done an impressive job in their revision to address my concerns, I read the new results with great interest. I think this revised version makes important strides in the direction of helping to explain nano/microparticle mediated neurostimulation in general. The key concern I expressed regarding control experiments with FUS alone has been unequivocally resolved.

The results concerning apparent net current generation (via a Faradaic mechanism) are convincing, though of course the deeper mechanism remains unresolved. These results make me wonder about old ideas of the "summation effect" proposed by Gildemeister in the 1930s and 1940s of high-frequency stimulation on cell membranes, where an AC signal results in net depolarization of excitable cells at time-scales up to a few seconds:

Gildemeister M. Zur theorie des elektrischen reizes, V: polarisation durch wechselstro"me. Ber Sachs Ges Wiss. 1930;81:303–313. Gildemeister M. Untersuchungen u"ber die wirkung der mittelfrequenzstro"me auf den menschen. Pflugers Arch. 1944;247: 366–404.

In the present form, I enthusiastically support publication and commend the authors on a strong effort to greatly improve their paper.

Response: We thank the reviewer for their suggestion for publication. The control and additional experiments suggested by the reviewer greatly improved the depth of the study and the manuscript. We again thank the reviewer for their constructive comments and questions. We will continue to explore the mechanism of action in our current and next studies.